# The complex basal morphology and ice dynamics of Nansen Ice Shelf, East Antarctica

Christine F. Dow[1], Derek Mueller[2], Peter Wray[1], Drew Friedrichs[3,4], Alexander L. Forrest[3,4], Jasmin B. McInerney[3,4], Jamin Greenbaum[5], Donald D. Blankenship[6], Choon Ki Lee[7], Won Sang Lee[7]

[1]Department of Geography and Environmental Management, University of Waterloo, Waterloo, Canada, N2L 3G1
[2]Department of Geography and Environmental Studies, Carleton University, Ottawa, Canada, K1S 5B6
[3]Department of Civil and Environmental Engineering, University of California – Davis, Davis, USA, 95616
[4]Tahoe Environmental Research Center, University of California – Davis, Incline Village, USA, 89451
[5]Scripps Institution of Oceanography, University of California, San Diego, La Jolla, CA, USA
[6]Institute for Geophysics, Jackson School of Geosciences, University of Texas at Austin, Austin, TX, USA, 78758
[7]Division of Glacial Environment Research, Korea Polar Research Institute, Incheon 21990, Republic of Korea

*Correspondence to*: Christine F. Dow (christine.dow@uwaterloo.ca)

**Abstract.** Ice shelf dynamics and morphology play an important role in the stability of floating bodies of ice by driving fracturing that can lead to calving, in turn impacting the ability of the ice shelf to buttress upstream grounded ice. Following a 2016 calving event at Nansen Ice Shelf, East Antarctica, we collected airborne and ground-based radar data to map ice thickness across the shelf. We combine these data with published satellite-derived data to examine the spatial variations in ice shelf draft, the cause and effects of ice shelf strain rates, and the possibility that a suture zone may be channelizing ocean water and altering patterns of sub-ice shelf melt and freeze-on. We also use our datasets to assess limitations that may arise from relying on hydrostatic balance equations applied to ice surface elevation to determine ice draft morphology. We find that the Nansen Ice Shelf has a highly variable basal morphology driven primarily by the formation of basal fractures near the onset of ice shelf suture zone. This morphology is reflected in the ice shelf strain rates but not in the calculated hydrostatic balance thickness, which underestimates the scale of variability at the ice shelf base. Enhanced melt rates near the ice shelf terminus and in steep regions of the channelized suture zone, along with relatively thin ice in the suture zone, appear to represent vulnerable areas in the NIS. This morphology, combined with ice dynamics, induce strain that has led to the formation of transverse fractures within the suture zone, resulting in large-scale calving events Similar transverse fractures at other Antarctic ice shelves may also be driven by highly variable morphology, and predicting their formation and evolution could aid projections of ice shelf stability.

## 1 Introduction

Ice shelves buttress large portions of the Antarctic Ice Sheet, preventing destabilization of grounded ice and related discharge into the ocean (Dupont and Alley, 2005; Fürst et al., 2016; Scambos et al., 2014). The stability of these ice shelves is therefore vital for determining the future rise of global sea level. Fracturing and calving are key processes in the evolution of ice shelf morphology and its ability to hold back grounded ice. Ice shelf fractures occur in regions that are under stress greater than the cohesive strength of the ice and can form or expand as a result of flexure (Vaughan et al., 2012); longitudinal stretching as ice moves beyond pinning points (Dow et al., 2018; Indrigo et al., 2020); advection of crevasses from grounded ice across the grounding line (Kulessa et al., 2014); and changes in buttressing causing ice shelf acceleration (Fürst et al., 2016). The primary controls of fracture formation include the ice thickness, the ice strain regime, and the ice rheology, along with additional factors such as the stress intensity factor (Lipovsky, 2018).

It is possible to predict the response of a flat body of floating ice to warming atmospheric and ocean conditions but, in reality, ice shelves consist of complex morphology and rheology making it a challenge to determine potential fracture and/or break-up locations. The increasing availability of high-resolution data products such as ice velocity (e.g., the Global Land Ice Velocity Extraction from Landsat 8 (GoLIVE) project, Scambos et al. (2016); and the Inter-Mission Time Series of Land Ice Velocity and Elevation (ITS_LIVE) project, Gardner et al. (2020)), and ice surface elevation (e.g., the Reference Elevation

Model of Antarctica (REMA, Howat et al., 2019)) opens up possibilities for large-scale spatial analyses of ice shelf processes. However, the use of inferred data (e.g., assumptions of hydrostatic equilibrium for floating ice bodies) may obfuscate ice shelf morphological details and lead to misconceptions regarding important processes and properties that could lead to fracturing.

The Nansen Ice Shelf (NIS) in Terra Nova Bay underwent a large (213 km$^2$ in ice extent) calving event in 2016. This was

driven by a transverse fracture that formed around 1987 in a thin suture zone in the centre of the ice shelf, which advected towards the shelf terminus while expanding laterally. Around the time of the calving event, a new fracture formed over the thinnest region of the central NIS (Dow et al., 2018). Both the initial location of the fractures and their rate of lateral expansion have been suggested to be linked to the presence of a basal channel causing thinning in the suture zone, the thickness of the ice at the edge of the suture zone, and the ice surface strain patterns (Dow et al., 2018). Here, we investigate the conditions

facilitating ice shelf fracture by analyzing ground-based and airborne ice-penetrating radar (IPR), and ice surface global navigation satellite system (GNSS) data that were collected around six months following the large calving event at NIS to assess ice shelf morphology and thinning rates. Finally, we use published satellite-derived digital elevation model (DEM), velocity, and basal melt products to examine the larger scale morphology of the NIS, calculate ice surface strain rates, and compare satellite-derived melt estimates with our radar-based calculations


We are interested in two primary questions about Nansen Ice Shelf. The first is how does basal morphology relate to surface topography and influence the distribution of marine and meteoric ice? The second is whether there is evidence of melt at the base of this cold-cavity ice shelf and, if so, does it occur within the suture zone. Both of these questions can provide information about features useful to identify in other ice shelves around Antarctica that may be susceptible to fracture, and for modelling

approaches to ice shelf cavities.

## 2. Site Description

The NIS, with an area of ~1800 km$^2$, is located in Terra Nova Bay and is part of the Victoria Land Coast of the Western Ross Sea region in East Antarctica. The two primary glaciers that supply the NIS are the Reeves Glacier to the south and the Priestley

Glacier to the north (Fig. 1a). The Priestley Glacier is up to ~ 900 m thick near its grounding line (Morlighem et al., 2020) and has ice surface velocities up to ~180 m a$^{-1}$ (Gardner et al., 2020). The Reeves Glacier separates into two distinct branches as it flows around the Teall Nunatak and down a substantial ice fall at its grounding line. The Reeves Glacier is up to ~ 900 m thick at its grounding line (Morlighem et al., 2020) with a surface ice velocity up to ~320 m a$^{-1}$ (Gardner et al., 2020). The grounding lines of the Priestley and Reeves glaciers have been estimated to be 70 km and 40 km from the ice shelf terminus, respectively

(Mouginot et al., 2017).

The NIS ranges in thickness from 900 m at the Priestley Glacier grounding line to 120 m at the ice shelf terminus. The thinnest region is found in a 30 km-long surface depression (Fig. 2a), which runs east-west across the NIS along a suture zone where the floating ice from the Reeves and Priestley glacier branches converge (Alley et al., 2016; Bell et al., 2017, Baroni et al., 1991; Dow et al., 2018). A supraglacial river was observed in the surface depression as early as 1974 and annually from 2014-

2016 (Bell et al., 2017). During the melt seasons of 2014-2016 the river flowed into the transverse fracture prior to the calving event in April 2016 (Dow et al., 2018).

Inexpressible Island is a small land mass and pinning point located at the north-eastern end of the ice shelf terminus dividing the main NIS from the smaller Hell's Gate Ice Shelf. A polynya stretches from Hell's Gate Ice Shelf, along the terminus of NIS, and reaches the northern margin of the Drygalski Ice Tongue. This polynya plays an important role in sea ice production

for the Ross Sea (Stevens et al., 2017) and is formed by katabatic winds, which also strip the NIS of much of its snow and firn cover leaving a significant portion of the surface as blue ice (Kurtz and Bromwich, 1983).

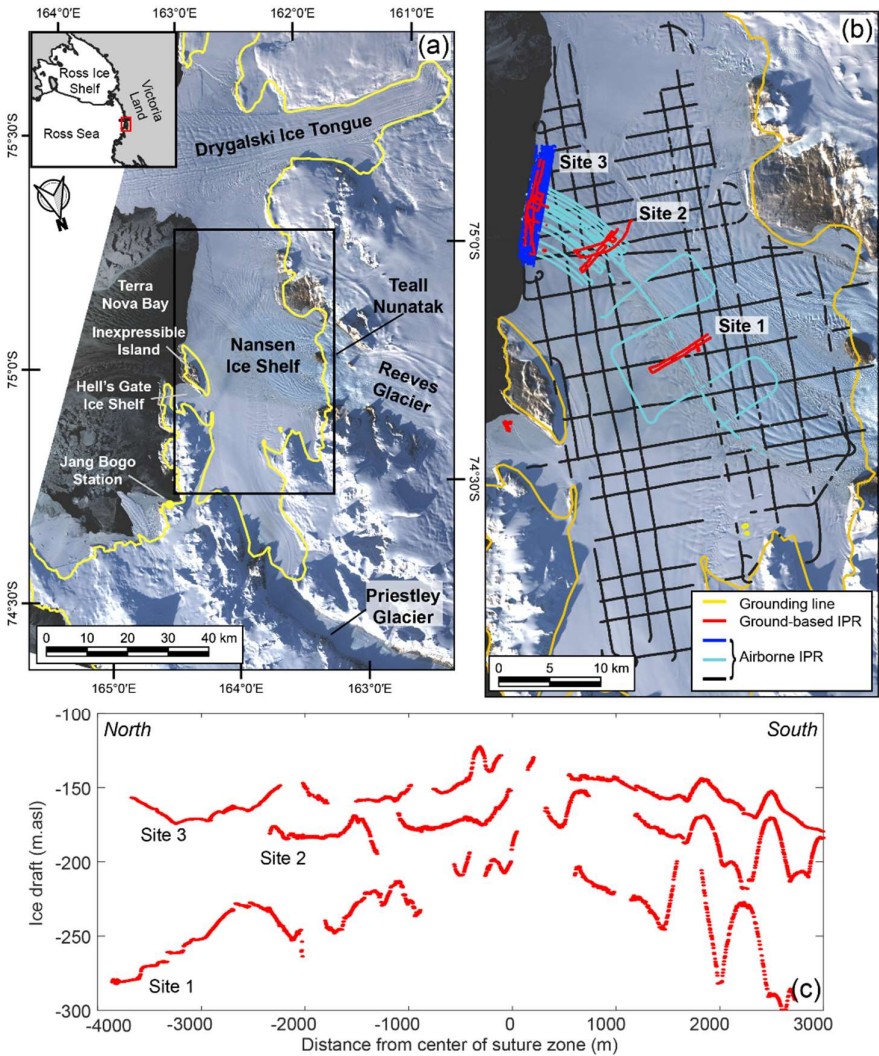

**Figure 1:** Site map with radar lines and cross sections. (a) The Nansen Ice Shelf (NIS) and grounding line/coastline (yellow line) from MEaSUREs Antarctic Boundaries v2 (Mouginot et al., 2017) on a background image from Landsat-8 acquired

March 22, 2017. Extent of (b) is shown by the black box. Inset: Victoria Land and western Ross Sea region, red box is the location of the NIS. (b) Ground-based ice-penetrating radar (IPR) lines at sites 1, 2 and 3 from November 2016 (red), Airborne IPR survey from January 2017 focused on the ice shelf terminus (dark blue) and suture zone (cyan), Airborne IPR survey from January 2017 of the entire NIS (black). (c) Three ice draft lines, one from each site. The location of these transects are shown in Fig. 2a.

## 3 Data sets

### 3.1 Ice penetrating radar

Between November 3-19 2016, ~80 km of ground-based ice penetrating radar (IPR) surveys were conducted on foot at three targeted sites focused on the suture zone of the NIS (Fig. 1b) resulting in a trace separation of ~1 m (see Appendix A), later down-sampled to a separation of 4 m. Some locations lacked an ice base reflection, especially at Site 3 closest to the terminus, and repeat surveys verified that it was location specific. Ice surface elevation and precise location data were collected simultaneously with the radar data using a GNSS receiver mounted on the IPR rig.

In addition to the concentrated ground-based radar surveys, a helicopter-based airborne geophysical survey was based out of the South Korean Jang Bogo Station between December 25 2016 and February 18 2017 (Lindzey et al., 2020; see Appendix A). In total, 1000 km of surveys were conducted over the NIS in both a grid-pattern and concentrated over the suture zone near the ice shelf terminus (Fig. 1 a, b). An error analysis was conducted using the cross-over surface elevations and ice thickness of both the ground-based and airborne radar and is detailed in Appendix A.

### 3.2 Digital Elevation Maps

Although the radar data provide information on the ice thickness and draft in our focused sites on the NIS, it is too sparse to accurately interpolate. As an alternative, for larger-scale morphological assessment of the ice shelf we use the 8 m resolution Reference Elevation Model of Antarctica (REMA) mosaic (Howat et al., 2019) for the NIS (Appendix Fig. C1a). We corrected the mosaic to the GLO04C geoid and applied hydrostatic calculations to invert for basal draft (*B)* (Fig. 2a), using the equation:

$$B = S - \frac{S \cdot \rho_s}{(\rho_s - \rho_i)}, \tag{1}$$

where $S$ is the measured surface elevation above the GL04C geoid, $\rho_s$ is the density of sea water (1028 kg m$^{-3}$) and $\rho_i$ is the density of ice (917 kg m$^{-3}$). The zone of firn-free blue ice covers the regions of ground-based radar survey but firn is present towards the grounding line of Reeves Glacier and Inexpressible Island, and therefore in these regions the hydrostatic calculations are less accurate. We lack information on the thickness of firn in these regions precluding the inclusion of different density profiles in our application of Eq. 1. Assuming a REMA surface elevation error of <1 m (Howat et al., 2019), the hydrostatic thickness in the firn-free region has an associated error of ± ~10 m. We also apply Eq. 1 to our in situ GNSS ice surface elevation data to compare directly with the recorded radar ice thickness (Fig. 3).

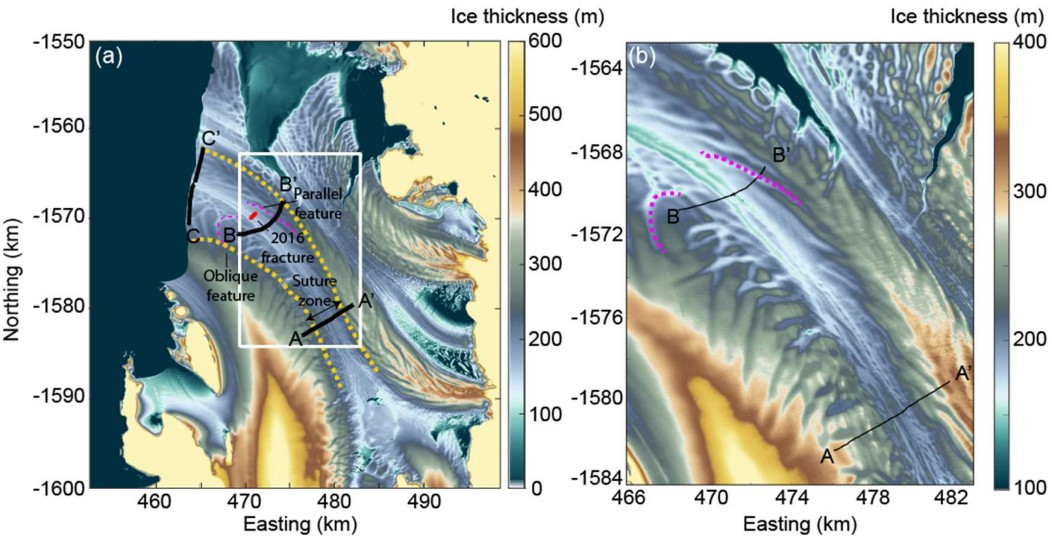

**Figure 2:** (a) Digital ice thickness model derived from REMA ice surface elevation and hydrostatic calculations with the three transects in Fig. 1b and Fig. 3 plotted in black., assuming no firn (which is valid over much of the ice shelf). The red line shows the 2016 full-thickness fracture and the purple dashed lines show examples of basal fractures. The extent of the suture zone is outlined with yellow dashed lines. (b) Zoom in of the ice thickness DEM as outlined by the white box in (a).

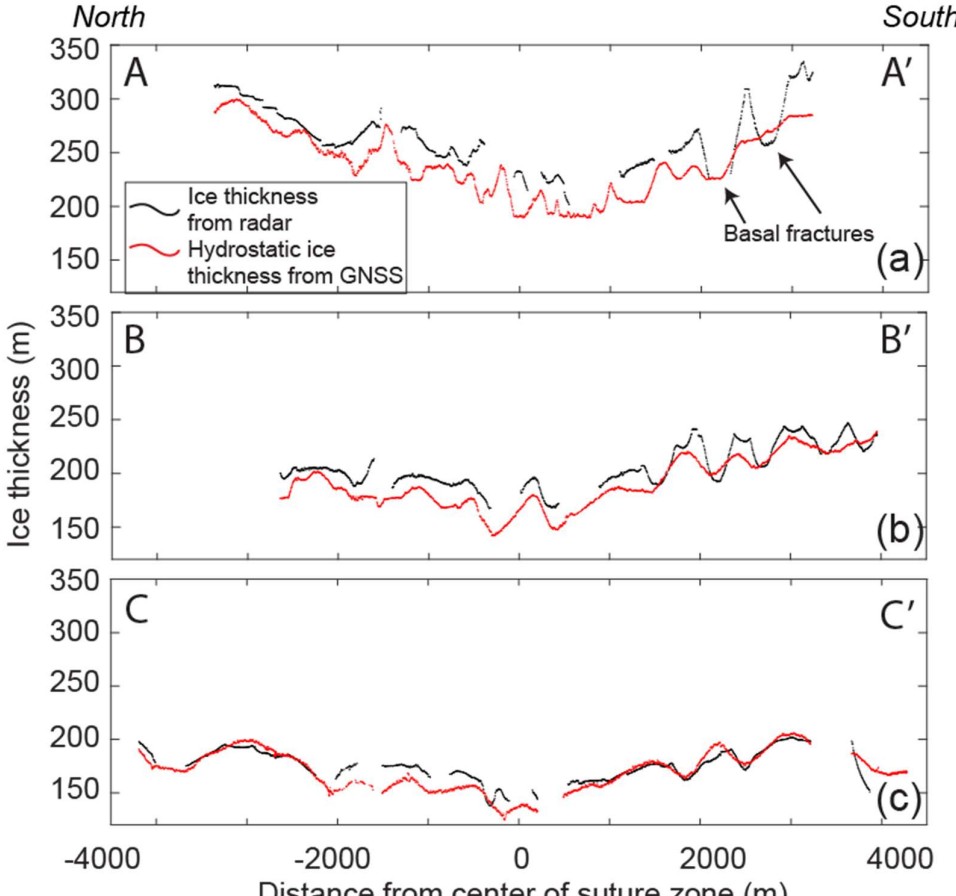

**Figure 3:** Cross-ice shelf transects of ice thickness from radar measurement (black) and from hydrostatic balance calculations using in situ GNSS surface elevations (red) at (a) Site 1, (b) Site 2 and (c) Site 3 on the NIS. The transects are marked on Fig. 2a.

### 3.3 Strain rates

NIS ice surface velocity data at 300 m resolution were obtained from GoLIVE (Scambos et al., 2016). The closest nearly cloud-free period (September 29 – October 31, 2016) prior to the November 2016 IPR surveys was selected for primary analysis. GoLIVE velocities were used here instead of ITS_LIVE velocities because the latter has an annual temporal coverage between image pairs, which obscured smaller-scale velocity features with short wavelengths. We calculated the principal strain in the longitudinal and transverse directions, after rotating to the direction of local flow, using equations detailed in Bindschadler et al (1996) and Alley et al. (2018), with a length scale of 300 m, the pixel size of the GoLIVE velocity data (Fig. 4b,c). Our approach of applying a length scale the same size as the pixel size follows the suggestions of Alley et al (2018) who state that such an approach is appropriate for ice shelves with highly variable basal draft topography. It is not possible to stack multiple velocity outputs in this region to reduce noise due to the combination of ice flow speed and the limited spatial extent

of the ice shelf morphological features. As an alternative, we demonstrate consistency of NIS transverse strain features across multiple years. These are shown in Appendix Fig. C2. where we also plot the larger-scale regional transverse strain rates from October 8-October 9, 2019, along with shear, longitudinal and transverse strain rates from October 10 – November 11, 2020.

We interpolated the 2016 strain values from the 300 m length scale calculation to the radar data locations to assess the relationship between topography and ice surface strain on the NIS (Fig. 4).

## 4 Results and analysis

### 4.1 Nansen Ice Shelf morphology

Our first aim is to examine the morphology of the Nansen Ice Shelf. Satellite products providing surface topography in the Antarctic are now high resolution (8 m horizontal from REMA mosaic) and provide excellent detail for analyses of surficial processes (Appendix Fig. C1a). Using REMA hydrostatic thickness for the wider region and our IPR data where available, the suture zone where the Priestley and Reeves glaciers first converge has an ice thickness of 200 m. The initial width of this thin-ice region is ~4.5 km with a difference in thickness of 50 m between the thin center and the thicker margin of the suture zone.

Within 15 km of the terminus, the thin-ice region widens to 10 km with a draft difference of 90 m and 60 m, respectively, between the center of the suture zone and the northern and southern suture zone margins. At the terminus, the northern flank of the thin-ice region is wider (6 km) than the southern flank (4 km) with an average ice draft slope of ~1.3˚ and ~1.1˚ at the ice shelf terminus, respectively (Fig. 2c).

The combination of the larger-scale DEM and the smaller-scale IPR data indicate that there is a complex morphology consisting of multiple basal fractures, within the suture zone (Bell et al., 2017). These fractures originate upstream in the respective floating margins of Priestley and Reeves ice, prior to the convergence of the ice bodies. Within the suture zone, the fractures are discrete semi-continuous features that run parallel (on the southern side) or oblique (on the northern side) to the center of the suture zone (Fig. 2b and Fig. 3.). These basal features are altered as they advect downstream but we continue to

refer to them as basal fractures for consistency. The southern flank fractures are always parallel to the suture zone but are discontinuous and range from 15 – 25 m in height with a width of ~500 m on average (as determined from our radar data; see Fig. 3). On the northern flank, the fractures are also ~20 m high and ~500 m wide (Fig. 3) and are 3-5 km long near the ice shelf terminus (Fig. 2a). In contrast to the southern fractures, these sweep away from the center of the suture zone to the north following a semi-circular pattern, and curve up to 90˚ from their original orientation.


There were several regions on the NIS where the ice-ocean boundary was not visible in the airborne and ground-based radar record (Fig. 5 a,d). Comparing these regions with the REMA hydrostatic thickness map shows they are associated with the thinnest region in the suture zone and with the basal fractures on both sides of the suture zone (Fig. 5a,b). Given the abundance of clear ice base reflectors in the remainder of the radar data, it is likely that these "echo-free zones" represent

areas of marine ice and/or frazil ice accumulation (Holland et al., 2009). Typically, the unconformity between meteoric ice and the subjacent horizontal layer of accumulating marine ice will reflect radar waves allowing this boundary to be identified (Fricker et al., 2001; Kulessa et al. 2014). Since marine ice is a lossy medium (Tulaczyk and Foley, 2020), no radio waves are returned below this depth and marine ice thickness can only be inferred from hydrostatic calculations. In our radargrams, there are regions where little to no signal is returned, except the signal associated with the air/ground wave near the surface

(Fig 5e and f). This complete absorption of all the radar energy in these locations can only be explained by marine ice near the upper surface of the ice shelf. A healed rift or very tall/high basal crevasse filled with marine ice, would represent a vertical section where radar waves are absorbed (c.f., Hillebrand et al. 2021). On the ice surface, in the suture zone, there are many stripes of white and clear blue ice that are associated with echo-free zones (Fig. 6c). These can be traced back to the

Reeves Glacier ice fall where crevasses fully fracture through the ice column, fill with sea water and refreeze (e.g.,

Khazendar et al., 2001, Tison et al, 2001). The filled rifts are then advected and stretched, producing visible stripes parallel

to the ice flow direction along the suture zone. Khazendar et al. (2001) postulate that ice folding due to lateral compression

may alter ice stratigraphy and dip angle.  These processes may contribute to the blue/white striping seen in the suture zone

and also imply that whatever ice is visible at the surface is not necessarily the same as the ice beneath.

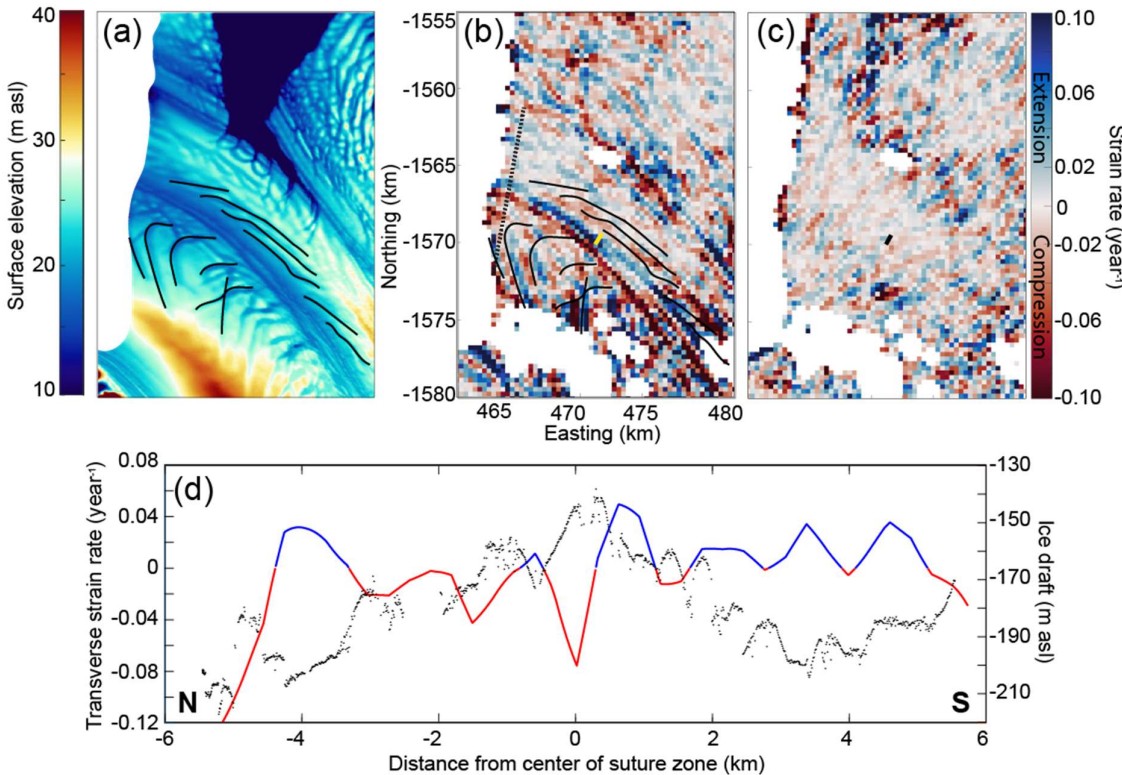

**Figure 4**: (a) NIS REMA mosaic surface elevation DEM. The black lines outline basal fracture locations. NIS ice surface (b)

transverse and (c) longitudinal strain rate calculations from 2016 GoLIVE velocity data. The black dashed line is the extent of

the airborne radar transect shown in (c). The yellow/black line shows the location of the new 2016 fracture. (d) Profile of the

radar transect showing ice shelf draft (black) and strain rate (red: compression; blue: extension) interpolated along the transect.

The extent of the region in Fig. 4b and c is shown in Fig. 2a.

In order to determine how the complex basal morphology of the ice shelf impacts compressional and extensional strain we

show horizontal transverse strain rates derived from 2016 GoLIVE ice surface velocity of the NIS (Fig. 4b) and a cross-section

of strain rates plotted against the basal draft (Fig. 4d). We note that the patterns of strain presented here are also visible in NIS

strain rate maps from multiple years, at different times of the year, and therefore appear to persist over time (Appendix

Fig.C2).Along the center of the suture zone there is an alternating region of transverse compression (red) on the northern side

and extension (blue) on the southern side: both regions have widths of ~800 m. When compared with the ice shelf draft, the

switch between compression and extension occurs at the apex of the thin-ice suture zone region (Fig. 4d), although with a

strain pixel size of 300 m there may be some error in the location of this transition. The transverse surface fracture, which

formed in 2016 is located within the strongly extensional side of the suture zone (Fig. 4b). To the north, where the basal

fractures curve, there is compression at the apexes of the basal fractures and extension at the keels. The patterns of strain can

be seen in the 2D map (Fig. 4b) as well as in the cross-section with the radar-derived basal draft (Fig. 4d). On the southern

side of the suture zone, the relationship between basal fractures and strain is less clear with compression in the apex of the first fracture at 1500 m from the center and again at 2800 m from the center at the apex of another fracture. Otherwise, the strain is generally extensional. Similarly, in the 2D map, the strain patterns do not seem to align particularly well with the fractures on the Reeves side. We also show the map of longitudinal strain rate in Fig. 4c. The pattern and signal in the longitudinal strain rate is poorly resolved compared to the transverse strain rate in Fig. 4b.

The primary findings from our analysis of ice shelf morphology are that the basal fractures under NIS appear to play a role in both marine ice accumulation and in altering the strain regime of the ice shelf. Furthermore, the suture zone where these fractures from Reeves and Priestley Glaciers coalesce exhibits strong strain signals, which are associated with locations of transverse fracture formation.

### 4.3 Ice shelf melt rates

Our second aim is to determine where mass loss or gain may be occurring at this cold-cavity ice shelf with particular focus on the suture zone. Ocean glider data that we collected at the terminus of NIS (see Appendix B) demonstrate that Ice Shelf Water (identified by potential temperature $< -1.94$ ºC) is present, indicating that there is basal melt of the ice shelf. However, these data cannot be used to determine precisely where that NIS melt originated from. To assess whether this melt may be occurring in the suture zone, potentially due to channelized flow, we examine the change in ice shelf shape in the suture zone as it advects downstream. We conducted a fluxgate polygon analysis following Neckel et al (2012) using cross-sections from ground-based radar and ice surface elevation transects, one from Site 1 (closest to the grounding line) and one from Site 3 (closest to the ice shelf terminus). Site 2 is not included in this analysis because the radar cross-sections are not laterally extensive enough to cover the entire channel (see Appendix Fig. C1a). We use streamline analysis applied to the beginning and end of the Site 1 cross-section with the 2018 ITS_LIVE Landsat 8 derived annual velocity database (Gardner et al., 2020) to determine where the Site 1 and Site 3 cross-sections intersect (Appendix Fig. C1b) and we use these streamlines at a resolution of 50 m to complete the polygon (Appendix Fig. C1a). For ice thickness along the streamlines we use the hydrostatic ice thickness derived in Section 3.2. The ITS_LIVE dataset is then used to extract velocity vectors along all sides of the polygon.

The cumulative ice flux gain or loss ($\emptyset$) is calculated using:

$$\emptyset = \sum_{i=2}^{i=N-1} v(i)cos\alpha(i)H(i)dl , \qquad (2)$$

where $i$ is the position along the edge of the polygon, $N$ is the total number of positions in the fluxgate, $v$ is the ice flow speed at polygon position $i$, $\alpha$ is the angle between the velocity vector and the direction perpendicular to the transect, $H$ is the ice thickness, and $dl$ is the spacing between polygon positions. Error in the fluxgate calculation is determined by applying the radar thickness uncertainty discussed in Appendix A3. Volume gain or loss over the polygon is converted to mass loss assuming an ice density of 917 kg m$^{-3}$. Total volume loss within the polygon is $0.0523 \pm 0.0083$ km$^3$ a$^{-1}$, or a mass loss of $0.048 \pm 0.0076$ Gt/year. The discharge at Site 1 is $0.262 \pm 0.004$ km$^3$ a$^{-1}$ and at Site 3 is $0.210 \pm 0.005$ km$^3$ a$^{-1}$ demonstrating an average cross-sectional mass loss of 2364 m$^2$ a$^{-1}$ as the ice flows ~22 km between these two sites. It takes an average of 123 years for the NIS to flow over this distance using velocities extracted from ITS_LIVE.

To estimate the cause of the mass loss between Site 1 and 3 we examine ice shelf basal melt rates derived from Cryosat-2 satellite radar altimetry data provided as a grid with 500 m spatial resolution over the NIS (Appendix Fig. C1c; Adusumilli et al., 2020). The basal melt rates from Cryosat-2 indicate there are regions of freeze-on (shown by negative values) initiating

where the suture zone begins with a maximum estimated mass gain of ~2 ± 0.5 ma$^{-1}$ (Appendix Fig. C1c). At Site 1, the maximum mass gain is 1.3 m a$^{-1}$ ± 0.5 ma$^{-1}$. Within the suture zone, there is then a transition into a region of melt 4 km upstream of Site 2. Site 2 itself is close to the maximum melt rate in the suture zone of 0.6 m a$^{-1}$ ± 0.5 ma$^{-1}$. The record does not cover Site 3 but the closest region has higher melt rates than Site 2 (~1 ± 0.6 ma$^{-1}$), although this may be unreliable due to the 2016 calving event, which is within the window of altimetry data amalgamation.

It is possible that mass loss is due primarily to surface melt and sublimation, which was calculated by Bromwich and Kurtz (1984) to be 0.25 m a$^{-1}$ and by Bell et al. (2017) to be 0.5 m a$^{-1}$. To investigate this and where basal melt may be occurring at the ice-ocean interface, we extract streamlines between Site 1 and Site 3 and apply Lagrangian methods as described in Das et al (2020) to estimate strain thinning, basal melt and surface melt along those streamlines:

$$\Delta H = \int_{t1}^{t2} -w_b - w_s - H\nabla.\boldsymbol{V} dt, \qquad\qquad (3)$$

where $w_b$ is basal melt, $w_s$ is surface melt, $\nabla.V$ is the ice velocity divergence from horizontal flow, and $t_1$ and $t_2$ are the times that ice passes through Site 1 and Site 3. Cumulative basal melt along the streamlines, $w_b$, is calculated from the Cryosat-2 basal melt dataset (Adusumilli et al., 2020; Fig 6.). However, the basal melt map only has data up to 2 km away from the Site 3 transect (Appendix Fig. C1c) and so cumulative basal melt will be slightly underestimated using this technique. Similarly, we calculate cumulative surface mass loss, $w_s$, between the two sites using a uniform surface ablation rate of 0.25 m a$^{-1}$ and 0.5 m a$^{-1}$. Because the region we are examining is composed of exposed blue ice we do not take accumulation into account. Velocity divergence for strain thickness change is calculated from ITS_LIVE 2018 ice surface velocities and flow vectors (Gardner et al., 2020; Fig. 6). This analysis assumes steady state surface melt, basal melt, and ice velocity. The average time for ice to travel between Site 1 and Site 3 is 123 years, which means that some alteration of these rates may have taken place producing errors in our analysis, although with a cold cavity this is likely less of an issue than in regions of the West Antarctic for example (Das et al, 2020). We compare the values of cumulative basal and surface ablation and strain thinning with the IPR-derived thickness change between the sites (Fig. 6). Because of the presence of basal features that are not fully aligned between the sites, we first smooth the thickness at each of Site 1 and Site 3 using a locally weighted least squares regression filter before calculating the thickness change.

The average thickness change between Site 1 and Site 3 is 80 m (Fig. 6). The cumulative basal melting analysis suggests that, because of initial freezing upstream in the middle of the suture zone (see Appendix Fig. C1c), the central region of the suture zone has net zero change in basal melt. However, the steeper margins of the suture zone have sufficient basal melt to account for ~20 m of thinning at the ice base between Site 1 and Site 3. If we combine this thinning from basal melt with a uniform surface ablation rate of 0.25 m a$^{-1}$ it produces ~50 m less thinning compared to the measured thickness change. However, when the thinning from basal melt and strain is combined with a thinning due to a uniform surface ablation of 0.5 m a$^{-1}$, there is a good correspondence between the estimated cumulative thinning rate and the measured thinning, particularly at the northern margin of the suture zone, suggesting this higher surface ablation rate is the more accurate estimate when averaged over time. Thickness change due to strain thinning reaches a maximum of 13 m between Site 1 and Site 3.

The primary findings from our investigations into ice shelf melt is that basal melt is likely occurring within the suture zone, with that melt focused on the flanks of the suture zone rather than its center.

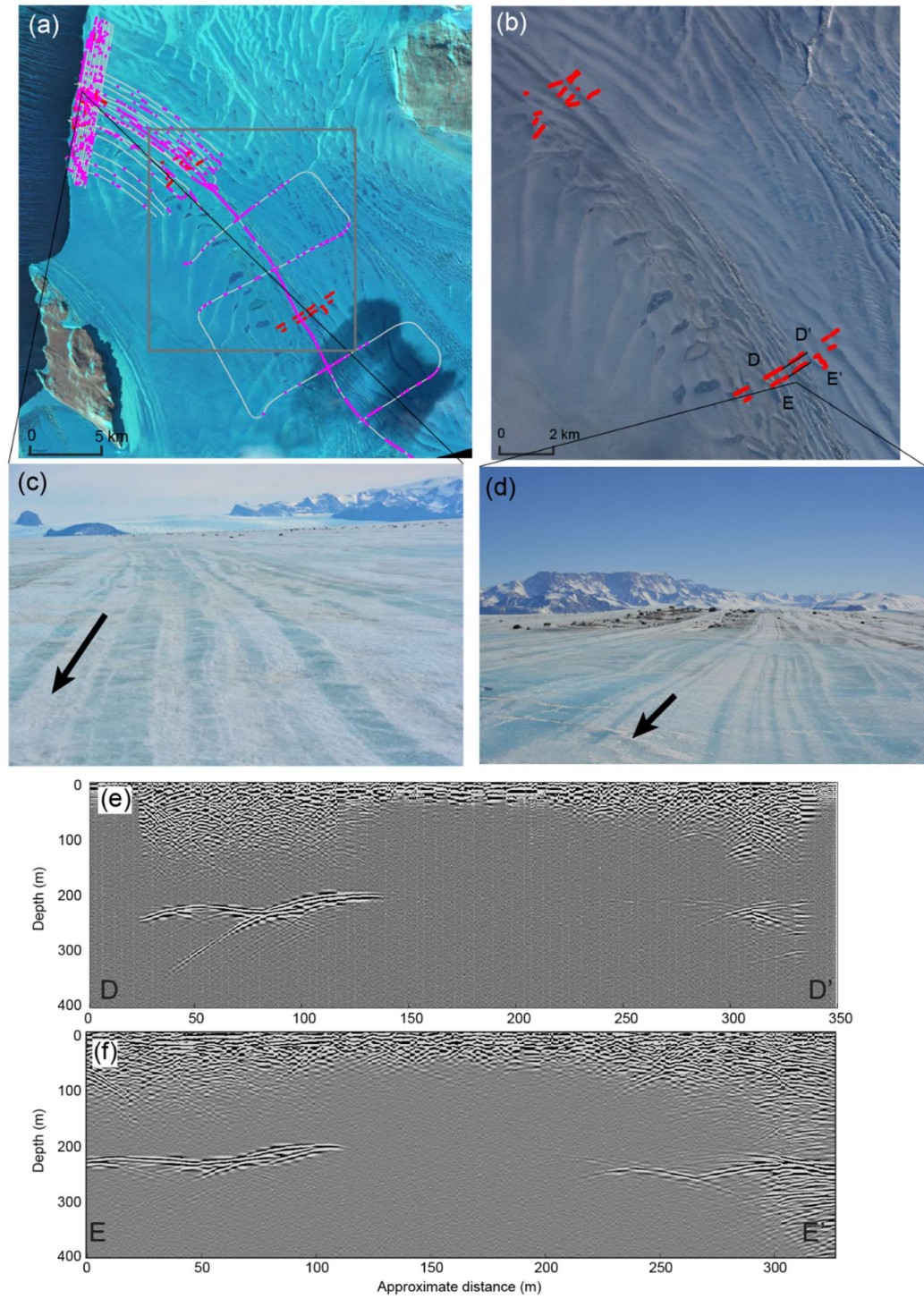

**Figure 5:** (a) Sentinel image of NIS from October 5th 2016. Pink points are regions with radar signal dropout from the airborne radar and red points show signal dropout from the ground-based radar. The grey lines show the airborne radar and the grey box shows the extent of (b). (b) Larger-scale image showing ground-based radar dropouts in red. The black lines show the extent of the radar transects plotted in (e) and (f). Background image is Landsat 8 March 22, 2017. (c) and (d) Photos of surface striping on NIS showing filled crevasses associated with the marine ice drop-out regions taken on November 11 and November 18, 2016, respectively. (e) and (f) Radargrams with signal dropout. Transect locations are shown in (b).

# 5 Discussion

## 5.1 Complex ice morphology

The ice draft elevation DEM we present computed using REMA surface elevation data reveals complex geometry of the suture zone. The thin ice of the suture zone is driven by a limited supply from the glaciers that only converge once they are already floating. The margins of the suture zone include regular basal fractures that have different characteristics depending on which side of the suture zone they form on. These appear to be driven by fracturing on the margins of the ice shelf before it converges in the suture zone. On the Reeves Glacier side, this fracturing can be seen in the 'flame-like' features near the ice fall (see green circle in Appendix Fig. C1a). These appear to be longitudinal crevasses that occur at the grounding line as the ice begins floating at the base of the ice fall and are re-oriented in the down-flow direction. The features are not continuous and perhaps represent periods of stick-slip motion on the ice fall causing periods of enhanced fracturing and flow followed by build-up of ice. These fractures can be traced downstream to the terminus in both REMA data (Fig. 2a) and in Landsat satellite imagery (Fig. 1a).

On the northern side, the equivalent features swing away from the suture zone by almost 90° as they approach the terminus. Again, tracing these upstream, they appear to originate from fractures on the southern margin of the Priestley Glacier. The fractures take on the sweeping shape likely due to the faster speed of Reeves Glacier (~200-250 m a$^{-1}$) compared to Priestley (~150-200 m a$^{-1}$), a behaviour that becomes increasingly enhanced towards the terminus (Fig. 2a). The difference in flow speed of the two portions of NIS is likely due to the pinning point of Inexpressible Island to the north (Fig. 4a). Examining the REMA elevation data for the upstream area of the NIS (as there are no radar data in that location), the Priestley basal fractures first appear with lengths of 2 km. The floating Priestley Glacier ice thins and stretches as it advects downstream towards the terminus, causing these basal fractures to expand laterally northwards, with lengths up to 7 km. The basal fractures on the Priestley side result in hydrostatic depressions on the surface that are easily identifiable through water ponding on the surface during the brief (<25 days per year; Bell et al., 2017) melt season of this ice shelf (Fig. 5a).

The basal fractures are clearly observable in the radar data (Fig. 3) but are less clear if using only hydrostatic analysis. We found that, at Sites 1 and 2, closer to the grounding line, the spatial pattern of hydrostatic thickness does not accurately reflect the variable topography in this region. For example, at Site 1 on the northern side of the suture zone, the basal fractures represented in the radar data are around 1600 m wide and 50 m high, which is similar to the fractures measured from hydrostatic balance (which are 1655 m wide and 53 m high). On the southern side of the suture zone, the radar thickness reveals fractures that are 448 and 522 m wide with heights of 64 m and at least 82 m. Yet in the thickness data derived from hydrostatic calculations, these fractures are difficult to discern and are limited in height (only up to 10 m).

Ice shelf draft estimations that are extrapolated using hydrostatic balance calculations do not take bridging stress (i.e., flexural stress between areas of relatively thick ice) or pinning points into account and therefore may not fully represent the ice thickness and basal ice morphology (Drews et al., 2015; Gladish et al., 2012; Mankoff et al., 2012; Vaughan et al., 2012). Our data suggest that these bridging stresses are likely to obscure basal morphology only when the basal fracture features are narrow and tall; in this case narrower than ~500 m and taller than 60 m. This effect has been noted at other ice shelves (e.g., Raymond and Gudmundsson, 2005; Nicholls et al., 2016) although it was assumed that the features had to be significantly smaller than the ice thickness to remove the hydrostatic signature from the ice surface. Yet at Site 1 where we have the most significant difference between measured and estimated morphology, the ice thickness ranges between 200-330 m, ~ half of the width of the basal fracture features. Our results therefore suggest that factors such as the height of the features must also therefore play a key role in limiting hydrostatic surface signatures and should be taken into account when utilising satellite-derived ice thicknesses for analysis.

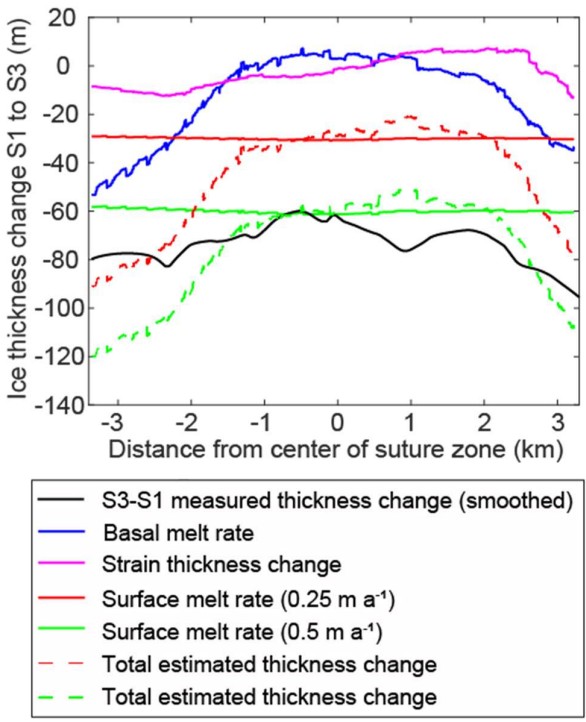

335

**Figure 6:** Cumulative ice thickness change between Site 1 and Site 3 calculated from IPR thickness change (black), Cryosat-2 basal melt data (blue), two surface melt rates (red and green), and strain thinning (pink). Total thinning (Cryosat-2 basal melt plus surface melt plus strain thinning) is shown by the dashed red and green lines for surface ablation rates of 0.25 and 0.5 m a$^{-1}$, respectively.

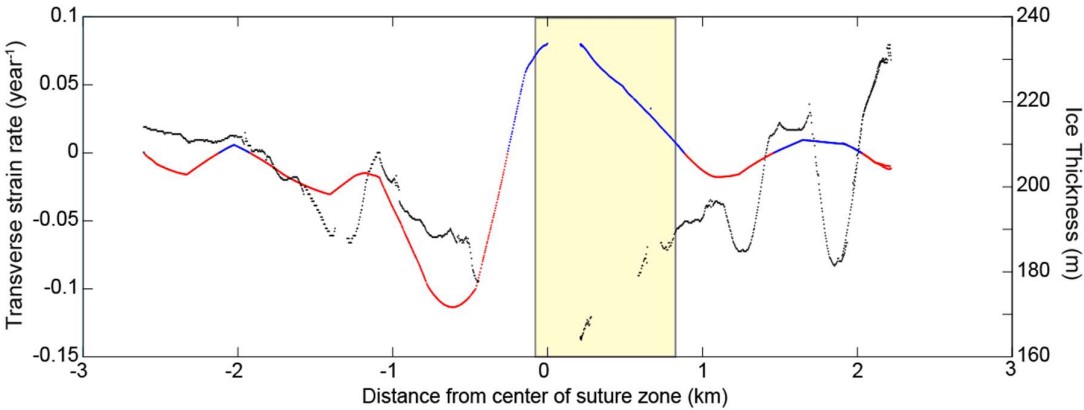

340

**Figure 7:** Site 2 transect ice thickness (black) with transverse strain rate (red: compression; blue: extension) interpolated along the transect. The location of the new surface fracture formed in 2016 is blocked in yellow and intersected this IPR transect. Note the radar drop out in the region of fracture formation.

345

**5.3 Ice shelf fractures and strain rates**

There is a clear relationship between the northern flank basal fractures and the ice surface transverse strain rates (Fig. 4b, d) where extension occurs over the keels and compression over the apexes. This is likely driven by the hydrostatic balance forcing surficial bending to conform to the basal variation in ice draft, with deeper keels and shallower apexes driving the formation of hills and troughs on the ice surface, respectively. As described in Vaughan et al. (2012), who applied a thin bending beam and finite element model to examine the stresses caused by variable ice shelf draft, there will be compression in the thinner ice regions and extension over the thicker ice regions.

This pattern of variable strain is stronger on the northern side potentially due to the orientation of the basal fractures relative to the flow direction of the ice (~ up to 90º) in comparison with the southern basal fractures, which are parallel to ice flow. Drews et al. (2015) examined the Roi Baudouin Ice Shelf, and suggested that surface velocities and related strain rates from satellite observation may be sufficient to characterize basal channel morphology. This seems to also be applicable to the NIS basal features and may be a useful tool when examining areas with significant variability in basal draft that might not be visible in the hydrostatic balance calculations due to bridging stresses.

The stripe of compressive strain along the suture zone center adjacent to a stripe of extensional strain on the southern side is an interesting characteristic corresponding with the thinnest region of ice. The compressive strain on the northern side is potentially due to the slower speed of Priestley Glacier coupled with pinning of this ice between Inexpressible Island and the Reeves Glacier ice. On the southern side, the extensional transverse strain is likely due to the faster speed of the less-constrained Reeves ice compared to Priestley's. Beyond the thinnest region of the suture zone there is more extension on the southern side of the ice shelf compared to the northern side (Fig. 4d). Alternatively, this strain pattern may be indicative of secondary ice flow into the thinnest region of the suture zone, particularly if there is basal melt occurring in this region (Wearing et al, 2021). This spatial difference in strain may contribute to the more significant lateral stretching and dampening in draft of the southern basal fractures between Site 1 and Site 3 as shown in Fig. 1c.

The more significant extensional strain on the southern side explains why the 2016 fracture originates at the center of the suture zone and expands to the south, but does not extend into the compressive region to the north (Fig. 4b). In 2016, the extent of this fracture reached as far as a compressive region associated with a basal fracture on the northern side where the ice becomes thicker (Fig. 7). It should be noted that only the transverse strain rate map provides information on the compression and extension in the keels and apexes of the basal fractures, along with the extensional and compressional striping along the suture zone. If only longitudinal strain had been calculated, these features would have been missed. The transverse strain rate therefore appears to be the primary control on the patterned variability in extension and compression on the NIS and, although larger-scale longitudinal extension played a primary role as to when the 2016 fracture formed (Dow et al., 2018), the transverse strain and ice thickness limited the lateral extent of the fracture.

**5.4 Marine ice**

Evidence of fracture propagation rates across other Antarctic ice shelves, such as Larsen C, suggests that the presence of marine ice, particularly in suture zones, hinders fracture expansion due to its higher fracture toughness (e.g., Hulbe et al., 2010; McGrath et al., 2014; Borstad et al., 2017). Echo-free zones in the radar records in the thinner regions of NIS (Fig. 5d) suggest the presence of accreted marine ice and/or unconsolidated frazil ice; for the purposes of this discussion we will refer to both types as "marine ice" (Holland et al., 2009). Marine ice is known to inhibit radar returns from the ice-ocean interface as it absorbs the radar signal very effectively (Moore et al., 1994). With IPR data alone we cannot assess whether these losses of signal represent full thickness rifts with frozen seawater or as a result of frazil ice accumulation within regions of reduced ice

draft. However, with the REMA hydrostatic DEM and our *in situ* survey of the ice shelf, it appears that the marine ice in the center of the suture zone originates from full-thickness fractures that refreeze with intruded ocean water at the base of Reeves Ice Fall. This is consistent with ice core analysis of marine ice frozen into a rift 7.5 km downstream of the Reeves grounding line by Khazendar et al. (2001).


Full rifts containing marine ice should therefore inhibit fracture propagation regardless of whether the fracture initiates at the surface or the base of the ice. However, it was across one of the regions of 'striped' ice with multiple full-thickness marine ice bands, and radar data echo-free zones, that the 2016 fracture formed (Fig. 5a, Fig. 7). It is possible that the bands of marine ice are too thin to impede propagation of the fracture, although this would be interesting to test this hypothesis with fracture

mechanics modeling. The fracture expanded southwards until it reached an area of meteoric ice, where, as mentioned above, transverse compressive strains dominate due to the basal morphology, preventing further expansion of the fracture. Borstad et al. (2017) suggested that the presence of a basal crevasse perpendicular to the fracture may hinder propagation of the fracture tip, and this may be another reason for the limited southern extension of the 2016 NIS fracture.

Outside of the central suture zone, regions of radar signal loss are associated with basal fractures and therefore likely due to the accumulation of marine or frazil ice within the fractures. Comparing the regions of marine/frazil ice accumulation in Fig. 5a with the basal melt map in AppendixFig. C1c, the transition between larger-scale basal freezing, located closer to the grounding line, to basal melting does not seem to impact the extent or number of echo-free zones associated with basal fractures. Marine or frazil ice is therefore present in at least the apexes of some of the basal fractures even towards the ice shelf

terminus where melt rates increase to >1 m a$^{-1}$. Given our analysis suggesting that most basal melt occurs on the steeper margins of the suture zone, this suggests that the basal fractures penetrated sufficiently far into the ice and filled with frozen ocean water (e.g., McGrath et al., 2014) such that some of the marine/frazil ice remains within these fractures in spite of the melting on slopes at the base of the NIS. Alternatively, ocean melt may be focussed on the deeper keels with frazil ice formation possible at the apexes of the basal fractures. The dampening of the amplitude of the basal features as they advect as shown in

Figure 3, suggests that such differential melting and freezing may be occurring (see e.g. Khazendar and Jenkins (2003), Jordan et al. (2014), and McGrath et al. (2014)).

A full characterization of marine ice on the NIS would require a geophysical investigation with active seismic lines or *in situ* sampling and could elucidate useful information about cavity circulation under this ice shelf and establish whether the presence

of the Terra Nova Bay polynya has an impact on basal ice accumulation. It would also allow ground truthing of the satellite melt map product.

**5.5 Ice shelf melt**

In addition to examining regions of the NIS where mass may be accumulating in the form of marine/frazil ice, we are interested in where mass is lost. We also assess whether we are able to classify the thin ice suture zone of the NIS at the convergence of

the Priestley and Reeves glaciers as a channel or whether it is a purely ice-dynamic feature with minimal oceanic enhanced melting. Even in a cold cavity, some enhanced melt would be expected with channelized flow and, as increasing numbers of ice shelf channels are being identified (Alley et al., 2016), including many in suture zones, it is important to establish the role they play in alteration of the ice shelf. The glider data demonstrate that ISW is being produced from this cold cavity although it cannot be used to determine whether it is sourced from the suture zone or a wider region of the ice shelf. Alley et al. (2019),

who examined shear margins around the Antarctic, used the presence of polynyas to detect where the thinner ice of a suture zone has allowed enhanced oceanic flow and therefore can be classified as a channel. The NIS is associated with the substantial Terra Nova Bay polynya, a major sea ice producer in the region (Ciappa et al., 2012), which stretches the full width of the

NIS. However, this is driven primarily by katabatic winds flowing over the ice shelf (Kurtz and Bromwich, 1983) and therefore we cannot use it for channelized flow classification. At the NIS, the difference in ice thickness between the suture area (~200 m thick) and the body of the ice shelf where the floating ice initially converges (~550 m on the Priestley side, ~300 m on the Reeves side) could drive ocean current acceleration and enhanced melt (Little et al., 2009). Furthermore, the change in shape of the suture zone as it advects downstream does suggest some alteration from ocean-driven melting. The thin-ice region has a longer, lower-gradient edge on the Priestley (northern) side and a steeper but shorter wall on the Reeves (southern) side, which may be due to Coriolis-forced enhanced melt towards the northern side of the channel. This is further suggested by the flowlines in Appendix Fig. C1b that show the northern side of the suture zone deviates further from the background flow compared to the southern side. Coriolis-forced acceleration of currents along the flanks of channels causing enhanced melt has been identified in other basal channels around the Antarctic (Alley et al., 2016; Dutrieux et al., 2013; Gladish et al., 2012; Mankoff et al., 2012). The pinning of the Priestley ice between Inexpressible Island and the Reeves ice inhibits enhanced lateral stretching in this region, whereas the Reeves ice flows beyond lateral pinning points sooner and is therefore able to expand to the south. If driven by ice dynamics alone, it would be expected that the Reeves side of the suture zone would be the less steep flank as opposed to the Priestley. As the opposite occurs it suggests that the shape of the suture zone is altered by Coriolis-driven enhanced melt, with accelerated flow in the regions of highest basal draft driven towards the northern side of the suture zone.

We have also presented multiple analyses to assess whether and where ice shelf melt is occurring including a flux gate calculation; IPR thickness change estimates between transects; and cumulative melt rates on the surface and base between our IPR sites, derived from ice surface velocity data combined with satellite-derived basal melt rates (Adusumilli et al., 2020) and surface ablation estimates (Bromwich and Kurtz, 1984; Bell et al., 2017; Fig. 6). Each of these techniques has limitations. The flux gate and cumulative melt analyses, while taking account of horizontal strain, do not take into account potential changing ocean or surface conditions over the >100 years for ice to advect the 22 km between Site 1 and Site 3. In addition, the satellite-derived basal melt rates signals (Adusumilli et al., 2020) are close to the noise floor (uncertainty) of the dataset for the NIS. Furthermore, the latter have been shown to differ from pRES radar measurements of basal melt at Filchner Ice Shelf due to less accurate velocity products for the satellite-based calculations (Zeising et al, 2022), and the same may be the case at NIS. However, our results from both *in situ* and satellite-derived data suggest that there is enhanced basal melt on the steeper slopes at the margin of the suture zone, which gives us confidence in our results. The central region of the suture zone, in contrast, appears to be dominated upstream by basal freezing, which is offset by melting closer to the terminus. Change in thickness in the thinnest central region is likely therefore primarily as a result of surface ablation (Fig. 6).

The maximum melt rate In the suture zone from the Cryosat-2 basal melt map is ~0.6 ± 0.5 m $a^{-1}$, except for at the ice shelf terminus where rates increase to ~1 ± 0.6 m $a^{-1}$(Adusumilli et al., 2020). Similarly, generally low melt rates (0.08-0.09 m $a^{-1}$) are predicted for the base of the nearby, cold cavity Ross Ice Shelf, with melt increasing to 2 m $a^{-1}$ near its terminus (Stevens et al., 2020; Das et al, 2020). However, these rates are low compared to other ice shelves such as Getz Ice Shelf which has an area averaged melt rate of 4.15 m $a^{-1}$ (Wei et al., 2020), and ice shelves that come into contact with relatively warm circumpolar deep water have even higher basal melt rates, such as Pine Island Glacier ice shelf (30 m $a^{-1}$; Dutrieux et al., 2013). Oceanographic data in Terra Nova Bay collected by Manzella et al. (1999) and by the ocean glider indicated relatively cold ocean water conditions, explaining why basal melt rates within the suture zone are likely to be small in comparison to the Amundsen Sea sector. The basal ice loss rates over the steeper slopes at the edge of the suture zone (Fig. 4b) do, however, suggest a basal melt component and that the suture zone is acting to channel water.

The transverse fracture discovered in 2016 occurs at the transition into the highest melt region of the suture zone (Appendix Fig. C1c). The formation of this fracture was argued to be due to an alteration of the strain regime from transverse to

longitudinal extension between 2014 and 2015, linked closely to the expansion and the calving of a fracture located much
closer to the ice shelf terminus (Dow et al., 2018). However, as this strain transition occurred in the same region as the
maximum melt, this region may be associated with a reduction in buttressing on the ice shelf cavity walls due to thinning ice,
compared to further upstream. This suggests that with ocean warming and related melt, which will also alter the draft
morphology and therefore the shelf strain rates, fracturing may occur further upstream and may laterally propagate more
rapidly, allowing more frequent calving events. To assess this would, however, require an ocean model driven with the complex
morphology that we have identified in order to determine the relative roles of supercooling and melt in the ocean cavity and
how this might change over time (Goldberg et al., 2019).

**6 Conclusions**

The Nansen Ice Shelf is a small, cold-cavity ice shelf, yet includes a variety of complex features due to the combination of ice
dynamics, the shape of the embayment and interaction with the ocean. The primary causes of the complex morphology at
Nansen Ice Shelf are the dynamics of upstream ice with basal fractures formed at the grounding line of Reeves Glacier and the
floating margin of Priestley Glacier ice before they join together in a suture zone. Comparison between ground-based radar
thickness data and ice surface GNSS elevation hydrostatic calculations suggests that bridging stresses can significantly dampen
the vertical surface expression of the basal morphology, in this case particularly for basal features narrower than ~500 m and
taller than 60 m. This is despite an ice thickness of between 200-330 m in this region suggesting that even features larger than
the ice thickness may not be observable in satellite data.

Changes in the basal fractures as they advect downstream is evident in both the ice shelf draft and the strain rate maps, and
demonstrates the competing elements of horizontal variability in transverse strain and differential ice velocities on either side
of the suture zone of these two glaciers. Application of only longitudinal strain rates would miss this complex relationship
between the basal fractures and the ice strain. Although longitudinal strain can provide information about controls initiating a
transverse surface fracture that was identified in the field in 2016, the more complex system demonstrated by the transverse
strain rates, which appears to limit the lateral extent of the fracture, suggests that for analyses of ice shelf stability, multi-
directional strain rates must be taken into account.


The basal fracturing leads to accumulation of marine ice in the full-thickness rifts associated with Reeves Glacier and
marine/frazil ice in the tips of basal fractures associated with Priestley Glacier, as demonstrated by radar echo-free zones and
stripes of blue ice on the ice shelf surface. The 2016 transverse surface fracture formed in the center of the suture zone in a
region of full-thickness rifts now filled with marine ice, which has previously been suggested to limit fracture propagation due
to higher fracture toughness. Here, the limited width of the re-frozen rifts (often less than 5 m) may restrict the ability of the
marine ice to hinder fracture propagation.

Our analysis of changes in ice morphology, flux gate volume, oceanographic data indicating the presence of ISW, and satellite-
derived ice shelf basal melt suggest that oceanographic melt is occurring within the suture zone. Basal melt appears greatest
on the steep margins of the suture zone whereas surface ablation seems to dominate mass change in the thinnest central region.
*In situ* sampling is required to determine the location and rates of melt, for example whether it is focussed only on the keels
between basal fractures. The Nansen Ice Shelf basal melt rates are small compared to warm cavities in the Amundsen Sea
region. However, it may have played a role in causing fracturing in a higher melt region within the suture zone of the Nansen
Ice Shelf. The combination of ice morphology, channelized melt, and strain patterns within the ice shelf all appear to play a
role in the formation of fractures in this ice shelf, which result in large-scale, periodic calving events (Dow et al., 2018). These

findings warrant investigation of other ice shelves with transverse fractures to establish causal mechanism behind fracturing at those locations and whether they are likely to lead to large-scale calving events.

## Appendix A: Ice Penetrating Radar methods

### A.1 Ground-based radar

An IceRadar (Blue Systems Integration Ltd.) equipped with 10 MHz resistively-loaded dipole antennae separated by 15 m was used to collect the on-foot radar lines (Mingo and Flowers 2010). The IPR transmitter pulsed 512 times per second and the average of a stack of 50-256 traces were recorded by the receiver at an interval of 1-3 seconds.

Precise location data were simultaneously collected at 1 Hz using a Topcon Hiper V L1/L2 Global Navigation Satellite System
(GNSS) receiver placed next to the IPR receiver. GNSS data were subsequently processed using Natural Resources Canada's (NRCan) precise point positioning (PPP) service and corrected to the GL04C geoid datum (e.g. Griggs and Bamber, 2011), after subtracting the height of the GNSS antenna center above the ice surface (Greene, 2021). Radar Tools (Wilson and Mueller, 2023) was used pick the air wave and reflected wave from each trace. Ice thickness ($H$) was calculated as:

$$H = \sqrt{\frac{v^2(t+d/c)^2}{4} - \left(\frac{d}{2}\right)^2} , \qquad\qquad\qquad\qquad (A1)$$

where $d$ is the antenna separation distance (15 m), $t$ is the two-way travel time (s) from the air wave to the reflected wave, $c$ is the speed of the radar wave in air ($3 \times 10^8$ m s$^{-1}$) and $v$ is the speed of the radar wave in ice ($1.68 \times 10^8$ m s$^{-1}$; Fujita et al.., 2000).

### A.2 Airborne radar

A radar system was installed on an Aérospatiale AS-350 helicopter in a fixed boom, and was complemented by a laser
altimeter, camera, Global Positioning System (GPS), and an internal navigation system (INS) for precise positioning. Two of the three fixed booms attached to the bottom of the helicopter contained separate antennae, installed in a cross-track polarized dipole arrangement, that both transmitted and received independently, with 60 MHz center frequencies, 15 MHz chirp bandwidths, and 1 µs pulse width.

### A.3 Error analysis

Where survey lines crossed, ice thickness and surface elevation values were compared at points of closest approach (within a maximum distance of 20 m) to evaluate repeatability and bias between datasets. Ground-based IPR ice thickness 'crossover' points (n= 13) agreed to within 2.2 m of each other on average with a basic bootstrapped 95% confidence interval (CI) of 0.5 to 3.5 m. Crossovers from airborne IPR thickness focused on the thin ice area and ice shelf terminus (n = 158) agreed to within 3.3 m (CI: 2.7 to 3.9 m), whereas the general NIS survey thickness crossovers (n= 99) agreed within 7.7 m (CI: 5.6 to 9.6 m).
Ice thickness in the ground-based IPR survey was 2.1 m thinner on average (n=113) than the airborne survey of the thin ice area with a mean absolute difference of 3.5 m (CI: 2.8 to 4.2 m).

Average surface elevation crossover errors in the ground-based (n=23) and airborne (n=557) IPR surveys were 0.71 m (CI: 0.21 to 1.11 m) and 1.02 m (CI: 0.80 to 1.19 m), respectively. Surface elevation from ground-based surveying was 0.08 m higher on average (CI: -0.10 to 0.23 m) than the airborne elevation (n = 189) with a mean absolute difference of 0.69 m (CI:
0.54 to 0.81 m).

The relative uncertainty in ice shelf draft can be derived from these two errors added in quadrature. For example, NIS basal draft estimations from the ground-based survey have an uncertainty of ±3.7 m, 95% of the time, whereas the airborne survey focused on the thin region and terminus has an uncertainty of ±4.1 m; the entire airborne survey has an uncertainty of ±9.7 m.

**Appendix B**

**B.1 Ocean glider data**

Between December 31, 2018 and January 10, 2019, a 1000 m depth rated G2 Slocum autonomous underwater glider was deployed along the NIS to collect ocenographic observations. Approximately 200 full-depth profiles were completed across more than 100 km of transects along the ice shelf terminus, including ~10 km beneath the NIS, near the suture zone (Appendix
Fig. C1 a,e). Hydrographic measurements were recorded on the glider using a SeaBird SBE-41 CTD (conductivity-temperature-depth) sensor to infer the presence of seawater modified by ice shelf melt. Although other water column parameters were collected (e.g. dissolved oxygen, chlorophyll, turbidity, etc.), traces of meltwater are identified by temperature and salinity alone.

Despite the presence of warm Antarctic Surface Water (AASW; potential temperature > -1.7 °C) at the surface of Terra Nova Bay during austral summertime (Rusciano et al., 2013; Yoon et al., 2020; Friedrichs et al., 2022), NIS basal melt is presumed to be driven largely by High Salinity Shelf Water (HSSW; Budillon and Spezie, 2000; Rusciano et al., 2013), a dense water mass that fills much of the water column in the western Ross Sea (Jendersie et al., 2018). This HSSW is one of the coldest water masses in the Southern Ocean (Grumbine, 1991), existing at temperatures close to the surface freezing point (~-1.9 °C,
depending on the salinity; Rusciano et al., 2013; Yoon et al., 2020). Its interaction with glacial ice then produces even colder Ice Shelf Water (ISW), uniquely identifiable in the region by temperatures close to the subsurface freezing point (<-1.9 °C, depending on pressure). We therefore take observations below the -1.94 °C isotherm (equivalent to the freezing point at a salinity of 34.7 and a depth of 50 m) as evidence of the presence ice shelf meltwater (Appendix Fig. C1d).

This ISW was distributed along the NIS terminus at intermediate depths, beneath the warm Antarctic Surface Water and above the deep High Salinity Shelf Water. There are three regions where this water appears. The southernmost, adjacent to Drygalski Ice Tongue, had ISW at depths between 58-490 m. The second region of ISW was towards the middle of the ice shelf with depths between 113-397 m. Our IPR data show that the draft of the NIS adjacent to this cold water mass ranges from 150-190 m in depth. The third region of ISW was close to Inexpressible Island at the northern margin of NIS and had depths between
128-502 m; here the NIS draft ranges between ~160 and ~230 m depth (as determined from REMA hydrostatic calculations). The second region of ISW at shallower drafts therefore may be linked to basal melt occurring in the suture zone as also suggested by our analysis of our IPR data and satellite-derived basal melt rates. However, the presence of a sub-mesoscale eddy at the terminus of Nansen Ice Shelf (Friedrichs et al, 2022) means that we cannot use directional data from the glider to determine whether the suture zone is a greater source of ISW than other regions of the sub-shelf cavity. Instead, the data merely
indicates that ice shelf melt is occurring at this cold cavity ice shelf.


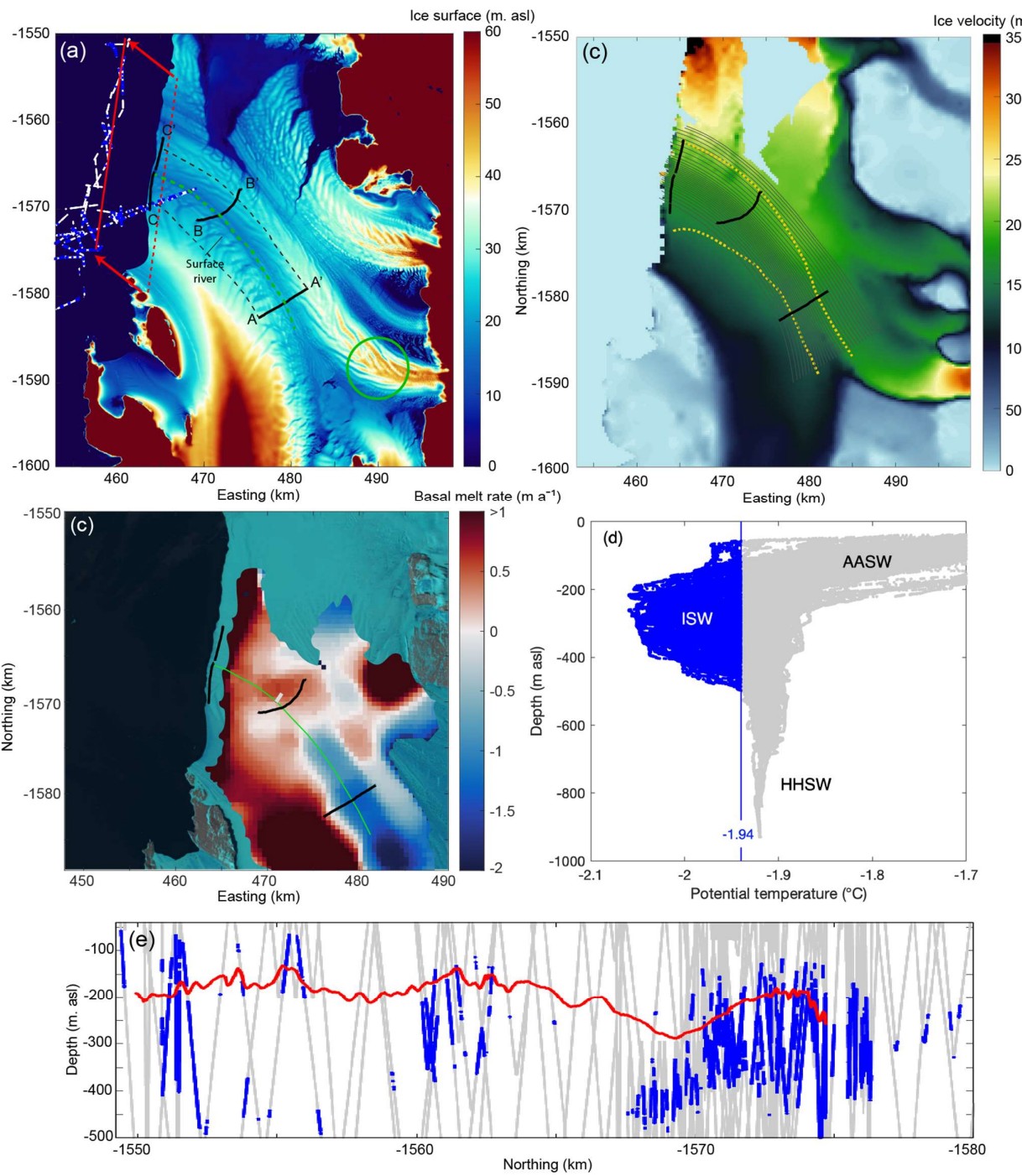

**Figure C1:** (a) REMA ice surface elevation with the three transects in Fig. 3 plotted in black. The dashed black lines show the edge of the polygon calculated for flux gate analysis. The green circle highlights an area referenced in the text in section 5.1. The DEM beyond the ice shelf terminus is set to 0 m due to the presence of data in the mosaic from prior to the 2016 calving event. Blue points are the locations where Ice Shelf Water was detected by the ocean glider with white points indicating all other locations where glider data was collected between 40 and 500 m depth. The dashed red line shows the location of REMA
hydrostatic ice draft plotted in (c), adjusted to the downstream location of the glider data (solid red line). (b) Annual ice surface velocity (2018) from ITS_LIVE (Gardner et al., 2020) with the suture zone marked by the yellow dashed lines and flow lines

marked in black. (c) NIS basal melt rates averaged between 2010-2018 from Cryosat-2 data; data from Adusumilli et al. (2020). The 2016 surface fracture is plotted in grey, and the site transects shown in Fig. 3 are plotted in black. The center of the suture zone is shown in green. The background is a Landsat-8 image acquired on January 2, 2019. (d) Temperature profiles from the ocean glider showing all data from (a). The locations of Ice Shelf Water are shown in blue, whereas all other observations are in grey.(e) Glider observations at intermediate depths (40-500 m). The locations of Ice Shelf Water are shown in blue, whereas all other observations are shown in grey. The solid red line indicates ice shelf draft from REMA hydrostatic inversions, projected over the glider transect.

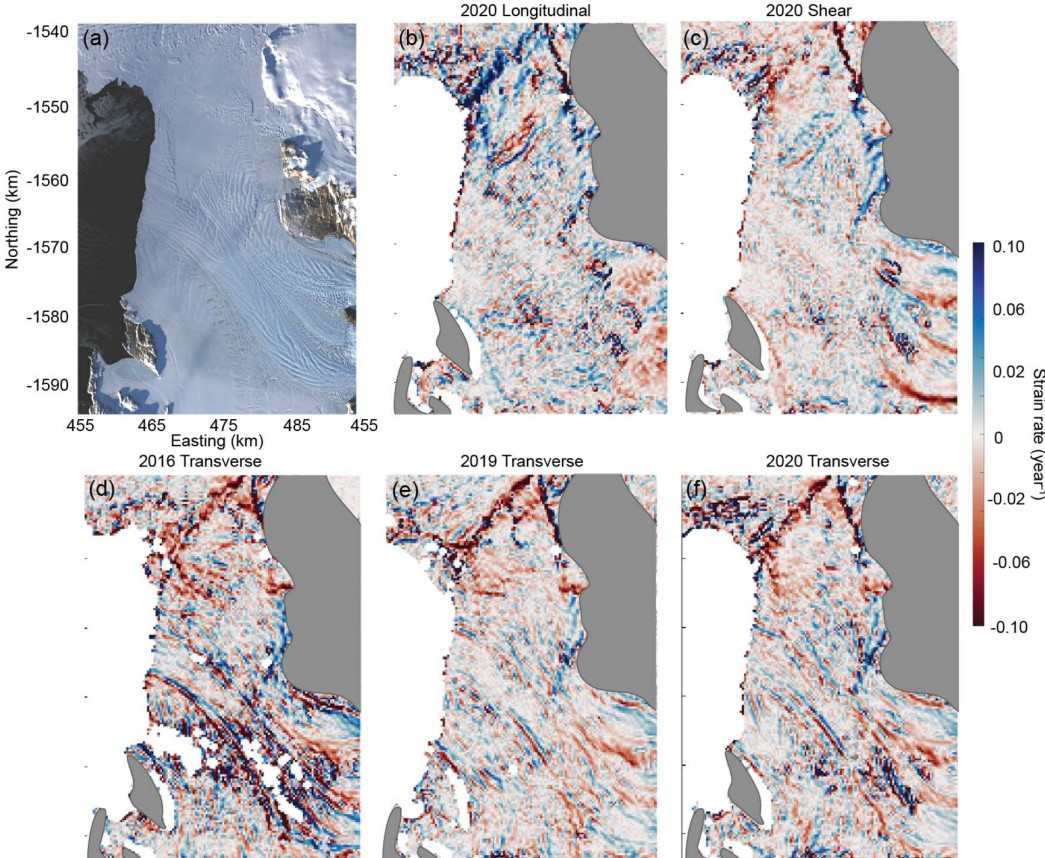

**Figure C2**: (a) The Nansen Ice Shelf (NIS) from Landsat-8 acquired March 22, 2017. (b) Longitudinal strain rate from GoLIVE velocity data acquired between October 10 and November 11 2020. (c)   Shear strain rate from the same data as (b). (d) Transverse strain rate from GoLIVE velocity data acquired between September 29 and October 31 2016. (e) Transverse strain rate from GoLIVE velocity data acquired between October 8 and November 9 2019. (f) Transverse strain rate from the same data as (b).

## Data availability

Ice penetrating radar data and locations of Ice Shelf Water from the ocean glider can be found here: 10.5281/zenodo.4891281. Landsat satellite imagery is freely available from USGS (https://earthexplorer.usgs.gov/), with REMA, ITS_LIVE, and GoLIVE data freely available from NSIDC (https://nsidc.org). Ice shelf basal melt data is available through UCSD Library Digital Collections.

## Author contribution

CFD and DM developed the project, analysed the radar and satellite data, and undertook the ground-based radar fieldwork. PW processed the radar data and provided initial analyses. DF, ALF, and JM collected, processed, and analysed the ocean glider data. JG and DDB contributed the airborne radar data and assisted with processing and interpretation. CKL and WSL provided field support and contributed to project design. CFD wrote the manuscript with input from all authors.

## Competing interests

The authors declare that they have no conflict of interest.

## Acknowledgements

CD was supported by the Natural Sciences and Engineering Research Council of Canada (NSERC; RGPIN-03761-2017) and the Canada Research Chairs Program (950-231237). DM was supported by NSERC (RGPIN-06244-2016), the Canada Foundation for Innovation, the Ontario Research Fund (314190) and a Canadian Arctic-Antarctic exchange Polar Continental Shelf Program (647-17). JSG was supported by NSF OPP-2114454.. Airborne radar and ocean glider data were collected by the support of the Korea Institute of Marine Science & Technology Promotion(KIMST) funded by the Ministry of Oceans and Fisheries(RS-2023-00256677; PM23020), which also support W S Lee and C-K Lee. DDB thanks the G. Unger Vetlesen Foundation for airborne radar support. The authors also thank Jin Hong Kim, Laura Lindzey, Enrica Quartini, and Dillon Buhl for assistance in the field and Laurent Mingo for help with radar interpretation.. We thank Ala Khazendar and three anonymous reviewers, along with the Associate Editor, Reinhard Drews, for their helpful comments on this manuscript.

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
