# Peer review of "The complex basal morphology and ice dynamics of Nansen Ice Shelf, East Antarctica"

_The Cryosphere, 2021_

## Referee Comment (RC1)

Review of **"The complex basal morphology and ice dynamics of Nansen Ice Shelf, East Antarctica"**
by Dow et al., submitted to The Cryosphere, July 2021

Reviewer: Ala Khazendar

**OVERVIEW**

In this manuscript, the authors analyze several data sets from the Nansen Ice Shelf in the Ross Sea, with focus on a suture zone at the confluence of two tributary glaciers. The authors reach conclusions on the morphology of the suture zone and its possible relationship to basal melting and channelized flow; and the suitability of the use of certain data sets.

The authors address a subject, suture zones, that is of increasing importance to the question of ice-shelf stability and evolution. Two of the data sets they present would be of much interest to the community, namely, ice-penetrating radar profiles across the suture zone and elsewhere on the ice shelf, and hydrographic glider measurements.

On the other hand, I found that some of the main conclusions drawn by the authors were not clearly supported by the available evidence, that some of the observations that could help in evaluating their arguments were not presented, or that their analyses did not take into consideration factors that could lead to different interpretations of the data. I believe that these issues need the attention of the authors, and I discuss them in detail below.

The manuscript is generally well written, making it possible to follow the arguments on the many aspects that the authors address. Some sections on related themes could potentially be combined to make the text flow better.

**REVIEW in DETAIL**

**Issues Related to the Hydrostatic Assumption and Data Use**

**L. 280** and elsewhere: The discussion of hydrostatic equilibrium
The authors write: "…Satellite-derived estimations of ice shelf draft data are limited by assumptions of hydrostatic balance, which do not take bridging stresses or pinning points into account."
This statement, and the ensuring discussion, raise several issues:

- What are the uncertainties of the REMA surface elevation data? These seem not to be considered in the analysis. Each 1 m of surface elevation uncertainty would translate into ~9 m of draft uncertainty, which would account for at least

part of the difference between REMA-inferred ice-shelf drafts and those from IPR.

- A similar argument applies to tidal effects. The authors dismiss those on L. 299 as "less than a meter in this region (Padman et al, 2002)." But, again, each meter of surface elevation variability due to tidal movement translates to ~9 m of draft estimates.
- The authors invoke "bridging stresses or pinning points (L. 282 and elsewhere)" to explain the limitations of the deriving ice-shelf draft from surface elevations. That might well be the case in parts of the ice shelf. But, are the authors implying that this is the case over all of the ice shelf? Pinning points are specific locations where the ice shelf is intermittently or permanently grounded on bathymetric features—a phenomenon distinct from bridging effects. Perhaps the authors could avoid lumping these two together, and clarify where they apply in the ice shelf.
- As a suggestion, what I think is a valuable discussion of features at the bottom of an ice shelf and their expression (or its absence) at the top surface can be found in Nicholls et al. (2006).

**L. 432**, Section 5.5 (Data applicability)

The authors write: "With the availability of high-resolution datasets such as REMA and GoLIVE, large-scale analyses of ice shelf characteristics can be made. However, from our application of multiple data sets including in situ data we find that some of the ice shelf properties are not well represented in the satellite-derived data sets.

The primary limitation of REMA is that hydrostatic calculations do not take into account bridging stresses and variability of ice rheology."

This is a theme that is emphasized by the authors. I fully agree with them that investigators should be careful in how they use satellite-derived data sets. The problem with the authors' stance as expressed and emphasized in the manuscript is that they appear to lay the blame with the data sets themselves. Statements such as the "primary limitation of REMA …" imply that this is an inherent defect of the data set. I disagree. The problem lies in the use of the data sets without paying the necessary attention to their limitations, which is a different matter.

**Issues Related to Ice-Shelf Basal Melting**

**L. 218**, Section 4.3 on ice shelf melt rates

The authors infer basal melting rates by a direct comparison of ice-shelf draft changes between two sites. Yet, ice-shelf thickness changes could result from several processes (e.g., Moholdt et al., 2014) including:
- velocity divergence as the ice shelf flows and spreads under gravity;
- downstream advection of thickness gradients;
- changes in surface mass balance;
- changes in firn air content that affect the density of the snow-firn-ice column;
- in addition to basal melting.

The firn air content might not be an issue due to the intensity of katabatic wind on NIS, as the authors point out. On the other hand, I think it would be helpful if the authors could clarify how they accounted for the other processes in inferring basal melting rates. For example, as they point out, "a river was observed in the surface depression as early as 1974 and annually from 2014-2016 (Bell et al., 2017)." The presence of this river in the surface depression that runs along the suture zone suggests that surface mass balance probably plays an important role in how the ice-shelf draft changes, not only basal melting.

**L. 382–394, L. 410** and elsewhere: The discussion of how the change in the shape of the suture zone suggest ocean-driven melting
As an example that summarizes the authors argument, they write on L. 410 "The higher ice loss rates in the middle of the suture zone compared to the thicker edges do, however, suggest an active melt component and that the suture zone is acting to channel water. "
Again, as discussed in the previous point, these arguments assume that all change in ice-shelf draft is due to the presumed ocean-induced basal melting, ignoring the others processes described above that could also modify the draft.

**L. 267**, Section 4.5 (Oceanography data)
The hydrographic glider measurements discussed by the authors could be one of the more interesting aspects of this work. It is therefore surprising that there are no figures presenting these data (other than the locations where fresher, cooler water was detected). Such figures could show:
  - temperature and salinity profiles with depth, and/or be T-S diagrams encompassing all relevant measurements in front of NIS;
  - the speeds and directions of the flow currents being discussed.
It is difficult to assess the authors' arguments and conclusions of this topic without such figures and the information they would convey.

**L. 412–422**: Discussion of melt water emergence from the suture zone.
In light of the absence of the hydrographic data described in the previous point, the statement by the authors that "meltwater observed by the glider underneath and directly offshore of the middle of the NIS calving front lines up with the region of enhanced melt within the suture zone and the thinner ice in the suture zone" is difficult to assess. What does "lines up" mean in this case? How is the origin of the observed melt water in the suture zone demonstrated? The authors do address some of the difficulty of establishing the connection between the suture zone and the melt water in L. 418-422.

**L. 486**, the "Conclusions" section:
The authors write "Our analysis of changes in ice morphology, flux gate volume, oceanographic data of freshwater and satellite-derived ice shelf melt all point toward active channelized melt within the suture zone."

This conclusion does not seem justified in light of the ambiguities of the analyses of ice-shelf morphology and oceanography discussed above.

**Issues Related to Marine Ice**

**L. 246**: The section entitled "Ice rheology" does not discuss rheology. The section mentions the two types of ice that might compose the ice shelf (meteoric and marine), but no rheological differences are discussed. For example, the early stages of marine ice formation could still be permeable (see next point below), making its temperature closer to the freezing point, hence contributing to a distinct rheology from that of the colder meteoric ice.

**L. 248**:  The authors write "…it is likely that these "echo-free zones" represent marine ice accumulation (Holland et al, 2009)."
This is a misunderstanding of Holland et al., 2009. When those authors, and others, state that marine ice basal returns are rarely detectable, they are referring to the *bottom* of the marine ice layer, not the *meteoric-marine ice interface*, which is often detectable by IPR. In other words, the absence of radar returns is likely not due to the presence of consolidated marine ice, as the interface between meteoric ice and any consolidated marine ice beneath it would have been detectable. For further background, the literature on marine ice in the Filchner-Ronne and Amery ice shelves (e.g., Fricker et al., 2001, which is already cited in the manuscript) is instructive.
A plausible hypothesis to explain the absence of radar returns is the presence of *unconsolidated* layers of a mushy mixture of frazil ice crystals and seawater and/or layers of slushy ice that have not yet fully consolidated to form solid marine ice.  Such mushy or slushy layers could result in the attenuation of radar signals without creating a clear interface where dielectric properties change abruptly, hence potentially accounting for the echo-free zones. Evidence for the possible presence of such unconsolidated frazil ice in the Nansen Ice Shelf itself can be found in Tison et al. (2001, I am a co-author of that work), and in Jansen et al. (2013) on the suture zones of the Larsen C Ice Shelf.

**Other Issues**

**L. 426**: "…However, as this strain transition occurred in the same region as the maximum melt it is potentially linked to the higher thinning rates of this region compared to further upstream."
There are a couple of issues with this statement:
- It is not clear how the strain transition is linked to the higher thinning rates. Could you please explain what you think are the mechanism(s) underlying such a link.
- This link, as presented here, appears to be mostly conjecture, yet on L. 488-490 (in the Conclusions section) it is reported as a more concrete finding of the work, which I do not think is justified, unless supported by further discussion/analyses.

**L. 138 (Figure 3)**: I apologize if a I missed something, but this figure does not seem to be correct. For example, it shows the suture zone to be thicker in the middle, unlike all the other figures.

**L. 64-73:** Regarding the flow speeds and thicknesses of the two tributary glaciers, why do the authors cite observations that are two decades old, rather than use the recent data available to them (e.g., GoLIVE; IPR)?

**L. 65-72:** A map of ice-shelf and tributary glacier flow speeds would be helpful in illustrating several features described here, including glacier speeds, the suture zone, and the flow around Teall Nunatak and Inexpressible Island. There are two figures showing surface elevation (2a and 4a), but none showing flow speeds.

**Minor Issues, Typos, etc.**

**Figure 1**: The label "Priestley Glacier" is difficult to read. Please consider other colors.

**L. 50**, and elsewhere; L. 127; L. 158; and L. 266: instead of "ice shelf terminus", "ice shelf edge", "ice shelf calving front" or "calving edge", consider more consistently using the same terminology, for example, "ice-shelf front".

**L. 73-74**: It might be helpful in Fig. 1a to point to where this surface-depression is.

**L. 223-224**: "…we compare the horizontal difference in basal draft…"
The word "horizontal" here is confusing. Perhaps "cross-sectional" or similar would be clearer.

**L. 260**: "… resulting in the formation of frigid ice shelf meltwater (potential temperature < -1.94°C) at or below the point of supercooling."
Please consider rephrasing to "… below the in situ freezing point, making the water supercooled" or similar, which is a bit clearer.

**L. 265**: "We plot the glider data representing the location of cold, fresh, ice shelf water on Figs. 2b, 5 and 1."
This might be a typo as Figure 1 does not show glider data.

**REFERENCES**

Jansen, D., Luckman, A., Kulessa, B., Holland, P. R., and King, E. C. (2013), Marine ice formation in a suture zone on the Larsen C Ice Shelf and its influence on ice shelf dynamics, *J. Geophys. Res. Earth Surf.*, 118, 1628– 1640, doi:10.1002/jgrf.20120.

Moholdt, G., Padman, L., and Fricker, H. A. (2014), Basal mass budget of Ross and Filchner-Ronne ice shelves, Antarctica, derived from Lagrangian analysis of ICESat altimetry, *J. Geophys. Res. Earth Surf.*, 119, 2361– 2380, doi:10.1002/2014JF003171.

Nicholls, K. W., et al. (2006), Measurements beneath an Antarctic ice shelf using an autonomous underwater vehicle, *Geophys. Res. Lett.*, 33, L08612, doi:10.1029/2006GL025998.

Tison, J.-L., Khazendar, A., and Roulin, E. (2001), A two-phase approach to the simulation of the combined isotope/salinity signal of marine ice, *J. Geophys. Res.*, 106( C12), 31387– 31401, doi:10.1029/2000JC000207.

---

## Referee Comment (RC2)

SUMMARY

Dow and colleagues present a multi-source dataset consisting of satellite data, airborne and ground-based radar and oceanographic data to describe the spatial variations in ice shelf draft, strain rates and relate them to sub-ice shelf melt and ice rheology. Finally, they make statements on the applicability of the remote sensing data sets to derive conclusions about the presented results.

GENERAL COMMENTS

The topics of understanding ice shelves by analyzing the spatial variations in draft and strain rates is a highly relevant topic with interesting results and the presentation of the new measurements is highly valuable for the research community. Yet, I think the current version of the manuscript shows still some major shortcomings (see detailed description below) that should be tackled.

In my opinion the main issue of the paper is the following:
The main conclusion of the authors is that they stress the limitations of the high-resolution data sets for deriving conclusions about ice shelf characteristics and I do not agree with this conclusion at all. It is true that their analyses show limitations, but this is might not be caused by the data itself but instead by the assumptions of the authors when using the data. For both the hydrostatic balance assumptions for REMA and the strain rates I can identify potential methodological inconsistencies that could affect the conclusions drawn in the paper (see more detailed comments below). Addressing this main comment will result probably in a highly changed paper where many of the results and conclusions need to be adapted.

SPECIFIC COMMENTS
- * It is often difficult to follow the structure of the paper with several sections that seem diluted between methods, results and discussion. Therefore, it is difficult to distil take-home messages from the paper and/or get an overview of the impact (e.g. I find it very difficult to summarize the paper). I think that by re-organizing the method, results, discussion in single sections on morphology, strain, melt and rheology would improve the readability a lot. Now for example, for the ice morphology section it is not very clear and rather arbitrary where the site description ends, where the results start and what the discussion is. The switch between results and discussion often feels very arbitrary and makes it difficult to follow as the flow is interrupted.
- * The abstract seems to read as a loose collection of individual sentences, which make it difficult (for me) to follow. I think that using some bridging words/terminology between the sentences would increase the readability. *Note: these are a subjective comments (my apologies) and I do see that R1 did make a more positive comment on the writing, so do not see it as hard advice. Nevertheless, it might also be an indication that it might be difficult for others to follow.
- Many of the results paragraphs are very descriptive paragraphs with results that are difficult to "see" in the figures. I think it would be beneficial if the figures would use direct labelling instead of looking them up in the caption which requires zigzagging

between the figure and label. By using text, arrows etc. to indicate where to look, it will be way easier to interpret the figures.

- L51-54 "*Around the time of this event, the ice surface strain patterns changed from extensional across-ice to extensional down-ice within ~8 km of the calving front and drove the formation of a new fracture over the thinnest region of the central NIS*"-> Not clear if this a new result or part of Dow et. al. (2018). The terms across-ice and down-ice are also very difficult to interpret as I would guess this is about the flow and not the ice. Perhaps use across-flow and down-flow?

- L59-60 "*we make recommendations of where and when satellite data is sufficient to analyse ice shelf properties without in situ data.*" I do not agree with these later recommendations (see my later comments) as this is not about the data but about the methodologies and assumptions behind the methodologies. This is therefore largely a data handling problem and not a data problem.

- L74-76: "A 30-km long depression + a river was observed + transverse fracture" -> if these features are important, they should be drawn and labelled (directly) on the map in Fig.1.

- Fig.1: perhaps direct labelling instead of the caption as it requires the reader to unnecessary zigzag between figure and caption .

- The site description could be integrated with the section 4.1 to join similar information together.

- L102 "*it is sufficiently sparse that DEMs interpolated from these data have significant errors in regions that lack high data density*" -> complicated sentence for simple thing, perhaps rephrase to "*it is too sparse to accurately interpolate*"

- L103 If REMA strips (and not the mosaic) are used, then they lack bias and tilt corrections (which could be significant), tide corrections, inverse barometer effects etc. All these corrections are necessary to draw any conclusion on the ice draft etc. Just assuming the strips are correct is methodological not-correct (which can explain many of the later critique on the REMA data) as these raw strips were never provided to be used without the necessary corrections and without them the absolute elevations (and hence drafts) cannot be interpreted

- L105: What was the offset between the strips? How was it corrected?

- L109-114: In my opinion the assumption of pure blue ice is a wrong assumption as the ice shelf locally (especially in the channels) seem to contain snow/firn cover. When this snow/firn is not taken into account any conclusion on bridging stresses is potentially wrong (see also later).

- Fig.2: perhaps use direct labelling for the green/yellow circle and lines instead of the caption as it requires the reader to unnecessary zigzag between figure and caption.

- Fig.2: it took me some time to understand that the red-dashed line was moved to overlap draft with meltwater. It could be helpful to indicate that in the figure (e.g. with arrows)

[Figure]

- Fig.3: the alignment for REMA is potentially problematic if the proper strip corrections are not applied
- L150: what about the spatial and overall accuracy of GOLive data? Small inconsistencies in the velocity data and/or geometric accuracies could have large impacts on the interpretation of the strain rates and I think this should be accounted for in the later analyses. Just assuming the velocity (which may be a wrong assumption) are correct can result in the observed misrepresentations (see also later comment)
- L160-161 " which produced similar results in both spatial pattern and magnitude of strain rates": what is similar? What are the differences? Perhaps quantify etc.
- Section 3.5: these are very interesting data and should be elaborated further. How was the meltwater classification done?
- Section 4.1: this reads very much as a continuation of the site description and could perhaps be integrated
- L175-185: these features are difficult to find and see on the map for a non-experienced reader. Would be a good idea to help the reader and indicate all the described features on the map.
- L197-199: "*Along the center of the suture zone there is an alternating region of horizontal compression (red) on the northern side and extension (blue) on the southern side: both regions have widths of ~800 m. When compared with the ice shelf draft, the switch between compression and extension occurs at the apex of the thin-ice suture zone region (Fig. 4d)*". I do not agree with these statements as I think it is almost impossible to interpret the strain locations relative to the radar given i) that the strain rate is only calculated every 300m and ii) the potential (spatial) uncertainty in the velocity data. For example, shifting the compression peak 150m to the right (which seems well within the strain uncertainty) would result in a compression peak that is nicely aligned with the apex (which would also make more sense given the discussion

later):

[Figure]

- L199-200: *"the switch between compression and extension … is limited horizontally"* This switch is by definition always localized to a point (as there is either compression or extension) so it seems a strange statement

- L200-206: any of the conclusions based on the location of patterns of strain vs. apex-keels are debatable as these patterns can be easily (mis-)aligned when using small shifts (which seem within the uncertainty of the data). I therefore do not think statements on alignment can be made and the assumption that it can be derived from satellite velocities is potentially overambitious.

- L224-227: *"The northern side of the suture zone has minimal change in relative basal draft but approaching the apex of the thin ice region, there is substantially more ice loss. The greatest mass loss is in the highest apexes of the central suture zone and the basal fractures on the southern side, with the keels of the latter relatively unchanged compared to the spatially-constant background melt rate."* It might be due to my misunderstanding of the methodology of alignment, but I do not necessarily agree. If you align the draft for the apex (instead of the edges (see example below)), you could conclude that the largest changes occurred at the edges and not at the apex.

[Figure]

- Fig.5. It is very difficult to interpret the relative basal draft (what is it, how was it calculated/quantified)

- L239-246: the analysis of the melt rates derived from Cryosat-2 is potentially very interesting and should be elaborated on further. What would be the total melt of transect of C-C' was advected (with the velocity) to A-A'? Would this integrated melt show a similar (smoothed) pattern as the simple observed difference between Site1.

- L249-251: *"Mapping these regions on top of the REMA 2016 hydrostatic thickness map shows they are all associated with thinner regions of ice and, in particular, with basal fractures on both sides of the suture zone along with the thinnest portion of ice in the*

*suture zone*" Where can I see this? It would be useful to replace Fig.6a with this overlay over REMA

- Section 4.5: although the data are very interesting I do find it very difficult to see any conclusion, take home message from this section. Here again it would be beneficial if integrated with the discussion section to remove the fragmentation and increase the impact
- Section 4.6: One of the main potential errors in the hydrostatic assumption is that the ice shelf is completely snow/firn free, whereas for example Sentinel-2 data shows that there is snow deposition/firn over transect A-A' which could provide an explanation for the lack of /muted channels in the hydrostatic REMA approximation. Especially as the snow/firn cover seems stronger in the south where larger offsets in the Dow analysis occur.

[Figure]

- L296-304: I do not agree with any of this paragraph as (also indicated earlier) i) the REMA strips require several corrections (tides, tilt+offset effects, barometric effect) before they allow to convert to draft ii) the snow/firn could result in local biases as well. Both these forgotten corrections makes the interpretation of the hydrostatic figures very much dependent on potentially wrong assumptions as both offsets+snow could result in similar results. Therefore the conclusion of bridging stresses is not necessarily supported here.
- Section 5.1 reads very much as a continuation of the results of 4.1 and perhaps it should be considered to be integrated.
- L335-339: many of the statements (e.g. alignment, bridging stress) are not necessarily supported by the results (see my earlier comments) and therefore I doubt the correct interpretation of this paragraph
- L435 " *we find that some of the ice shelf properties are not well represented in the satellite-derived data sets.*": see my earlier comments, but again I do not think it is a fair comment to blame the data. These mis-representations are either the potential result of wrong assumptions (e.g. for hydrostatic balance) or by using the data (e.g. strain rate) without accounting for inherent (spatial) uncertainties that should be accounted for.
- L436-451: I do not agree (see my earlier comments)
- L455 "*If only longitudinal strain had been calculated, these features would have been missed*" Yes, but why would you only calculate longitudinal strain and neglect

transverse strain? This is again not a problem of the data, but of a potential wrong assumption (fracture dynamics rely only on longitudinal strain).

TECHNICAL COMMENTS

- L20 "*Nansen Ice Shelf has a highly variable morphology*"
- L41-45: "*The increasing variability of … opens up possibilities*"
- Caption  Fig.2 "*The green circle highlights an area referenced in the text*" This is a rather non-helpful caption as I now still don't know what I am looking at (and why?) and requires me to go and search the paper
- Fig.2 and later figures: I do find it confusing that S1-S2-S3 and A-A', B-B', C-C' are used interchangeably. Would be clearer with consistency throughout all figures etc.
- Line 166: difficult for non-experts to see this suture zone. Perhaps direct label the suture zone on Fig.2?
- Fig.4: it would be beneficial if the strain rates were overlaid (semi-transparent in color) over the REMA DEM (e.g. in grey) as it would allow to link the strain to the DEM. Now it is basically impossible to see direct linkages between the different panels.
- Fig.4 colorbar: it would be beneficial if compression and extension is directly labelled on the colorbar as it would make things clearer without the need to read the entire caption.
- L194: "*The extent of the region in Fig. 4b is shown in Fig. 2*". Sentence is obsolete in the main text (can be part of caption) and breaks the flow.
- Fig. 5b: perhaps add transect B-B' (S2) to allow the reader to check for temporal consistency etc?

---

## Author Comment (AC1)

OVERVIEW

In this manuscript, the authors analyze several data sets from the Nansen Ice Shelf in the Ross Sea, with focus on a suture zone at the confluence of two tributary glaciers.

The authors reach conclusions on the morphology of the suture zone and its possible relationship to basal melting and channelized flow; and the suitability of the use of certain data sets.

The authors address a subject, suture zones, that is of increasing importance to the question of ice-shelf stability and evolution. Two of the data sets they present would be of much interest to the community, namely, ice-penetrating radar profiles across the suture zone and elsewhere on the ice shelf, and hydrographic glider measurements. On the other hand, I found that some of the main conclusions drawn by the authors were not clearly supported by the available evidence, that some of the observations that could help in evaluating their arguments were not presented, or that their analyses did not take into consideration factors that could lead to different interpretations of the data. I believe that these issues need the attention of the authors, and I discuss them in detail below.

The manuscript is generally well written, making it possible to follow the arguments on the many aspects that the authors address. Some sections on related themes could potentially be combined to make the text flow better.

*Thank you for your helpful comments on this manuscript. We have addressed your concerns listed above in the response below.*

REVIEW in DETAIL

Issues Related to the Hydrostatic Assumption and Data Use

L. 280 and elsewhere: The discussion of hydrostatic equilibrium

The authors write: "…Satellite-derived estimations of ice shelf draft data are limited by assumptions of hydrostatic balance, which do not take bridging stresses or pinning points into account."

This statement, and the ensuring discussion, raise several issues:

- What are the uncertainties of the REMA surface elevation data? These seem not to be considered in the analysis. Each 1 m of surface elevation uncertainty would translate into ~9 m of draft uncertainty, which would account for at least part of the difference between REMA-inferred ice-shelf drafts and those from IPR.

*We realise now following these comments and those from reviewer 2 that we were not clear enough about how and why we conducted our analysis of hydrostatic balance. In Figure 2 we do apply hydrostatic analysis to REMA but this is essentially for visual purposes only and we don't*

*use it for small-scale analysis within the manuscript. We do use it to discuss general thickness trends in section 4.1 but this is a large-scale (multiple kilometer) analysis and therefore smaller scale errors are not relevant. Instead, for all small-scale comparisons we used our in situ GPS surface data along with radar thickness data collected concurrently on the ice shelf. We have clarified this throughout the manuscript and in particular in the methodology where we now discuss hydrostatic correction more generally rather than only in application to REMA. In the hydrostatic equilibrium results we have clarified to say:*

*"Ice shelf draft estimations that are extrapolated using hydrostatic balance calculations do not take bridging stresses or pinning points into account and therefore may contain errors. Our in situ ground-based and airborne radar thickness data provide high-resolution data of ice shelf draft features and can be compared with hydrostatically-derived thickness using simultaneously collected GPS surface elevation data."*

*We have moved away from assessing the satellite datasets and instead focus on applicability of the datasets as suggested later in the review (also see our response to that comment below). As such we have removed section 5.5 'Dataset applicability' and moved the relevant parts of that section to other areas of the Discussion. Addressing the above comment, we have also clarified our approach to hydrostatic calculations in the new version of Discussion section 5.1, where we state that we are using in situ surface elevation data..*

- A similar argument applies to tidal effects. The authors dismiss those on L. 299 as "less than a meter in this region (Padman et al, 2002)." But, again, each meter of surface elevation variability due to tidal movement translates to ~9 m of draft estimates.

*Yes this is the case, but we continue in this section to explain that the surveys were conducted within a week of each other and checked with a cross-over analysis (of both the surface elevation and ice thickness). Now that we have clarified that we are discussing in situ dGPS measurements of surface elevation rather than REMA surface elevation we hope that this is satisfactory. We also point out that the differences in measured ice thickness and thickness assumed using hydrostatic calculations is on the order of 10s of meters rather than <10m.*

- The authors invoke "bridging stresses or pinning points (L. 282 and elsewhere)" to explain the limitations of the deriving ice-shelf draft from surface elevations. That might well be the case in parts of the ice shelf. But, are the authors implying that this is the case over all of the ice shelf? Pinning points are specific locations where the ice shelf is intermittently or permanently grounded on bathymetric features—a phenomenon distinct from bridging effects. Perhaps the authors could avoid lumping these two together, and clarify where they apply in the ice shelf.

*This is a good point and we have rephrased to clarify:*

*"This suggests the difference in thickness at Site 1 and 2 between measured and calculated could instead be due to bridging stresses from the highly variable basal draft and/or from pinning of the ice shelf from valley walls or nunataks. However, the nearest pinning points from Site 1 are the Teal Nunatak at the Reeves Glacier grounding line and Inexpressible Island, both ~15 km from Site 1. Given this distance, it is more likely that bridging stresses are impacting the narrower basal draft features in the suture zone of the ice shelf (McGrath et al, 2012; Bassis and Ma, 2015), although this would have to be confirmed using modelling approaches."*

- As a suggestion, what I think is a valuable discussion of features at the bottom of an ice shelf and their expression (or its absence) at the top surface can be found in Nicholls et al. (2006).

*Thank you for this suggestion. We have added this reference into our discussion of the differences between surface depressions and basal topography.*

L. 432, Section 5.5 (Data applicability)

The authors write: "With the availability of high-resolution datasets such as REMA and GoLIVE, large-scale analyses of ice shelf characteristics can be made. However, from our application of multiple data sets including in situ data we find that some of the ice shelf properties are not well represented in the satellite-derived data sets. The primary limitation of REMA is that hydrostatic calculations do not take into account bridging stresses and variability of ice rheology."

This is a theme that is emphasized by the authors. I fully agree with them that investigators should be careful in how they use satellite-derived data sets. The problem with the authors' stance as expressed and emphasized in the manuscript is that they appear to lay the blame with the data sets themselves. Statements such as the "primary limitation of REMA ..." imply that this is an inherent defect of the data set. I disagree. The problem lies in the use of the data sets without paying the necessary attention to their limitations, which is a different matter.

*This is a good point, and one also made by reviewer 2. In the new version of the manuscript we no longer discuss the limitations of the datasets but instead focus on how the data are utilized, as suggested. We have removed reference to this in the introduction and we have removed section 5.5 order to address this. Instead the discussion on hydrostatic balance has been moved to the discussion section on complex ice morphology (section 5.1), the discussion on strain to section 5.2 (ice shelf fractures and strain rates), and the discussion on melt to section 5.4 (ice shelf melt). All of these parts of former section 5.5 have been adjusted to fit in with their new sections and discussion focussed on interpretation of the data rather than issues with the datasets themselves.*

Issues Related to Ice-Shelf Basal Melting

L. 218, Section 4.3 on ice shelf melt rates

The authors infer basal melting rates by a direct comparison of ice-shelf draft changes between two sites. Yet, ice-shelf thickness changes could result from several processes (e.g., Moholdt et al., 2014) including:

• velocity divergence as the ice shelf flows and spreads under gravity;

• downstream advection of thickness gradients;

• changes in surface mass balance;

• changes in firn air content that affect the density of the snow-firn-ice column;

• in addition to basal melting.

The firn air content might not be an issue due to the intensity of katabatic wind on NIS, as the authors point out. On the other hand, I think it would be helpful if the authors could clarify how they accounted for the other processes in inferring basal melting rates. For example, as they point out, "a river was observed in the surface depression as early as 1974 and annually from 2014-2016 (Bell et al., 2017)." The presence of this river in the surface depression that runs along the suture zone suggests that surfac mass balance probably plays an important role in how the ice-shelf draft changes, not only basal melting.

*Following this comment and a similar comment by reviewer 2 we re-examined our calculations of basal draft change and found that our technique was not correct and was misleading in terms of the location of most melt. We have therefore removed the basal draft relative change comparison from Figure 5b and the related discussion in section 4.3 (ice shelf melt rates). Instead of this prior focus on change in basal draft, we now examine changes in ice thickness between the sites. We have also now used the Cryosat-2 basal melt data to calculate cumulative melt rates between Site 1 and Site 3. We then compared the melt over this distance with the thickness change between the two transects at Site 1 and 3. We find that the basal melt from the Cryosat-2 analysis can explain approximately 50% of the thickness change. To investigate the remaining thickness change we run the same cumulative melt integration between these two transects for a surface melt rate of 0.25m/year (following Bromwich and Kurtz (1984), and also 0.5 m/year (following Bell et al, 2017). Adding this surface melt rate to the Cryosat-2 basal melt rate produces a thickness change within a reasonable range of the thickness change from the Site 1 and 3 radar transects. This shows the steeper sides of the suture zone are melting by ~120 m between these two transects, but only by 60 m in the center. The change in ice thickness at the suture zone center is primarily due to surface ablation but as the ice thickens towards the side of the suture zone basal melt from oceanic processes is also occurring. In general the influence of*

*the basal and surface melting appears to be 50/50. Following from our error with the relative basal draft calculations this new analysis has provided more accurate information about where active melt is taking place. We have adjusted the manuscript accordingly when presenting the results in Section 4.3, when discussing the role of ocean melt, and in the conclusions; we now suggest that the primary role is on the slopes of the suture zone rather than channelised in the thinnest region of the suture zone. We have also replaced the subfigure in Figure 5b with one showing the integrated basal melt, surface ablation, total melt and measured thickness change.*

*We also agree many aspects could impact mass change and we have covered these in lines 219-229. By using multiple techniques to assess whether the mass change is as a result of dynamic thinning or due to melting we are narrowing the options, although it is still difficult to definitively state exactly how much melting occurs without in situ measurement by drilling into the ice shelf, for example. We directly address the potential for velocity divergence, downstream thickness advection and surface mass balance, and we believe our new analysis as described above has strengthened our arguments. Some of the causes can be discounted, such as changes in the firn air content as we are in a blue ice zone and there is no firn. And the surface river may erode some surface ice, but this is on the range of ~2m and limited to a width of <200m. It therefore cannot explain the larger scale change in mass that we observe between the sites. We don't have an analysis of longitudinal extension that may thin the ice although the longitudinal strain map in Figure 4c doesn't indicate consistent extensional strain (with instead stripes of extension and compression) which would usually be associated with enhanced longitudinal thinning. We have expanded on these various points in the text along with substantial alterations to Section 4.3 following the analysis described in the paragraph above.*

L. 382–394, L. 410 and elsewhere: The discussion of how the change in the shape of the suture zone suggest ocean-driven melting As an example that summarizes the authors argument, they write on L. 410 "The higher ice loss rates in the middle of the suture zone compared to the thicker edges do, however, suggest an active melt component and that the suture zone is acting to channel water. " Again, as discussed in the previous point, these arguments assume that all change in ice-shelf draft is due to the presumed ocean-induced basal melting, ignoring the others processes described above that could also modify the draft.

*See above response. We have now removed these sentences from the manuscript.*

L. 267, Section 4.5 (Oceanography data)

The hydrographic glider measurements discussed by the authors could be one of the more interesting aspects of this work. It is therefore surprising that there are no figures presenting these data (other than the locations where fresher, cooler water was detected). Such figures could show:

- temperature and salinity profiles with depth, and/or be T-S diagrams

encompassing all relevant measurements in front of NIS;

- the speeds and directions of the flow currents being discussed.

It is difficult to assess the authors' arguments and conclusions of this topic without such figures and the information they would convey.

> *We are glad that these glider data are of interest. We are, unfortunately, limited in how much data that we can present within this manuscript as they are under review and embargoed for a different journal with focus on the oceanography conditions. A velocity instrument was not present on the glider so we are unable to present data on the speed and direction of the flow currents. We have, however, added a T-S diagram to Figure 2 demonstrating the water column characteristics for the region where meltwater was detected. We also direct the reader towards Friedrichs et al (in review) for additional data.*

L. 412–422: Discussion of melt water emergence from the suture zone.

In light of the absence of the hydrographic data described in the previous point, the statement by the authors that "meltwater observed by the glider underneath and directly offshore of the middle of the NIS calving front lines up with the region of enhanced melt within the suture zone and the thinner ice in the suture zone" is difficult to assess. What does "lines up" mean in this case? How is the origin of the observed melt water in the suture zone demonstrated? The authors do address some of the difficulty of establishing the connection between the suture zone and the melt water in L. 418-422.

> *Our comparison between the location of the water and the suture zone is achieved by examining the thickness of the ice shelf upstream of the freshwater signals. We have clarified this in Figure 2 by adding arrows as suggested by reviewer 2. We also clarify where we mentioned 'lined up' by changing the wording to say:*
>
> *"The meltwater observed by the glider underneath and directly offshore of the middle of the NIS calving front appears to be spatially aligned with the suture zone. However, the range of depth in the meltwater recorded by the glider means that it is hard to determine where that water originated from, although some is close to the ice draft depth of 150-190 m in the central region (Fig. 2c)."*
>
> *The caveat in the second sentence and our change in wording now better reflects that there is some uncertainty in the origin given that the meltwater wasn't all measured directly beneath the shelf.*

L. 486, the "Conclusions" section:

The authors write "Our analysis of changes in ice morphology, flux gate volume, oceanographic data of freshwater and satellite-derived ice shelf melt all point toward active channelized melt within the suture zone."

This conclusion does not seem justified in light of the ambiguities of the analyses of ice-shelf morphology and oceanography discussed above.

> *We have adjusted this sentence to indicate uncertainties in the measurements, our removal of the relative change in basal draft and replacement with the thickness change and cumulative Cryosat-2 melt analysis. It now reads:*
>
> *"Our analysis of changes in ice morphology, flux gate volume, oceanographic data of freshwater, and satellite-derived ice shelf melt suggest that active oceanographic melt may be occurring within the suture zone focused toward the thicker edge of the suture zone rather than the center, with the caveat that in situ sampling is required to fully determine the location and rates of melt."*

Issues Related to Marine Ice

L. 246: The section entitled "Ice rheology" does not discuss rheology. The section mentions the two types of ice that might compose the ice shelf (meteoric and marine), but no rheological differences are discussed. For example, the early stages of marine ice formation could still be permeable (see next point below), making its temperature closer to the freezing point, hence contributing to a distinct rheology from that of the colder meteoric ice.

> *Thank you for pointing this out. We have retitled that section (4.4) to say 'Marine Ice' and elsewhere change 'ice rheology' to 'ice provenance' to clarify that we are discussing how the ice was formed.*

L. 248: The authors write "…it is likely that these "echo-free zones" represent marine ice accumulation (Holland et al, 2009)."

This is a misunderstanding of Holland et al., 2009. When those authors, and others, state that marine ice basal returns are rarely detectable, they are referring to the *bottom* of the marine ice layer, not the *meteoric-marine ice interface*, which is often detectable by IPR. In other words, the absence of radar returns is likely not due to the presence of consolidated marine ice, as the interface between meteoric ice and any consolidated marine ice beneath it would have been detectable. For further background, the literature on marine ice in the Filchner-Ronne and Amery ice shelves (e.g., Fricker et al., 2001, which is already cited in the manuscript) is instructive. A plausible hypothesis to explain the absence of radar returns is the presence of *unconsolidated* layers of a mushy mixture of frazil ice crystals and seawater and/or layers of slushy ice that have not yet fully consolidated to form solid marine ice. Such mushy or slushy layers could result in the attenuation of radar signals without creating a clear interface where

dielectric properties change abruptly, hence potentially accounting for the echo-free zones. Evidence for the possible presence of such unconsolidated frazil ice in the Nansen Ice Shelf itself can be found in Tison et al. (2001, I am a co-author of that work), and in Jansen et al. (2013) on the suture zones of the Larsen C Ice Shelf.

*We have now clarified this in section 4.4 by changing "it is likely that these "echo-free zones" represent marine ice accumulation" to "it is likely that these "echo-free zones" represent marine ice formation and/or the presence of frazil ice". We then continue by clarifying following your suggestions above by stating:*

*"Mapping these regions on top of the REMA hydrostatic thickness map shows they are all associated with thinner regions of ice and, in particular, with basal fractures on both sides of the suture zone along with the thinnest portion of ice in the suture zone. The lack of radar echos in these regions are likely due to the presence of unconsolidated frazil ice accumulating at the base of the ice shelf (Tison and Khazendar, 2001; Jansen et al, 2013)"*

*Some of our echo free zones are full-thickness rifts originating from the Reeves grounding line (following your own work in the region) and as such, there is no meteoric-marine transition because there is no meteoric ice present. We clarify this in this section by saying:*

*"On the ice surface, in the suture zone, there are many stripes of clear blue ice between larger regions of white aerated ice which also produce no radar echos. These can be traced back to the Reeves Glacier ice fall where crevasses fully fracture through the ice column, fill with sea water and refreeze (Khazendar et al, 2001)."*

Other Issues

L. 426: "…However, as this strain transition occurred in the same region as the maximum melt it is potentially linked to the higher thinning rates of this region compared to further upstream."

There are a couple of issues with this statement:

- It is not clear how the strain transition is linked to the higher thinning rates. Could you please explain what you think are the mechanism(s) underlying such a link.

- This link, as presented here, appears to be mostly conjecture, yet on L. 488-490 (in the Conclusions section) it is reported as a more concrete finding of the work, which I do not think is justified, unless supported by further discussion/analyses.

*We agree that our meaning here was not clear and have changed to say:*

*"However, as this strain transition occurred in the same region as the maximum melt, this region may be associated with a reduction in buttressing on the valley walls as a result of thinning ice, compared to further upstream"*

*The longitudinal extensional strain regime only reached from the terminus to partially up the ice shelf and therefore it is likely inhibited by stronger buttressing further upstream.*

*The link between the change in strain direction (from transverse to longitudinal) and the location of the new 2016 fracture was discussed in Dow et al (2018) and is referenced in lines 416-419 where we say:*

*"The formation of this fracture was argued to be due to an alteration of the strain regime from transverse to longitudinal extension between 2014 and 2015, linked closely to the expansion and the calving of a fracture much closer to the ice front (Dow et al, 2018)."*

*As such we argue that the strain regime of the ice shelf does play a significant role in fracturing of the ice shelf and suggest that it should remain in the conclusion.*

L. 138 (Figure 3): I apologize if a I missed something, but this figure does not seem to be correct. For example, it shows the suture zone to be thicker in the middle, unlike all the other figures.

*Our y-axes have the thicker ice at the top and thinner ice at the bottom, therefore the suture zone is still thinner. We have added the axes labels to all three subfigures to make it clearer.*

L. 64-73: Regarding the flow speeds and thicknesses of the two tributary glaciers, why do the authors cite observations that are two decades old, rather than use the recent data available to them (e.g., GoLIVE; IPR)?

*Good point! We've changed this to use the Go-Live and IPR data.*

L. 65-72: A map of ice-shelf and tributary glacier flow speeds would be helpful in illustrating several features described here, including glacier speeds, the suture zone, and the flow around Teall Nunatak and Inexpressible Island. There are two figures showing surface elevation (2a and 4a), but none showing flow speeds.

*We have added an extra panel to Figure 2 to show the ice velocity in the region.*

Minor Issues, Typos, etc.

Figure 1: The label "Priestley Glacier" is difficult to read. Please consider other colors.

*We have altered the Priestley Glacier labels and adjusted the colors to make the figure clearer.*

L. 50, and elsewhere; L. 127; L. 158; and L. 266: instead of "ice shelf terminus", "ice shelf edge", "ice shelf calving front" or "calving edge", consider more consistently using the same terminology, for example, "ice-shelf front".

*We have changed the manuscript to refer only to ice-shelf front.*

L. 73-74: It might be helpful in Fig. 1a to point to where this surface-depression is.

*We now add in a label showing the location of the river in Figure 2a which will indicate the location of the surface depression. We looked at also adding an indicator of the surface depression to the satellite image on Figure 1a but it was difficult to fit with the scale of the image and the labels so in this line we now direct readers to the elevation map of Figure 2 to see the surface depression.*

L. 223-224: "…we compare the horizontal difference in basal draft…" The word "horizontal" here is confusing. Perhaps "cross-sectional" or similar would be clearer.

*We have changed this to 'cross-sectional' as suggested.*

L. 260: "… resulting in the formation of frigid ice shelf meltwater (potential temperature < -1.94°C) at or below the point of supercooling." Please consider rephrasing to "… below the in situ freezing point, making the water supercooled" or similar, which is a bit clearer.

*We have changed to "resulting in the formation of ice shelf meltwater below the in situ freezing point (potential temperature < -1.94°C), making the water supercooled"*

L. 265: "We plot the glider data representing the location of cold, fresh, ice shelf water on Figs. 2b, 5 and 1." This might be a typo as Figure 1 does not show glider data.

*Thanks for catching this. Corrected.*

---

## Author Comment (AC2)

Dow and colleagues present a multi-source dataset consisting of satellite data, airborne and ground-based radar and oceanographic data to describe the spatial variations in ice shelf draft, strain rates and relate them to sub-ice shelf melt and ice rheology. Finally, they make statements on the applicability of the remote sensing data sets to derive conclusions about the presented results.

GENERAL COMMENTS

The topics of understanding ice shelves by analyzing the spatial variations in draft and strain rates is a highly relevant topic with interesting results and the presentation of the new measurements is highly valuable for the research community. Yet, I think the current version of the manuscript shows still some major shortcomings (see detailed description below) that should be tackled.

In my opinion the main issue of the paper is the following:

The main conclusion of the authors is that they stress the limitations of the high-resolution data sets for deriving conclusions about ice shelf characteristics and I do not agree with this conclusion at all. It is true that their analyses show limitations, but this is might not be caused by the data itself but instead by the assumptions of the authors when using the data. For both the hydrostatic balance assumptions for REMA and the strain rates I can identify potential methodological inconsistencies that could affect the conclusions drawn in the paper (see more detailed comments below). Addressing this main comment will result probably in a highly changed paper where many of the results and conclusions need to be adapted.

*Thank you for your constructive comments on this manuscript. Reviewer 1 also pointed out that we misrepresent issues with usage of data as problems with the data itself rather than its application. We have removed reference to this in the introduction and we have removed section 5.5 order to address this. Instead the discussion on hydrostatic balance has been moved to the discussion section on complex ice morphology (section 5.1), the discussion on strain to section 5.2 (ice shelf fractures and strain rates), and the discussion on melt to section 5.4 (ice shelf melt). All of these parts of former section 5.5 have been slightly adjusted to fit in with their new sections and discussion focussed on interpretation of the data rather than issues with the datasets themselves.*

*We have directly addressed your comments on our usage of REMA and strain rates below. In short, we very minimally use REMA in this manuscript, with our hydrostatic analysis focused on in situ surface elevation data from GPS collected simultaneously with our ice penetrating radar thickness data. We were not sufficiently clear about this in the manuscript and have corrected this. However, this removes concerns about directly analysing REMA data for hydrostatic calculations. In terms of strain data we agree that the datasets have associated errors but we have expanded upon our explanation of the application demonstrating that the patterns we observe are consistent across multiple years. We are also interested in the pattern rather than the magnitude of strain for this manuscript and therefore we suggest the consistency in strain patterns over multiple years supports our arguments. We have adjusted our discussion and conclusion following suggestions from both yourself and reviewer 1, but our larger scale arguments remain the same.*

SPECIFIC COMMENTS

* It is often difficult to follow the structure of the paper with several sections that seem diluted between methods, results and discussion. Therefore, it is difficult to distil take-home messages from the paper and/or get an overview of the impact (e.g. I find it very difficult to summarize the paper). I think that by re-organizing the method, results, discussion in single sections on morphology, strain, melt and rheology would improve the readability a lot. Now for example, for the ice morphology section it is not very clear and rather arbitrary where the site description ends, where the results start and what the discussion is. The switch between results and discussion often feels very arbitrary and makes it difficult to follow as the flow is interrupted.

*Thank you for your suggestion. However, we are following standard Cryosphere guidelines for paper layout, beginning with introduction, study area, methods, followed by results and the discussion. It would also be challenging to separate these entirely as some of the analysis for each method is included in the same discussion section. As such, we have kept the layout as it is. However, we have changed the wording in the introduction to state that we are examining each of these areas of research, which we hope makes the paper clearer .*

*"We analyze and compare several datasets including a satellite-derived digital elevation model (DEM), ground-based and airborne ice-penetrating radar (IPR), ice velocity-derived strain rate data, and oceanographic data. These data sets are used to examine the morphology of NIS and how this interacts with ice strain, basal melt and ice, all important factors in shelf fracturing"*

* The abstract seems to read as a loose collection of individual sentences, which make it difficult (for me) to follow. I think that using some bridging words/terminology between the sentences would increase the readability. *Note: these are a subjective comments (my apologies) and I do see that R1 did make a more positive comment on the writing, so do not see it as hard advice. Nevertheless, it might also be an indication that it might be difficult for others to follow.

*We have examined the abstract within the space constraints and believe that our current approach combines sufficient information and the main points of the paper. Given the support of reviewer 1, and your subsequent comment that your suggestions were facultative, we did not rewrite the Abstract. We have altered it slightly, however, to account for our move away from examining the applicability of satellite datasets.*

Many of the results paragraphs are very descriptive paragraphs with results that are difficult to "see" in the figures. I think it would be beneficial if the figures would use direct labelling instead of looking them up in the caption which requires zigzagging between the figure and label. By using text, arrows etc. to indicate where to look, it will be way easier to interpret the figures.

*We have added text and arrows into Figures 2 and 4 to aid with this including labelling the fracture and the river location. We have also added a legend to Figure 1.*

L51-54 "*Around the time of this event, the ice surface strain patterns changed from extensional across-ice to extensional down-ice within ~8 km of the calving front and drove the formation of a new fracture over the thinnest region of the central NIS*"-> Not clear if this a new result or part of Dow et. al. (2018). The terms across-ice and down-ice are also very difficult to interpret as I would guess this is about the flow and not the ice. Perhaps use across-flow and down-flow?

> *Yes, this is part of Dow et al (2018) and to clarify we have added in the reference again at the end of the sentence. We have changed the wording to across-flow and down-flow as suggested.*

L59-60 "*we make recommendations of where and when satellite data is sufficient to analyse ice shelf properties without in situ data.*" I do not agree with these later recommendations (see my later comments) as this is not about the data but about the methodologies and assumptions behind the methodologies. This is therefore largely a data handling problem and not a data problem.

> *We agree with your comments and have removed discussion of this and have deleted this sentence from the introduction (see our response in the general comment section).*

L74-76: "A 30-km long depression + a river was observed + transverse fracture" -> if these features are important, they should be drawn and labelled (directly) on the map in Fig.1.

> *As Figure 1 is to introduce the location of the NIS and the radar survey lines, we have instead added the river and fracture labels to Figure 2 which includes the ice surface and thickness information and have referenced Fig 2a in this sentence.*

Fig.1: perhaps direct labelling instead of the caption as it requires the reader to unnecessary zigzag between figure and caption .

> *We have added a legend to describe the radar lines in Figure 1.*

The site description could be integrated with the section 4.1 to join similar information together.

> *We experimented with this but struggled with this setup because the hydrostatic calculations have to be introduced before we can discuss the ice morphological features. We also feel like we can't move the site description entirely as this is part of the introduction to the region. As such we have retained our study site section.*

L102 *"it is sufficiently sparse that DEMs interpolated from these data have significant errors in regions that lack high data density"* -> complicated sentence for simple thing, perhaps rephrase to *"it is too sparse to accurately interpolate"*

 *We have changed as suggested*

L103 If REMA strips (and not the mosaic) are used, then they lack bias and tilt corrections (which could be significant), tide corrections, inverse barometer effects etc. All these corrections are necessary to draw any conclusion on the ice draft etc. Just assuming the strips are correct is methodological not-correct (which can explain many of the later critique on the REMA data) as these raw strips were never provided to be used without the necessary corrections and without them the absolute elevations (and hence drafts) cannot be interpreted

 *We chose to use the strips rather than the mosaic because they span a shorter time scale than the mosaic. However, given your advice, we have changed to the mosaic. As mentioned above, the DEM and ice thickness information from REMA is used very minimally in this manuscript. All analysis that we do for hydrostatic corrections are from our in situ surface GPS and radar thickness data. We do use it to discuss general thickness trends in section 4.1 but this is a large-scale (multiple kilometer) analysis and therefore smaller scale errors are not relevant. As such, any error from using the REMA strips rather than the mosaic would have only been observable in Figure 2 and would not have any impact on the remainder of the manuscript. We realise that we were not clear about this in the manuscript and have adjusted accordingly. In the methodology we now discuss hydrostatic correction more generally than only application to REMA. In the hydrostatic equilibrium results we have clarified to say:*

 *"Ice shelf draft estimations that are extrapolated using hydrostatic balance calculations do not take bridging stresses or pinning points into account and therefore may contain errors. Our in situ ground-based and airborne radar thickness data provide high-resolution data of ice shelf draft features and can be compared with hydrostatically-derived thickness using simultaneously collected surface elevation data."*

 *We have also clarified in the new version of discussion section 5.1 where we state that we are using in situ surface elevation data when we are discussing hydrostatic calculations.*

L105: What was the offset between the strips? How was it corrected?

 *As we have switched to the REMA mosaic as suggested we don't address this.*

L109-114: In my opinion the assumption of pure blue ice is a wrong assumption as the ice shelf locally (especially in the channels) seem to contain snow/firn cover. When this snow/firn is not taken into account any conclusion on bridging stresses is potentially wrong (see also later).

*As we state in the manuscript, firn is present on the ice shelf closer towards the Reeves ice fall and towards Inexpressible Island. However, where we do the hydrostatic analysis at sites 1-3 there is no firn because of the katabatic winds that strip any accumulation. We can confirm this because we ran these radar lines on foot so we were able to directly observe the surface conditions. This can also be seen in the photos in Figure 6. The whiter areas are solid, but bubbly (meteoric) ice, whereas the very blue areas are solid refrozen ice. We discuss this in lines 77-78 where we say:*

*"The katabatic winds that allow persistence of a significant polynya at the terminus of the NIS also strips the ice shelf of much of its snow and firn cover leaving a significant portion of the surface as blue ice"*

*and also in lines 109-111 where we say:*

*"The zone of firn-free blue ice covers the regions of ground-based radar survey but firn is present towards the grounding line of Reeves Glacier and Inexpressible Island, so in these regions the hydrostatic calculations are less accurate."*

Fig.2: perhaps use direct labelling for the green/yellow circle and lines instead of the caption as it requires the reader to unnecessary zigzag between figure and caption.

*As we now label the fracture following your above suggestion, we have removed the yellow circle. The green circle is relevant to aspects that we discuss in the text but would be too long to explain in the caption and so we retain this. We do, however, add that the text description of the green circle can be found in section 5.1 of the manuscript.*

Fig.2: it took me some time to understand that the red-dashed line was moved to overlap draft with meltwater. It could be helpful to indicate that in the figure (e.g. with arrows)

*Thanks for this suggestion. We have made this change.*

Fig.3: the alignment for REMA is potentially problematic if the proper strip corrections are not applied

*As explained above, these are radar lines and hydrostatic thickness from our surface GPS records that we collected during the radar surveys. We do not use REMA here. See above for where we have clarified this in the manuscript.*

L150: what about the spatial and overall accuracy of GOLive data? Small inconsistencies in the velocity data and/or geometric accuracies could have large impacts on the interpretation of the strain rates and I think this should be accounted for in the later analyses. Just assuming the velocity (which may be

a wrong assumption) are correct can result in the observed misrepresentations (see also later comment)

> *While GOLive has inherent error, as does any derived velocity product, the correspondence between the topography and the strain data which follows known ice shelf strain physics gives us confidence that this is a realistic representation and does not impact our analysis. We have discussed a pixel size sensitivity test to determine that the strain signals are not corresponding to one pixel which could cause error. We now include more information about the error that this could create in the magnitude of the strain values. We also point out that these patterns in strain are visible over multiple years and multiple image pairs as we state:*
>
> *"We note that the patterns of strain discussed here are also visible in NIS strain rate maps from multiple years, at different times of the year, and therefore appear to persist over time (e.g. see Fig. 5 in Dow et al, 2018)."*
>
> *In this manuscript we are interpreting only the pattern of strain for our discussion rather than the magnitude and we therefore believe that this is sufficient evidence to support our arguments.*

L160-161 " which produced similar results in both spatial pattern and magnitude of strain rates": what is similar? What are the differences? Perhaps quantify etc.

> *We have now clarified by stating the error in strain produced between the 300m and 600m pixel test:*
>
> *"We ran a sensitivity test with a length scale of 600 m, which produced similar results in the spatial pattern of strain but with the magnitude of strain rates smaller in the suture zone of NIS by ~0.1 day$^{-1}$ or less." These are minor changes in the strain compared to the contrast between extensional and compressional strain in this region and we are primarily interested in the pattern of strain for this study as discussed above.*

Section 3.5: these are very interesting data and should be elaborated further. How was the meltwater classification done?

> *We have now added a temperature-salinity plot into Figure 2 to demonstrate how the meltwater classification was completed for this study. This plot shows practical salinity versus potential temperature for the data presented in Fig. 2c. On this plot, we note the isotherm beneath which water is classified as Ice Shelf Meltwater and reference this in the text where we first discuss the classification.*

Section 4.1: this reads very much as a continuation of the site description and could perhaps be integrated

> *This is the only part of the manuscript where we utilize REMA hydrostatic calculations and we do this to examine larger scale changes on the scale of multiple km. As such, errors inherent in REMA*

*hydrostatic inversions will not impact this analysis. The section however, is discussing larger-scale variations in ice thickness which would not be possible without the hydrostatic calculations. Since we have to introduce how we apply hydrostatic calculations prior to discussion of these results, we therefore retain this section in the results part of the manuscript.*

L175-185: these features are difficult to find and see on the map for a non-experienced reader. Would be a good idea to help the reader and indicate all the described features on the map.

*We have added labels showing parallel and oblique features to Figure 2 to address this.*

L197-199: "*Along the center of the suture zone there is an alternating region of horizontal compression (red) on the northern side and extension (blue) on the southern side: both regions have widths of ~800 m. When compared with the ice shelf draft, the switch between compression and extension occurs at the apex of the thin-ice suture zone region (Fig. 4d)*". I do not agree with these statements as I think it is almost impossible to interpret the strain locations relative to the radar given i) that the strain rate is only calculated every 300m and ii) the potential (spatial) uncertainty in the velocity data. For example, shifting the compression peak 150m to the right (which seems well within the strain uncertainty) would result in a compression peak that is nicely aligned with the apex (which would also make more sense given the discussion later):

*Yes, it is certainly possible that the strain data are not exactly aligned with our radar data, although the excellent correspondence between smaller extensional and compressional regions with the topography (e.g. at -500m and 1.5km) suggest that the alignment is generally very good. We now discuss the potential offset by saying:*

*"When compared with the ice shelf draft, the switch between compression and extension occurs at the apex of the thin-ice suture zone region (Fig. 4d), although with a strain pixel size of 300 m there may be some error with the location of this transition."*

L199-200: "*the switch between compression and extension … is limited horizontally*" This switch is by definition always localized to a point (as there is either compression or extension) so it seems a strange statement

*We agree that this was not a well-phrased statement. We have removed it to avoid confusion.*

L200-206: any of the conclusions based on the location of patterns of strain vs. apex-keels are debatable as these patterns can be easily (mis-)aligned when using small shifts (which seem within the uncertainty of the data). I therefore do not think statements on alignment can be made and the assumption that it can be derived from satellite velocities is potentially overambitious.

*As we include in the discussion, there are physical reasons to back up the observations here along with records at other ice shelves. Namely, surface depressions coincident with thinner ice/higher basal draft tend to have compressional regimes and surface hills coincident with thicker ice/deeper*

*basal draft are extensional (following, for example, Vaughan et al 2012). We also present the strain in 2D (Fig 4b) to demonstrate that the larger scale pattern follows the surface topography (Fig 4a). We have added in a caveat about the pixel size for the transition zone between compression and extension in the thinnest region of the suture zone; here we find that applying a shift in the strain plotting by +- 300m can impact whether this apex is primarily compressional, extensional, or both (see above comment for our text adjustment). However, with an error of +- 300 m the remainder of the strain results that we report along this transect are consistent due to the larger scale variation of the basal draft.*

L224-227: *"The northern side of the suture zone has minimal change in relative basal draft but approaching the apex of the thin ice region, there is substantially more ice loss. The greatest mass loss is in the highest apexes of the central suture zone and the basal fractures on the southern side, with the keels of the latter relatively unchanged compared to the spatially-constant background melt rate."* It might be due to my misunderstanding of the methodology of alignment, but I do not necessarily agree. If you align the draft for the apex (instead of the edges (see example below)), you could conclude that the largest changes occurred at the edges and not at the apex.

*This is an excellent point and we completely agree; thank you for pointing this out. This was a mistake on our part in the application of the relative change calculations. The change could be on the margins compared to the central suture zone and our method did not address this. We have removed the plot of the relative basal draft change from Figure 5b and the related sentences in section 4.3. Instead we now focus on the change in thickness between the sites, which does not demonstrate that the central suture zone has such significant change in shape as we previously argued but instead shows the steeper margins of the suture zone is where the most active thickness change is taking place. Following your suggestion we have further analysed this using the Cryosat-2 basal melt data along with estimations of surface ablation, which supports this new argument (see our more detailed response below). We have also therefore altered our statements about where melt is most likely taking place in the results, the discussion and the conclusion.*

Fig.5. It is very difficult to interpret the relative basal draft (what is it, how was it calculated/quantified)

*See above. We have removed this plot and the related discussion.*

L239-246: the analysis of the melt rates derived from Cryosat-2 is potentially very interesting and should be elaborated on further. What would be the total melt of transect of C-C' was advected (with the velocity) to A-A'? Would this integrated melt show a similar (smoothed) pattern as the simple observed difference between Site1.

*We have performed this analysis as suggested with integration of melt rates between A-A' and C-C'. We then compared the melt over this distance with the thickness change between the two transects at Site 1 and 3. We find that the basal melt from the Cryosat-2 analysis can explain approximately 50% of the thickness change. To investigate the remaining thickness change we run*

*the same cumulative melt integration between these two transects for a surface melt rate of 0.25m/year (following Bromwich and Kurtz (1984), and also 0.5 m/year (following Bell et al, 2017). Adding this surface melt rate to the Cryosat-2 basal melt rate produces a thickness change within a reasonable range of the thickness change from the Site 1 and 3 radar transects. This shows the steeper sides of the suture zone are melting by ~120 m between these two transects, but only by 60 m in the center. The change in ice thickness at the suture zone center is primarily due to surface ablation but as the ice thickens towards the side of the suture zone basal melt from oceanic processes is also occurring. In general, the influence of the basal and surface melting appears to be 50/50. Following from our error with the relative basal draft calculations this new analysis has provided more accurate information about where active melt is taking place. We have adjusted the manuscript accordingly when discussing the role of ocean melt and now suggest that the primary role is on the slopes of the suture zone rather than channelised in the thinnest region of the suture zone. We have also replaced the subfigure in Figure 5b with one showing the integrated basal melt, surface ablation, total melt and measured thickness change. Thank you for this suggestion for additional analysis as we believe it has strengthened the manuscript.*

L249-251: "*Mapping these regions on top of the REMA 2016 hydrostatic thickness map shows they are all associated with thinner regions of ice and, in particular, with basal fractures on both sides of the suture zone along with the thinnest portion of ice in the suture zone*" Where can I see this? It would be useful to replace Fig.6a with this overlay over REMA

*We have changed the wording to 'comparing these regions with the REMA hydrostatic thickness map'. We tried many methods to combine the two datasets for clear visual comparison by the reader but the results were very messy and difficult to analyse without access to the original Matlab plots. This is why we include the surface elevation map of REMA in Figure 4a to allow visual comparison between the elevation (and therefore ice thickness) and our strain plots in 4b and 4c.*

Section 4.5: although the data are very interesting I do find it very difficult to see any conclusion, take home message from this section. Here again it would be beneficial if integrated with the discussion section to remove the fragmentation and increase the impact

*These are results and therefore, at this stage of the manuscript there are no conclusions or take-home messages. In the Discussion we amalgamate different results to discuss the system on a larger scale. We feel this interpretation should be conducted in the Discussion, not the Results section, which is one of the reasons (in addition to The Cryosphere's instructions to authors) why we are retaining our layout.*

Section 4.6: One of the main potential errors in the hydrostatic assumption is that the ice shelf is completely snow/firn free, whereas for example Sentinel-2 data shows that there is snow

deposition/firn over transect A-A' which could provide an explanation for the lack of /muted channels in the hydrostatic REMA approximation. Especially as the snow/firn cover seems stronger in the south where larger offsets in the Dow analysis occur.

*As above, we don't use REMA for hydrostatic analysis in this manuscript. We know that the sites that we do these calculations (Sites 1-3) are snow/firn free because we are using radar and surface data that we collected on foot in these locations.*

*In addition, if there was the presence of firn/snow in the southern region, it would act to reduce the total density of the column, resulting in thinner ice for a given surface elevation. As shown in Figure 3, our offset issue is that measured ice thickness is greater than hydrostatically calculated ice thickness and therefore if firn were present it would exacerbate rather than reduce this offset.*

L296-304: I do not agree with any of this paragraph as (also indicated earlier) i) the REMA strips require several corrections (tides, tilt+offset effects, barometric effect) before they allow to convert to draft ii) the snow/firn could result in local biases as well. Both these forgotten corrections makes the interpretation of the hydrostatic figures very much dependent on potentially wrong assumptions as both offsets+snow could result in similar results. Therefore the conclusion of bridging stresses is not necessarily supported here.

*See above points. We do not use REMA – this was our fault for not being clear in the manuscript but we have now corrected this.*

Section 5.1 reads very much as a continuation of the results of 4.1 and perhaps it should be considered to be integrated.

*See above response. We have also now expanded this section by moving some of the discussion of hydrostatic calculations from the former section 5.5 here.*

L335-339: many of the statements (e.g. alignment, bridging stress) are not necessarily supported by the results (see my earlier comments) and therefore I doubt the correct interpretation of this paragraph

*See our above responses to your comments. We are confident in our results and therefore in this Discussion section.*

L435 " *we find that some of the ice shelf properties are not well represented in the satellite-derived data sets.*": see my earlier comments, but again I do not think it is a fair comment to blame the data. These mis-representations are either the potential result of wrong assumptions (e.g. for hydrostatic balance) or by using the data (e.g. strain rate) without accounting for inherent (spatial) uncertainties that should be accounted for.

*We have deleted this paragraph and section and have moved the discussion to other, more relevant sections.*

L436-451: I do not agree (see my earlier comments)

*See above responses.*

L455 "*If only longitudinal strain had been calculated, these features would have been missed*" Yes, but why would you only calculate longitudinal strain and neglect transverse strain? This is again not a problem of the data, but of a potential wrong assumption (fracture dynamics rely only on longitudinal strain).

*In previous publications (e.g. Lai et al, 2020, as we cite in the manuscript) only the longitudinal strain was used for analysis and here we agree that using transverse strain is useful, and therefore important to point out. We have moved the components of this section to their respective discussions so the focus is no longer on the datasets, but instead on their applicability.*

TECHNICAL COMMENTS

L20 "*Nansen Ice Shelf has a highly variable* morphology"

*Done*

L41-45: "*The increasing variability of … opens up possibilities*"

*Done*

Caption Fig.2 "*The green circle highlights an area referenced in the text*" This is a rather non-helpful caption as I now still don't know what I am looking at (and why?) and requires me to go and search the paper

*See above response to this comment.*

Fig.2 and later figures: I do find it confusing that S1-S2-S3 and A-A', B-B', C-C' are used interchangeably. Would be clearer with consistency throughout all figures etc.

*We retain our labelling because A-A' etc only show one radar transect at each site. As we have multiple radar transects at the sites and may wish to refer to them in future publications we keep the site names and specify individual transects within each site using the capital letters.*

Line 166: difficult for non-experts to see this suture zone. Perhaps direct label the suture zone on Fig.2?

*We have added labels to Figure 2 to direct readers to the location of the surface river, which in the middle of the suture zone. This should clarify the location of the suture zone. We tried also outlining the suture zone on the figure but it became too messy and the river location was the clearest option.*

Fig.4: it would be beneficial if the strain rates were overlaid (semi-transparent in color) over the REMA DEM (e.g. in grey) as it would allow to link the strain to the DEM. Now it is basically impossible to see direct linkages between the different panels.

*See above response to this comment*

Fig.4 colorbar: it would be beneficial if compression and extension is directly labelled on the colorbar as it would make things clearer without the need to read the entire caption.

*We have added 'extension' and 'compression' alongside the colorbar to clarify.*

L194: "*The extent of the region in Fig. 4b is shown in Fig. 2*". Sentence is obsolete in the main text (can be part of caption) and breaks the flow.

*Moved to caption as suggested.*

Fig. 5b: perhaps add transect B-B' (S2) to allow the reader to check for temporal consistency etc?

*We have removed this subfigure (see above responses).*

---

## Referee Report (RR1)

Review of the modified version of **"The complex basal morphology and ice dynamics of Nansen Ice Shelf, East Antarctica"**
by Dow et al., submitted to The Cryosphere

Reviewer: Ala Khazendar

**OVERVIEW**

This modified version includes some improvements. These include the abandonment of blaming certain data sets for how they might be used by investigators, and the recognition that ice-shelf draft changes cannot be attributed solely to submarine melting. A T-S diagram of detected water properties is also a welcome addition.
On the other hand, I still find several issues with the methods and assumptions used to draw several of the conclusions in the manuscript. I describe these in detail below.

**REVIEW in DETAIL**

**Issues related to the fluxgate mass loss calculations**

**L. 245-251:**
In this discussion, the authors are considering "mass loss", yet Equation 2 calculates *volume* flux. How were the volume fluxes at the gates converted into mass fluxes?

**L. 249:**
On a more specific point, this paragraph states that:
"The discharge at Site 1 is 0.226 ± 0.004 km3 a-1 and at Site 3 is 0.222± 0.005 km3 a-1 demonstrating an average cross- sectional mass loss of 181 m2 a-1 as the ice flows ~22 km between these two sites."

The discharge values shown here at the two sites are the same when taking into account the errors. This means that no conclusions can be drawn on cross-sectional volume loss other than that none can be ascertained.

**Issues related to ice-shelf melt rate calculations**

**L. 255-256:**

"This translates to average vertical melt rates of 0.75 m a-1 (keel), 0.45 m a-1 (apex), 0.95 m a-1 (keel), and 0.68 m a-1 (apex) between the two sites."

How does the change in ice thickness between the sites translate into "melt rates"? Again, as I pointed out in the first round of the review, several factors can contribute to

ice-shelf thickness changes and these need to be considered when attempting to isolate the signal of basal melting from ice-thickness changes.

**L. 266-267:**

"It is possible that mass loss is due primarily to surface melt and sublimation, which was estimated by Bromwich and Kurtz (1984) to be 0.25 m a-1 and by Bell et al. (2017) to be 0.5 m a-1."

These numbers are then used later in this section to calculate "cumulative surface mass loss" between sites 1 and 3. This raises two questions:
-   Estimating how surface processes are contributing to ice thickness change cannot only consider loss due to surface melt and sublimation, it also needs to include accumulation (precipitation, wind-blown snow, etc.). Accumulation at the surface does not seem to have been considered here.
-   A "uniform" surface ablation rate was used in the cumulative surface change. There is an underlying assumption here that the values used were also uniform in time, i.e., have not changed over the several decades it takes the ice to advect between sites 3 and 1. How is this assumption justified?

**L. 267-268:**

"To investigate this, we integrate the basal shelf melt rates between Site 1 and Site 3 using the Cryosat-2 basal melt dataset along with ITS_LIVE 2018 ice surface velocities…"

Similar to the remark above: the basal melt and ice surface velocity values used in the integration are assumed here to have not changed over the several decades it takes the ice to advect between sites 3 and 1. How is this assumption justified?

**L. 453-462:**

The issues described above regarding the methods used to calculate melting rates raise doubt on the conclusions in this discussion regarding the different melting rates at the margins and at central part of the suture zone. Those conclusions, nonetheless, seem to be already supported by data from Adusumilli et al. (2020), shown in Fig. 5a of this manuscript. The authors, however, say that the Adusumilli et al. data for Nansen are "close to the noise floor (uncertainty) of the dataset for the NIS". Yet, in other parts of the manuscript they use the same Adusumilli et al. data without any reservation about uncertainty. For example, they compare the satellite-derived basal melt rates with the ocean glider data to find "a good correspondence between the locations of meltwater near Inexpressible Island and the center of the NIS terminus, and the areas of enhanced melt from the ice shelf, L. 273-275."
The authors need to decide how much confidence they have in the Adusumilli et al. (2020) satellite-derived data, and apply that consistently in their analyses. If they

decide that they do have sufficient confidence in those data, then the authors can rely on the satellite-derived data, especially given the problems described above in their fluxgate and point analyses. If they do not have sufficient confidence in the satellite-derived data, then their conclusions regarding the correspondence between glider observations and melting locations under the ice shelf do not hold.

**Issues related to meltwater characterization**

**L. 473-482:**

The authors put emphasis on the depths and locations at which what they call "meltwater" was detected in order to connect it to the basal topography of the ice shelf, but they do not provide any definition of what they consider to be "meltwater". In other words, what are the temperature and salinity ranges of the "meltwater" plotted in Figs. 2b and 2e? The quotes below refer vaguely only to "cold, fresh ice  shelf meltwater" or "supercooled" water.

L. 174-176: "Beneath this layer, the dense High Salinity Shelf Water (potential density > 1028 kg m-3) is presumed to drive basal melt (Rusciano et al., 2013), resulting in the formation of ice shelf meltwater below the *in situ* freezing point (potential temperature < -1.94  C), making the water supercooled (Fig. 2d)."

L. 285-286: "to determine whether the calculated basal melt in the suture zone is observable with *in situ* data, we plot the glider data representing the location of cold, fresh, ice shelf meltwater on Figures 2b, and 5."

**Issues related to the assumption of hydrostatic equilibrium**

**L. 345-347:**

"Our *in situ* data calculations demonstrate that the mismatch between hydrostatic thickness and radar ice thickness is therefore greater closer to the grounding line. This could potentially be due to bridging stresses from the highly variable basal draft and/or from pinning of the ice shelf from valley walls or nunataks."

Or it could be due to the well-established fact that, in the grounding zone, ice surface can lie lower than the hydrostatic equilibrium level over some distance (up to a few kilometers) downstream from the grounding line (Brunt et al., 2011). Work on the Amery ice shelf (Chuter et al., 2015) found that, while there is a mean thickness difference of 3.3% between radio echo sounding measurements and the CryoSat-2-derived thicknesses, that discrepancy rises to 4.7% near the grounding line.

**Other issues**

**L. 421:**

"Alternatively, ocean melt may be focussed on the deeper keels with frazil ice formation possible at the apexes of the basal fractures. The dampening of the amplitude of the basal features as shown in Figure 3, suggests that such differential melting and freezing may be occurring."

Several studies have already observed and modeled this phenomenon, including Khazendar and Jenkins (2003), Jordan et al. (2014), and McGrath et al. (2014). It would be appropriate to cite these previous studies in connection to what is being described here.

**Throughout:**

The Adusumilli et al. (2020) data are invoked frequently in the discussion, but are only explicitly cited once in the main text (L. 259) and another time in the caption of Fig. 5. Otherwise, they are referred to as just "the satellite-derived basal melt rates" or "the Cryosat-2 basal melt dataset." I think there should be more explicit mentions of the source of these satellite-derived melt rates.

**Abstract:**

Related to the preceding point, the Abstract states "We use a combination of airborne and ground-based radar data, satellite-derived data, and oceanographic data collected at the Nansen Ice Shelf…". This can give the incorrect impression that the satellite data are analyzed as part of this study. Again, the source of these data should be made explicit in the Abstract, or at least refer to them as "already published satellite-derived data", or similar.

**Minor Issues, Typos, etc.**

**L. 161:**
"…year-1 or less We interpolated…"
Missing punctuation.

**L. 205:**
"Tison and Khazendar, 2001"
This paper has more than 2 authors, so citation should be "Tison et al., 2001".

**L. 333:**
"…to the terminus At Site 1 and 2, further upstream, …"
Missing punctuation.

**L. 421:**
"focussed"
Typo.

**L. 464:**
"…increase to ~1 ± 0.6 m a-1 Similarly, generally…"
Missing punctuation.

**L. 482:**
"(Friedrichs, in review)"
The reference for this paper is missing from the References list.

**REFERENCES**

Brunt, K. M., Fricker, H. A. & Padman, L (2011). Analysis of ice plains of the Filchner-Ronne Ice Shelf, Antarctica, using ICESat laser altimetry. J. Glaciol. 57, 965–975.

Chuter, S. J., and J. L. Bamber (2015), Antarctic ice shelf thickness from CryoSat-2 radar altimetry, Geophys. Res. Lett., 42, 10,721–10,729, doi:10.1002/ 2015GL066515.

Jordan, J. R., Holland, P. R., Jenkins, A., Piggott, M. D., and Kimura, S. (2014), Modeling ice-ocean interaction in ice-shelf crevasses, *J. Geophys. Res. Oceans*, 119, 995– 1008, doi:10.1002/2013JC009208.

Khazendar, A., and A. Jenkins (2003), A model of marine ice formation within Antarctic ice shelf rifts, J. Geophys. Res., 108(C7), 3235, doi:10.1029/2002JC001673.

McGrath, D., Steffen, K., Holland, P. R., Scambos, T., Rajaram, H., Abdalati, W., and Rignot, E. (2014), The structure and effect of suture zones in the Larsen C Ice Shelf, Antarctica, *J. Geophys. Res. Earth Surf.*, 119, 588– 602, doi:10.1002/2013JF002935.

---

## Author Response (AR2)

**Response to reviewer and editor comments**

We have included our response to the editor and reviewer comments below, indented in italics.

……………………………………………………………………………………………………………………………………………………………………

Thank you for submitting the revisions tc-2021-168. In the meantime, I have received two re-reviews. Both reviewers acknowledge your responses and note corresponding improvements in the revised version. I add to this, that the paper presents a strong observational dataset of airborne and ground-based IPR with much scientific potential.

Unfortunately, a number of critical comments have been raised again and the overall impression of the paper from both reviewers remains weak. I concur with the majority of the concerns raised, and below I also provide some additional thoughts from my own reading. There is a lack of clear take-away messages and some claims made in the paper are not fully substantiated by the data analysis.

> *Thank you for your comments on the manuscript. We have addressed your comments below and have altered the paper accordingly. To first address your concern listed above about the lack of a clear take-away message the primary outputs of the paper are as such:*
>
> *1. In situ radar data provide evidence of highly complex topography at Nansen Ice Shelf.*
> *2. This complex topography is reflected in strain rates observable at the ice surface.*
> *3. The cause of the complex topography is a combination of fracture mechanisms at the suture zone plus active melt in the suture zone.*
> *4. This creates a weak environment, susceptible to fracture.*
>
> *The unique nature of our dataset (particularly the combination of radar and glider ocean data with satellite data) is highly relevant to the Antarctic science community and our findings at Nansen Ice Shelf are important for examining other ice shelves, particularly for those that don't have radar data and rely only on large-scale datasets such as REMA. We have adjusted our introduction and abstract accordingly to make these aims and findings clearer.*

Hence, the manuscript is not ready for publication in The Cryosphere. Substantial revisions are required including data analysis. If you decide to address this in another round at TCD, I will send out the paper for another review including at least one new reviewer. Alternatively you may want to consider to fully restructure the paper with a clearer research focus and make it a new submission. Both options require a significant amount of work, but given the reviews and my own reading I don't see a way around this. I hope that you perceive the remarks as constructive and helpful for improvements.

> *In the response to reviewers below we detail how our plan to substantially revise our manuscript and provide additional data analysis will address your concerns and make our work publication-ready for The Cryosphere. We thank you for this opportunity.*

The comments mentioned below are in addition to the two reviews, they are not intended to replace any of the comments from the reviewers.

Lack of basal radar reflections:
I am not convinced that frazil ice accumulation is the only explanation for the lacking ice-ocean boundary in the radar data. Did you consider the basal slope? In Figure 6d I cannot estimate the slope because the x-label is given in Trace number (should be 'distance') and the y label in traveltime (should be 'depth'). Do you have an example where the lack of signal occurs in both airborne and ground-based IPR? This could be helpful to investigate effects of the different radar systems. (Fig 6a suggests that the lack of basal reflections does not occur in the spatially more extensive airborne dataset. Is this linked to the different radar systems used?).

> *We have adjusted Figure 6d so that it plots distance and depth rather than trace number and traveltime. There is also drop out in the same regions in the airborne data and we have added this to Figure 6a. This is a useful addition because it further demonstrates the link between the stripes of clear blue ice regions and the radar dropout that we discuss in relation to the ground based radar data. Given that these are different radar systems this strengthens our argument that the dropouts are due to frazil ice accumulation rather than slope because of the differences in Fresnel zone. We have also adjusted the caption of Figure 6 accordingly.*
>
> *If we were to do a basal slope analysis we would likely find a correspondence between slope and radar drop out because, as we argue in the paper, the drop-out is due to marine ice accumulation in basal crevasses or full-thickness rifts, which by definition will have steep slopes. Our argument is based on marine ice rather than basal slopes alone because we see the correspondence between ice type (the clear blue ice that can be seen to form, for example, in full ocean rifts at the base of the Reeves Ice Fall) and radar dropout. This correspondence can even be seen at the satellite level as demonstrated in Figure 6. Our new addition of the airborne radar dropout as you suggested will strengthen this argument. Finally, we do note that not all basal reflections with steep slopes have drop-out. For example, see Figure 3a – the feature with the green dot is fully visible but the feature with the red dot drops out. These have similar slopes so the slope alone is not the cause for drop-out.*

Flux gate analysis:
Why not close the gates (i.e. consider the flux into and out of a box)? Given that the flux differences between some gates are small, it seems important to exclude influx/outflux of ice between the two main gates. Neckel et al., 2012, J. Glac. would be an example for this (no need to cite this paper of which I am a co-author).

> *Thank you for this suggestion  - we now use a polygon based 'closed box' flux following the methods of Neckel et al. With this method we see a volume loss of 0.0523 +/- 0.0083 km3/a, or 0.0476 Gt/year in terms of mass loss. We had an error in our flux gate code that meant we were underestimating the mass loss in our previous version of this paper.*

From thickness change to basal/surface melting:
I stumbled across section l. 252f because at first I didn't grasp the meaning of 'vertical melt rates'. I believe it is a lumped term that should include either basal or surface melting, but this was not clear to me at first and others may feel the same way.

> *We have removed this paragraph as stated below. We have changed the wording of 'vertical melting' to be clearer that we mean lumped basal and surface melting.*

I am not convinced that the points marked in Fig. 3 belong to the same draft apexes and keels (i.e. that they are part of one feature). More radar lines would be needed to trace this coherently. Therefore, I am unsure about the inferences drawn starting l 252f.

> *We agree that this is too speculative and have removed from the manuscript.*

Later on, I miss a number of details about how the mass budget was closed. Do you assume steady state and is that justified given current dH/dt estimates for the NIS? Very likely many of the smaller-scale features such as the head of ice-shelf channels or basal fractures are not steady-state features in an Eulerian sense an so I don't quite understand how the analysis works. I was entirely lost at the statement that "..100% of the thickness change in the suture zone center can be attributed to surface ablation processes." (l. 281). How is it possible that strain thinning plays apparently now role in the thickness change for an ice shelf?

> *We have rewritten this paragraph to clarify how we made the calculations (pasted below) including how the mass budget is closed in our analysis. While the basal fractures are not steady features (and therefore we can't assume that the features at site 3 were once the exact same thickness as at site 1) we address this by calculating a smoothed thickness change using a locally weighted least squares regression filter. We have improved this comparison compared to the previous version by smoothing the individual site radar thickness measurements prior to estimating the thickness change rather than smoothing after estimating the thickness change.*
>
> *The point about strain thinning is well taken and we have added this to the analysis, however, the strain thinning is relatively small (on the scale of meters rather than tens of meters), which is consistent with our reported strain rates in Figure 4, and our conclusions do not change (see below). We are following the methodology of Das et al (2020) who use a similar approach and we now direct readers to their equations as well. If clearer we can move the first section below to methods rather than results. We also now acknowledge that there may be change over time of basal and surface melt rates but we have no ability to determine this information (and neither can anyone else who calculates basal melt rates with a spatial analysis unless the ice shelf is changing very rapidly such as at Pine Island). Das et al (2020) for example have the same limitation in their analysis and estimates of thickness change at Ross Ice Shelf and consider regions of flux that take 60 years. This can be compared to our estimate region of consistent melt over 120 years, which is a limitation of the analysis but likely not unreasonable for a cold cavity ice shelf. The primary purpose of this analysis is to examine the relative importance of basal melting, surface melting and strain thinning in the changing thickness of the ice shelf. The figure that we have produced demonstrates that, while thickness change in the central portion of*

*the suture zone could be attributed solely to surface melt, the edges of the suture zone have thinned enough to suggest basal melt is playing a role.*

*We have included a reformatted figure showing the thickness change estimates from all the sources we analyse, now including strain thinning and our improved smoothed thickness change. We have changed the presentation to show thickness change in the negative (more negative = greater thinning), as this is perhaps easier to visualise for the reader. We are happy to revert to the previous presentation if preferred.*

*Our adapted paragraph is below:*

"It is possible that mass loss is due primarily to surface melt and sublimation, which was estimated by Bromwich and Kurtz (1984) to be 0.25 m a[-1] and by Bell et al. (2017) to be 0.5 m a[-1]. To investigate this and where basal melt may be occurring at the ice-ocean interface, we extract stream lines between Site 1 and Site 2 and apply Langrangian methods as described in Das et al (2020) to estimate strain thinning, basal melt and surface melt along those streamlines:

$$\Delta H = \int_{t1}^{t2} -w_b - w_s - H\nabla.\boldsymbol{V} dt$$

where $w_b$ is basal melt, $w_s$ is surface melt, $\nabla V$ is the ice velocity divergence from horizontal, and $t_1$ and $t_2$ are the times that ice passes through Site 1 and Site 3. Cumulative basal melt along the streamlines, $w_b$, is calculated from the Cryosat-2 basal melt dataset (Adusumilli et al., 2020). However, the basal melt map has data up to 2 km away from the Site 3 transect (Fig. 5a) and so cumulative basal melt will be slightly underestimated using this technique. Similarly, we calculate cumulative surface mass loss, $w_s$, between the two sites using a uniform surface ablation rate of 0.25 m a[-1] and 0.5 m a[-1]. Because the region we are examining is composed of exposed blue ice we do not take accumulation into account. Velocity divergence for strain thickness change is calculated from ITS_LIVE 2018 ice surface velocities and flow vectors (Gardner et al., 2020; Fig. 5b). This analysis assumes steady state surface melt, basal melt and ice velocity. The average time for ice to travel between Site 1 and Site 3 is 120 years, which means that some alteration of these rates may have taken place producing errors in our analysis, although with a cold cavity this is likely less of an issue than in regions of the West Antarctic for example (Das et al, 2020). We compare the values of cumulative basal and surface ablation and strain thinning with the IPR-derived thickness change between the sites. Because of the presence of basal features that are not fully aligned between the sites, we first smooth the thickness at each of Site 1 and Site 3 using a locally weighted least squares regression filter before calculating the thickness change.

The average thickness change between Site 1 and Site 3 is 80m (Fig 5b). The cumulative basal melting analysis suggests that, because of initial freezing upstream in the middle of the suture zone (see Fig. 5a), the central region of the suture zone has net zero change in basal melt. However, the steeper margins of the suture zone have sufficient basal melt to account for ~20m of thinning at the ice base between Site 1 and Site 3. If we combine this thinning from basal melt with a uniform surface ablation rate of 0.25 m a[-1] it produces ~50 m less thinning compared to the measured thickness change. However when the thinning from basal melt is combined with a

thinning due to a uniform surface ablation of 0.5 m a$^{-1}$, there is a good correspondence between the estimated cumulative thinning rate and the measured thinning, particularly at the northern margin of the suture zone. Thickness change due to strain thinning is limited to <13m between Site 1 and Site 3. This analysis suggests that basal melting is an important process for thinning on the slopes of the suture zone but less important in the center of the suture zone."

Derivation of the hydrostatic ice thickness:
This has been done many times and I understand the desire to not dwell on this topic for longer than needed. However, given that you compare the hydrostatic ice thickness with IPR thickness and you use it to derive basal melt rate fields, it suggest to get both relative and absolute error estimates. Your estimate of \pm 10 m (l. 114) contains only a \pm 1 error in surface elevation. There are other sources of uncertainties such as:

> *Our application of the hydrostatic thickness calculation is merely to compare with the IPR thickness as a matter of discussion about whether hydrostatic thickness can be misleading in the absence of radar data. We do not use hydrostatic thickness to calculate basal melt rates. Cumulative basal melt rates are calculated using the Adusumilli dataset, which are compared with the change in thickness between site 1 and site 3, both of which have thickness determined from the IPR.*
>
> *The argument about the difference in thickness at site 1 between the hydrostatic and measured ice thickness is a very minor part of the manuscript. Our argument is instead focussed on the lack of highly variable basal topography observable in the site 1 hydrostatic dataset compared to the measured IPR basal topography.*
>
> *As we are unable to correct for tide given the lack of a detailed sub-fortnightly tidal model in this area, and we cannot access mean dynamic topography corrections (see below), we will state in the manuscript that the offset in thickness can be corrected with an additional 2.5 m added to the surface elevation recorded at the radar and that this may be due to tides, mean dynamic topography or pinning and we are unable to extract the exact cause. We do believe that this is a real signal but it has so little impact on the manuscript arguments that we will include the above caveats and it will not be discussed further in the manuscript. We have included the following wording in section 5.2:*
>
> *"At Site 1 and 2, further upstream, there was a vertical offset where the ice thickness measured using the radar was greater, on average, than the ice thickness derived from surface elevation measurements (Fig. 3). This offset can be corrected by raising the surface elevation by an additional 2.5 m and, given that we have not corrected for Mean Dynamic Topography (Griggs and Bamber, 2009) or tidal variation, some of this offset may be explained by these factors. Regardless of the mean thickness offset, the spatial pattern of hydrostatic thickness at Site 1 and 2 also does not accurately reflect the variable topography in this region."*

• What about the uncertainty of ice density? A marginal difference (let's say 915 instead of 917 kg m^(-

3)) already translates into 4 m for a freeboard height of 20 m. a. s. l. . Similar for the ocean water density.

> *Yes, this is true but it is highly unlikely that ice density will change between our Site 1 and Site 3 locations. If we choose a different ice density for Site 1 to better match the ice profiles, the resulting profile won't match in Site 3. This is a case of using our prior knowledge about where hydrostatic balance is most likely to work i.e. as far from pinning points and the grounding line as possible and using this to calculate the best match of ice and ocean density. For the ocean water density, we use glider measurements taken near Site 3. However, as discussed above we are now not discussing the thickness difference between Site 1 and Site 3 in the manuscript.*

• It appears that firn thickness at NIS increases with decreasing distance to the grounding-line. This will result in relative errors which may play into your basal melt rate calculations with similar arguments for the density as mentioned above.

> *As we state several times in the manuscript, the ground-based radar data are recorded on blue ice. The firn thickness may impact errors for a basal melt rate calculated using REMA near the grounding line for example, but since we don't do this and use our more accurate radar data to calculate basal melt rates and surface GPS data at the radar sites to make the hydrostatic calculations this is not an issue.*

• How good approximates the GL04C sea level in this area?

> *In the appendices we perform an error analysis on the cross-overs between the IPR and the airborne radar. The former was corrected to GL04C and the latter was corrected based on measured sea level (as there were flight lines over the ocean). Therefore our error analysis has included possible error with the GL04C.*

• Other studies (e.g., Shean et al., 2019, https://doi.org/10.5194/tc-13-2633-2019; Griggs & Bamber (2009, https://doi.org/10.3189/002214311796905659) consider effects of mean dynamic topography and inverse barometer effects. Is this irrelevant for the NIS?.

> *We attempted to obtain the Mean Dynamic Topography dataset that Griggs and Bamber used but it's no longer available from the datalink provided by Andersen and Knudson (2009) and we cannot find it elsewhere. We had a similar issue obtaining information about inverse barometer effects although it's unlikely to have changed over such a short time period.  See above for our new addition to the discussion addressing this. We believe our cross-over analysis will cover any errors associated with these factors but, as discussed above, the difference in thickness is such a minor part of the manuscript we are willing to pull back from it.*

Derivation & interpretation of strain rates:
Figs 4b,c show a fairly noisy structure (no surprise when deriving strain rates at such high spatial resolution) with some coherent bands in the along- and across-flow directions. The fast-ice/thin ice/sea

ice zone that is clearly evident in the surface elevation (Fig 4a) has surprisingly no correspondence in the strain rate fields. Fig 2c shows masked velocities in this area. Somehow I don't manage to bring this information together. I would feel more comfortable if the strain rate fields would more clearly show some first-order signatures (e.g., compression near inexpressible island) for orientation prior to going into details on scales <500 m.

*We have now included a new figure in the manuscript (new Figure 5) showing transverse strain rates from multiple years with the edge of Drygalski Ice Tongue, Inexpressible Island and the fast ice zone included. We also plot longitudinal strain and shear strain from the 2020 32 day velocity output as there are fewer gaps over the ice shelf compared to 2016. This new figure shows shear and longitudinal compression at Inexpressible Island, a change in longitudinal strain at the boundary between ice shelf and fast ice and consistency in the transverse strain between years.*

*We've also included below a larger view image of the region with longitudinal, shear and transverse strain from 2020 that shows the clear extensional and compressional signals for the Reeves Glacier ice as it passes the grounding line, along with the Drygalski Ice Tongue lateral shear and transverse extension as is begins to pass land-based pinning points. We suggest that the figure pasted above is more appropriate for our discussion but we have included the one below to demonstrate the first-order signatures.*

[Figure]

The statement that when calculating the strain rate fields on a 600 m grid "..produced similar results in both spatial pattern and magnitude of strain rates, but with smaller strain rate magnitudes in the suture zone on the order of ~0.08 [per] year" is confusing. 0.08 per year is almost the maximum values shown in Fig 4b,c. This seems like a substantial difference that should be investigated. How do your results compare to Antarctic wide estimates, e.g., from Alley et al., 2018 (J. Glac.)?

*We mean that the strain pattern persists although the strain rates themselves change (see new Figure 5). We are not interested in the rate itself particularly but the pattern. The size of the*

*features producing the strain rates are around 400-500 m, therefore it is not surprising that larger length scales see less variability between strain pixels. It's essentially a smoothing exercise by increasing our length scale as detailed in Alley et al (2018) as we quote below. We now add in the reference to Alley and remove this statement about length scale tests since, on reflection, it does not add to our discussion. Instead we now present multiple Go-LIVE strain outputs from different years to demonstrate that this pattern we see is constant in time (although not in space as the strain patterns migrate with the ice). This new figure is included in an Appendix. Our strain rate values compare well with those presented in Alley et al. (2018).*

Calculating strain rates on <500 m assumes a very high quality of the surface velocities both in terms of magnitude and direction. Another figure (maybe in the Appendix) that clearly shows that the strain-rates adequately capture ice dynamic signals rather than measurement noise can strengthen your arguments. It might also be worthwhile to check if the smaller scale surface undulations could lead to errors in the satellite-derived velocities which may not have used such highly resolved DEMs (e.g., using InSAR with a more coarsely resolved DEM to correct for the topographic phase).

*Our new figure of transverse strain rates over several years (new Figure 5) shows that the pattern of strain persists.  In terms of our approach, we are following the method of Alley et al (2018) who say: "This commonly used approach utilizes the highest resolution possible given the pixel size and may be appropriate for assessments of small-scale patterns of basal melt rates on ice shelves with complex basal topography. Then basal melt rates were recalculated with strain rates determined using length scales of ~8× the ice thickness. These viscous-scale calculations might be appropriate for large-scale averages and for ice shelves with less complex basal topography". Because Nansen Ice Shelf falls in the category of having complex basal topography we argue that our approach is both standard and valid for this type of analysis. We now cite Alley et al (2018) to justify our approach:*

"Our approach of applying a length scale the same size as the pixel size follows the suggestions of Alley et al (2018) who state that such an approach is appropriate for ice shelves with highly variable basal draft topography."

*Due to this variable topography and the ice flow speed of around 200 m/year (over features around 500 m wide) we are restricted to image pairs over a short time period (such as the 32 day separation of the GoLIVE images). Alternative velocity products (e.g. Measures and ITS LIVE) supply annual velocity outputs which are averaged from larger pixel sizes and therefore only capture the large-scale changes in strain rather than the smaller scale features we observe due to the basal ice draft and hydrostatic balancing. (Note: although there is supposed to be a feature available for ITS-LIVE to choose image pairs it's currently not available and we have been unable to access it.) As we understand it, steep or complex surface topography can impact velocity calculations, however Nansen Ice Shelf is flat with the complex topography on the base of the ice rather than the surface; surface undulations are on the scale of 3 or 4m over a distance of >1km. This small change in surface elevation over distance should not impact the velocity calculations, particularly when the results are consistent between years. Our findings here with extension in keels and compression in troughs fully supports other findings about ice shelf strain*

*characteristics and the physics of ice shelf balancing and strain (e.g. Vaughan et al. (2012)) and therefore we argue that our outputs are both realistic and valid.*

Terminology:
To me the term 'fracture' suggests a basal or surficial crack that emerged because some strain/stress limit was reached. However, some of the features discussed (e.g., the longitudinal features that are directed along-flow) could also have an oceanographic origin. Maybe consider first using a more general term such as 'basal undulations' and then more specific terms once you circled in on a possible formation mechanism.

*We have changed as suggested.*

Editorial remarks:

I don't see how eq (3) is actually used. Vertical strain rates are not shown. Also, is it necessary to introduce a variable (\varepsilon_1) that is always zero?

*We have removed this equation.*

L 185: Wrong internal reference? Fig 1c does not show the slope.

*The sentence states "At the terminus, the northern flank of the thin-ice region is wider (6 km) than the southern flank (4 km) with an average ice draft slope of ~1.3˚ and ~1.1˚ at the ice shelf terminus, respectively". Fig 1c shows the ice draft across a transect at the terminus and our statement therefore stands. This includes distance and depth of the ice draft from which the slope can be estimated but we don't believe we need to physically add a slope line to this Figure as it can be computed by the data presented.*

L 300: What does gain of 0.5 mean ? There are many different types of gains and I don't see how a single number would characterize this.

*We have changed to: "a gain of $t^{0.5}$ was applied to the amplitude of each trace, where t represents time".*

Fig 4d y label should be strain rate (not strain)

*We have made this change.*

Fig 5 maybe a red-blue color scale with white at 0? It is difficult to differentiate between melting and refreezing which should be a first-order outcome of this Figure.

*We have changed the color scale to red-blue.*

Fig 6 trace number should be distance, y label depth.

*We have made this change.*

Figure 6 b/c should detail viewing direction relative to ice-flow direction.

*We have added arrows for ice flow direction.*

(A1) should have a v^2

*Thanks for pointing this out. We missed that and changed the equation.*

**Reviewer 1**

L. 245-251:
In this discussion, the authors are considering "mass loss", yet Equation 2 calculates *volume* flux. How were the volume fluxes at the gates converted into mass fluxes?

*Yes that is correct, we were only calculating volume loss rather than mass loss. We now use a polygon flux gate method as suggested by the editor and convert to mass in Gt.*

L. 249:
On a more specific point, this paragraph states that:
"The discharge at Site 1 is $0.226 \pm 0.004$ km3 a-1 and at Site 3 is $0.222 \pm 0.005$ km3 a- 1 demonstrating an average cross- sectional mass loss of 181 m2 a-1 as the ice flows ~22 km between these two sites."
The discharge values shown here at the two sites are the same when taking into account the errors. This means that no conclusions can be drawn on cross-sectional volume loss other than that none can be ascertained.

*We have changed our method for flux gate calculations, now using a polygon method examining flux in and out of the channel margins in addition to S1 and S3. With our error analysis included in these calculations we obtain a volume loss of $0.0523 +/- 0.0083$ $km^3/a$. We made an error in the last iteration with the flux gate calculations so the higher mass loss rate we report here is more accurate and beyond the error margin.*

Issues related to ice-shelf melt rate calculations

L. 255-256:

"This translates to average vertical melt rates of 0.75 m a-1 (keel), 0.45 m a-1 (apex), 0.95 m a-1 (keel), and 0.68 m a-1 (apex) between the two sites." How does the change in ice thickness between the sites translate into "melt rates"? Again, as I pointed out in the first round of the review, several factors can contribute to ice-shelf thickness changes and these need to be considered when attempting to isolate the signal of basal melting from ice-thickness changes.

*Following these comments and those from the AE, we have removed this sentence and our comparison of keels and apexes.*

L. 266-267:
"It is possible that mass loss is due primarily to surface melt and sublimation, which was estimated by Bromwich and Kurtz (1984) to be 0.25 m a-1 and by Bell et al. (2017) to be 0.5 m a-1." These numbers are then used later in this section to calculate "cumulative surface mass loss" between sites 1 and 3. This raises two questions:

- Estimating how surface processes are contributing to ice thickness change cannot only consider loss due to surface melt and sublimation, it also needs to include accumulation (precipitation, wind-blown snow, etc.). Accumulation at the surface does not seem to have been considered here.

*We do not have information on these processes at Nansen Ice Shelf but, as we are operating in a region of blue ice we can assume that wind-blown snow is not an issue and it is likely precipitation is not either. We now state this in the text: "Because the region we are examining is composed of exposed blue ice we do not take accumulation into account".*

- A "uniform" surface ablation rate was used in the cumulative surface change. There is an underlying assumption here that the values used were also uniform in time, i.e., have not changed over the several decades it takes the ice to advect between sites 3 and 1. How is this assumption justified?

*It is beyond the scope of this study to establish past surface ablation. We have added the following in the manuscript to point out the limitation of our analysis:*

"This analysis assumes steady state surface melt, basal melt and ice velocity. The average time for ice to travel between Site 1 and Site 3 is 120 years, which means that some alteration of these rates may have taken place producing errors in our analysis, although with a cold cavity this is likely less of an issue than in regions of the West Antarctic for example (Das et al, 2020)."

L. 267-268:
"To investigate this, we integrate the basal shelf melt rates between Site 1 and Site 3 using the Cryosat-2 basal melt dataset along with ITS_LIVE 2018 ice surface velocities…"
Similar to the remark above: the basal melt and ice surface velocity values used in the integration are assumed here to have not changed over the several decades it takes the ice to advect between sites 3 and 1. How is this assumption justified?

*The same argument can be made of anyone looking at ice shelf changes or indeed any change over time with limited data (e.g. Das et al, 2020, Lai et al, 2020). We don't have past ice velocities of Nansen Ice Shelf and therefore we have to make the assumption that these velocities are representative. Without this, very little temporal analysis would be possible in*

*regions like the Antarctic beyond the satellite record. As above, we now acknowledge that these values may have changed over time and that, with a cold cavity, decadal change is less likely an issue to our analysis compared to warm-based cavities in the West Antarctic for example.*

L. 453-462:
The issues described above regarding the methods used to calculate melting rates raise doubt on the conclusions in this discussion regarding the different melting rates at the margins and at central part of the suture zone. Those conclusions, nonetheless, seem to be already supported by data from Adusumilli et al. (2020), shown in Fig. 5a of this manuscript. The authors, however, say that the Adusumilli et al. data for Nansen are "close to the noise floor (uncertainty) of the dataset for the NIS". Yet, in other parts of the manuscript they use the same Adusumilli et al. data without any reservation about uncertainty. For example, they compare the satellite-derived basal melt rates with the ocean glider data to find "a good correspondence between the locations of meltwater near Inexpressible Island and the center of the NIS terminus, and the areas of enhanced melt from the ice shelf, L. 273-275."

The authors need to decide how much confidence they have in the Adusumilli et al. (2020) satellite-derived data, and apply that consistently in their analyses. If they decide that they do have sufficient confidence in those data, then the authors can rely on the satellite-derived data, especially given the problems described above in their fluxgate and point analyses. If they do not have sufficient confidence in the satellitederived data, then their conclusions regarding the correspondence between glider observations and melting locations under the ice shelf do not hold.

*We have provided the following in this manuscript:*

*- Evidence from high resolution ground based radar surveys of ice mass loss*
*- Evidence from in situ ocean based records of ice mass loss*
*- Evidence from satellite based data of ice mass loss.*

*This quantity of data is more than many studies are able to accumulate particularly for such an understudied region. We acknowledge in the manuscript that each of these methods has their own limitations yet the fact that they are independent methods and all point towards to the same answer gives us confidence in our arguments. Glaciology and Antarctic investigations in particular are inherently limited by the data that is available both in terms of time span and spatial sampling, and we are being very open about the limitations of the methods that we are presenting. We therefore argue that our statements and approach are fully valid and useful for the scientific community.*

Issues related to meltwater characterization

L. 473-482:
The authors put emphasis on the depths and locations at which what they call "meltwater" was detected in order to connect it to the basal topography of the ice shelf, but they do not provide any definition of what they consider to be "meltwater". In other words, what are the temperature and salinity ranges of the "meltwater" plotted in Figs. 2b and 2e? The quotes below refer vaguely only to "cold, fresh ice shelf meltwater" or "supercooled" water.

("L. 174-176: "Beneath this layer, the dense High Salinity Shelf Water (potential density > 1028 kg m-3) is presumed to drive basal melt (Rusciano et al., 2013), resulting in the formation of ice shelf meltwater below the *in situ* freezing point (potential temperature < -1.94▢C), making the water supercooled (Fig. 2d)."

"L. 285-286: "to determine whether the calculated basal melt in the suture zone is observable with *in situ* data, we plot the glider data representing the location of cold, fresh, ice shelf meltwater on Figures 2b, and 5." )

> *For clarification we are changing our wording from meltwater to Ice Shelf Water and include additional details in the text:*
>
> *Lines 175-177: "... is presumed to drive basal melt (Rusciano et al., 2013), resulting in the formation of Ice Shelf Water colder than the surface freezing point (potential temperature < -1.94 degC, Fig. 2d). We therefore take the presence of this cold water as evidence of ice shelf melt."*
>
> *Line 286: "... data representing the location of Ice Shelf Water (ISW, identified by potential temperature < -1.94 degC) on Figures 2b and 5. This ISW was distributed..."*

Issues related to the assumption of hydrostatic equilibrium

L. 345-347:
"Our *in situ* data calculations demonstrate that the mismatch between hydrostatic thickness and radar ice thickness is therefore greater closer to the grounding line. This could potentially be due to bridging stresses from the highly variable basal draft and/or from pinning of the ice shelf from valley walls or nunataks."

Or it could be due to the well-established fact that, in the grounding zone, ice surface can lie lower than the hydrostatic equilibrium level over some distance (up to a few kilometers) downstream from the grounding line (Brunt et al., 2011). Work on the Amery ice shelf (Chuter et al., 2015) found that, while there is a mean thickness difference of 3.3% between radio echo sounding measurements and the CryoSat-2-derived thicknesses, that discrepancy rises to 4.7% near the grounding line.

> *We did consider this effect but, in this location we are 25 km from the grounding line and we had assumed that was too far to feel the effects. However, as the AE also brought up the issue of various factors that we cannot adjust for (Mean Dynamic Topography etc.) we have pulled this statement out of the manuscript. Our primary interest is that the surface topography does not reflect the large complexity of ice draft topography and that argument stands regardless of whether there is an offset or not. We have added the following to the manuscript:*
>
> "At Site 1 and 2, further upstream, there was a vertical offset where the ice thickness measured using the radar was greater, on average, than the ice thickness derived from surface elevation

measurements (Fig. 3). This offset can be corrected by raising the surface elevation by an additional 2.5 m and, given that we have not corrected for Mean Dynamic Topography (Griggs and Bamber, 2009) or tidal variation, some of this offset may be explained by these factors. Regardless of the mean thickness offset, the spatial pattern of hydrostatic thickness at Site 1 and 2 also does not accurately reflect the variable topography in this region."

Other issues
L. 421:
"Alternatively, ocean melt may be focussed on the deeper keels with frazil ice formation possible at the apexes of the basal fractures. The dampening of the amplitude of the basal features as shown in Figure 3, suggests that such differential melting and freezing may be occurring."

Several studies have already observed and modeled this phenomenon, including Khazendar and Jenkins (2003), Jordan et al. (2014), and McGrath et al. (2014). It would be appropriate to cite these previous studies in connection to what is being described here.

*We have added these citations.*

Throughout:
The Adusumilli et al. (2020) data are invoked frequently in the discussion, but are only explicitly cited once in the main text (L. 259) and another time in the caption of Fig. 5. Otherwise, they are referred to as just "the satellite-derived basal melt rates" or "the Cryosat-2 basal melt dataset." I think there should be more explicit mentions of the source of these satellite-derived melt rates.

*We have added more citations for the Adusimilli dataset although often once datasets have been introduced in manuscripts (e.g. Measures, REMA) they are not re-cited.*

Abstract:
Related to the preceding point, the Abstract states "We use a combination of airborne and ground-based radar data, satellite-derived data, and oceanographic data collected at the Nansen Ice Shelf...". This can give the incorrect impression that the satellite data are analyzed as part of this study. Again, the source of these data should be made explicit in the Abstract, or at least refer to them as "already published satellite-derived data", or similar.

*We have made this change.*

Minor Issues, Typos, etc.
L. 161:
"...year-1 or less We interpolated..."
Missing punctuation.

*We have made this change.*

L. 205:
"Tison and Khazendar, 2001"
This paper has more than 2 authors, so citation should be "Tison et al., 2001".

*We have made this change.*

L. 333:
"…to the terminus At Site 1 and 2, further upstream, …"
Missing punctuation.

> *We have made this change.*

**Reviewer 2**

Dow et. al. have improved the paper by implementing many of the suggested changes and removing the problematic interpretation of the some of the satellite data.

Nevertheless and despite of some important rephrasing being done, I think the paper still contains many parts that might indicate over-interpretation of the data or conclusions that are not supported by data. I explain my concerns in more detail below, but think that all claims related to hydrostatic balance, bridging effects and basal melt rates need to be downscaled.

• Dow et al. see discrepancies between the ice thickness from radar and the hydrostatic ice thickness from GPS especially for A-A' close to the grounding line and they relate this to bridging stress. I do not agree with this. Bridging stresses could indeed play a role for the basal structures that are narrow and tall, but cannot explain the overall offset for A-A', where the GPS hydrostatic balance seems over whole transect is almost ~10-20m off. If there is such large overall offset, I think something is wrong with the comparison anyway. A possible explanation (also indicated by Reviewer #1 and my limited understanding of marine ice radar) could be an error in the ice radar interpretation as the marine basal radar responses are potentially missing in the radar ice thickness. If the thickness measurements are based on the meteoric–marine ice interface, which could be the case in a suture zone with refreezing, it would mean the ice radar thickness measurements are underestimated. Closer to the calving front this effect this effect is as melting again dominates. In any case, based on the large offset between ice thickness from radar and the hydrostatic ice thickness from GPS, I don't think it is fair to focus only on the bridging as potential explanation and draw conclusion from it (L346-360). In my opinion the mismatch between the ice thickness from radar and the hydrostatic ice thickness from GPS is too large to draw any meaningful conclusion without investigating why it happened.

> *Firstly we agree that the ice thickness measurements may be underestimated if marine ice is present. However, if there is thicker ice present at Site 1 (due to marine ice accumulation) than we measure by the radar that increases the offset between radar and hydrostatic ice thickness rather than decreases it. However, as the AE also brought up the issue of various factors that we cannot adjust for (Mean Dynamic Topography etc.) we have pulled this statement out of the manuscript. Our primary interest is that the surface topography does not reflect the large complexity of ice draft topography and that argument stands regardless of whether there is an offset or not. We have added the following to the manuscript:*

*"At Site 1 and 2, further upstream, there was a vertical offset where the ice thickness measured using the radar was greater, on average, than the ice thickness derived from surface elevation measurements (Fig. 3). This offset can be corrected by raising the surface elevation by an additional 2.5m and, given that we have not corrected for Mean Dynamic Topography (Griggs and Bamber, 2009) or tidal variation, some of this offset may be explained by these factors. Regardless of the mean thickness offset, the spatial pattern of hydrostatic thickness at Site 1 and 2 also does not accurately reflect the variable topography in this region."*

• Dow et al still make strong claims about basal melt rates without correcting for con-divergence of the ice or variations in melt over time. Therefore all claims (e.g. L255-265 + discussion) related to melt on specific locations (or deriving the maxima) is just hand-waiving and needs to be corrected for possible confounding factors (e.g. con-divergence) and the results and discussion should be adapted accordingly.

*We now take advection and divergence of the ice into account by applying a polygon approach to the mass flux as suggested by the AE. We also now include strain thinning in our analysis of cumulative thinning between Site 1 and Site 3. In terms of variations in basal and surface melt over time, this is not something anyone can reliably calculate for melt rates in Antarctica beyond the last decade or so of available satellite data and therefore any prior work discussing basal melt over long time periods will have the same issues (e.g. Das et al, 2020). This does not negate the usefulness of the results that we present in this manuscript and we stand by our analysis. We have added the following to the manuscript to address this:*

"This analysis assumes steady state surface melt, basal melt and ice velocity. The average time for ice to travel between Site 1 and Site 3 is 120 years, which means that some alteration of these rates may have taken place producing errors in our analysis, although with a cold cavity this is likely less of an issue than in regions of the West Antarctic for example (Das et al, 2020)."

*However, we acknowledge that examining the change in keel depth is overstepping in the analysis due to possible differences in initial thickness and depth of basal fractures and have removed this. Our analysis of thinning drivers between Site 1 and Site 3 are compared with smoothed ice thickness values from Site 1 and Site 3 to avoid the issue of directly comparing different keels and apexes.*

• Dow et al the extrapolate the Cryosat-2 melt over 123 years and add arbitrary surface melt/sublimation numbers to it to explain the thickness changes between S1 and S3. There is however so much that might have happened in 123 years that could also have affected S1 and S3, but that they don't take into account (e.g. temporal variations in basal melt rate). I therefore thing many of the conclusions drawn here are over interpreted and not supported by the data.

*Melt and sublimation rates were produced by other scientific studies with robust methods, we have cited them accordingly and have no reason to reject the only available data on this key process. We performed this analysis and extrapolation after you suggested in it round 1 of*

*reviews and we appreciated the input. See above for our addition into the manuscript to acknowledge that melt rates may have changed over time. Beyond this we are limited by the information that we can gather today.*

• (related to previous two comments) In their response to Reviewer #1 (who asked for corrections based on divergence, advection, SMB, firn air), they rule out firn, SMB and river erosion, but don't tackle divergence or advection which both could have a significant effect on the interpretation. The flux gate analysis accounts for con- and divergence, but the rest of the spatial comparisons does not in my humble opinion.

*We have now taken account of divergence and advection by applying a polygon flux gate method and strain thinning analysis and find little ice gain or loss from divergence and advection in the region examined between our radar sites (see above addition into the manuscript).*

**References**

Das, I., Padman, L., Bell, R.E., Fricker, H.A., Tinto, K.J., Hulbe, C.L., Siddoway, C.S., Dhakal, T., Frearson, N.P., Mosbeux, C. and Cordero, S.I., 2020. Multidecadal basal melt rates and structure of the Ross Ice Shelf, Antarctica, using airborne ice penetrating radar. *Journal of Geophysical Research: Earth Surface*, *125*(3), p.e2019JF005241.

Lai, C.Y., Kingslake, J., Wearing, M.G., Chen, P.H.C., Gentine, P., Li, H., Spergel, J.J. and van Wessem, J.M., 2020. Vulnerability of Antarctica's ice shelves to meltwater-driven fracture. *Nature*, *584*(7822), pp.574-578.

---

## Author Response (AR3)

Dear Authors,

let me apologize for the long time that it has taken to get back to me. One reason was that I couldn't easily find new reviewers. The old ones declined to continue with the review, in parts because they found that initial revisions have not improved the paper to the degree required. Because of this I decided for one new reviewer and to complement this review with an editor review. This took me much longer than expected (mostly due to teaching obligations with field mapping courses). Please accept my apologies for the delay.

Unfortunately, the evaluation of the new reviewer is weak, which means that we now have five external evaluations which have raised substantial criticism. A number of points have improved (i.e. the strain rates are more convincingly displayed, and the mass budget between Site 1 and Site 3 is now closed), but take away messages still remain in parts inconclusive.

I understand that this is an unsatisfactory result, and below I mention some points which are hopefully constructive and helpful. If you decide to revise the paper, I will have to send it out to re-review again. We need at least one positive external review to move on to TC.

Kind regards, Reinhard Drews

*Thank you for your comments on the paper and for the effort to find reviewers. It is unfortunate that the new reviewer didn't like the paper but we believe that was primarily a misunderstanding of what our manuscript was discussing. They were mostly interested in the calving event and fracturing at Nansen, which we agree is a good area for discussion but we published on this already (Dow et al, 2018) and so we don't wish to repeat information from that study here. We have added in a paragraph to the introduction and at the end of our results sections to provide clearer takeaways from the manuscript. Furthermore we have rearranged the 'methods' into 'datasets' and moved some of the analysis into the results which we believe streamlines the manuscript. Finally, we have moved several subfigures to the Appendix so that the remaining figures better clarify the primary outputs of the manuscript. We appreciate the opportunity to address these comments. Note that line numbers are in reference to the tracked changes version.*

EC1: Derivation of basal melt rates from discharge
* * *
The comments from previous reviewers and myself have been incorporated, and the mass budget between Site 1 and Site 3 is now closed. However, I disagree with multiple statements in the paper that basal melt in the suture zone was calculated from in-situ data (e.g. l. 295; l. 447; l. 483). It wasn't. Analysis around l. 268f explains the observed thickness change with strain thinning, basal melting and surface melting. But it doesn't "derive" basal melting per se. Instead it uses existing basal melt rate estimates (Adsumuili et al.) and puts those into an ice dynamic context. Derived are maybe the surface melting rates, b/c 0.5 m/a explain the thickness change better than 0.25 m/a.

The glider data may support existing basal melt rate estimates, but they also don't derive them. Language according to this should be changed throughout the manuscript.

> *We have gone through the manuscript and changed the language as suggested both for the basal melt calculations and the glider data.*

EC2: Interpretation of enhanced melting at the flanks vs keel.
* * *
Section 5.5. suggests that the Adsumuili data indicates that basal melting is in some parts stronger at the flanks than at the keels. This point is interesting, but not rigorously evaluated. If melting is stronger at the flanks, then the zone should widen downstream. If the Coriolis force leads to stronger melting at one flank, then the suture zone should deviate from flowlines (cf. Drews et al., 2021 -- no need to cite this paper). Is this the case? Without this kind of analysis, degree of novelty of these findings is limited as they have been mentioned in the previous studies cited in the paper.

> *Thank you for this suggestion. To demonstrate this more rigorously we have now included flowlines from a transect across the initial suture zone which demonstrates that the suture zone is widening more on one side than the other, strengthening our argument that Coriolis is involved. These flowlines are now included in Appendix Figure C1b and explained in the manuscript. We hope this addresses your above comment.*

EC3: Hydrostatic equilibrium.
* * *
This point is technically well done, but it takes up a lot of space in section 5.2 for what is known already. It is not new that narrow surface expressions in REMA are not well represented in the basal morphology. The arguably new point (i.e. also features larger than one ice thickness show signs of bridging, l.368) appears buried. I suggest to significantly shorten this section (e.g., in order to provide more space for EC2) and to highlight this new point more clearly.

> *We have significantly shortened this section by amalgamating it with section 5.1 and emphasised the point about features larger than one ice thickness. This is also included in the conclusion. The reviewer asks for inclusion of REMA hydrostatic thickness analysis in this section but, following your points here we haven't added this in order to keep this section short (see below for our response to their comments and sample figures).*

EC4: Integration of glider data
* * *
I now well understand that you argue that the glider data may pick up zones of localized ice-shelf melt. However, I don't find this convincing. Figure 7a detected ice-shelf water in most areas apart from two more localized (white) zones where no such water was detected. Do you suggest that that the higher basal melt rates in the suture zone (up until the ice-shelf center) are somehow flushed out localized at

the front? If so, could you indicated the pathways on the Figure?

> *Unfortunately at the Nansen Ice Shelf front there is a large eddy (see Friedrichs et al, 2022 Fig4) so from the directionality data we can't confirm where exactly the fresh water is coming from. This is very unsatisfying so all we can really say is that there is definitely freshwater (and therefore melt) occurring at this cold cavity ice shelf. We have gone through the manuscript to clarify this and pull back on assumptions about where that melt came from. It seem a shame not to report the evidence of ISW from within the cavity since so few datasets like these exist, particularly for cold-cavity ice shelves, so we have retained it in the Appendix. We have added the following to the manuscript in Appendix B:*

> *"However, the presence of an eddy at the terminus of Nansen Ice Shelf (Friedrichs et al, 2022) means that we cannot use directional data from the glider to determine whether the suture zone is a greater source of ISW than other regions of the sub-shelf cavity. Instead the data merely indicates that ice shelf melt is occurring at this cold cavity ice shelf."*

Also, basal melting 15 km upstream from the ice-shelf front is absent and the freeze-on rates appear unrelated to the suture zone. How would that change the presence/absence of ISW in the glider data?

> *The basal freezing upstream implies that the recorded ISW is locally from the front 15 km of the shelf. We've removed our paragraph speculating about the source of the ISW aside from saying it's from the NIS and so we won't comment on this in the manuscript.*

EC5: Detection of marine ice
* * *
From what I understand (I may not have) you suggest that the suture zone has some fractures which go through the entire ice shelf, and which are filled with marine ice. This would be a nice new finding for me. However the radar data analysis is not fully clear in that regard. It is correct that marine ice attenuates radar waves rapidly, but then you would still expect a marine--meteoric ice interface in the radar data (cf. Kulessa et al. -- research at the Larsen Ice Shelf). This doesn't seem to be the case. Is this because the marine ice structures are vertical and not horizontal? If so, how would this imprint on the radio-wave propagation? Do you see any internal signatures in the zones where the basal returns drop out, if so, how do they related to your hypothesis of vertical marine ice?

> *Yes we are suggesting that this is the case. We observe 'echo-free zones' on the NIS in regions within the suture zone and elsewhere where rifts may be found. In contrast to reports from other ice shelves (e.g. Larsen, Amery, etc) where a reflector is observed at depth and the presence of marine ice is inferred to be below this, we find no internal reflectors in the ice below the ringing of the air/ground wave. This we interpret to be former rifts or crevasses filled with marine ice and we assume that the marine ice body is vertical (but may have deformed from vertical by strain over time). To make this finding clearer, we have added more text and improved our figure 5 to show where the echo-free zone occurs in two transverse radar lines across the suture zone.*

*The text we have added to explain these features is in lines 235-251:*

*Given the abundance of clear ice base reflectors in the remainder of the radar data, it is likely that these "echo-free zones" represent areas of marine ice  and/or frazil ice accumulation (Holland et al., 2009). Typically, the unconformity between meteoric ice and the subjacent horizontal layer of accumulating marine ice will reflect radar waves allowing this boundary to be identified (Fricker et al., 2001; Kulessa et al. 2014).  Since marine ice is a lossy medium (Tulaczyk and Foley, 2020), no radio waves are returned below this depth and marine ice thickness can only be inferred from hydrostatic calculations.  In our radargrams, there are regions where little to no signal is returned, except the signal associated with the air/ground wave near the surface (Fig 5e and f).  This complete absorption of all the radar energy in these locations can only be explained by marine ice near the upper surface of the ice shelf.  A healed rift or very tall/high basal crevasse filled with marine ice, would represent a vertical section where radar waves are absorbed (c.f., Hillebrand et al. 2021).  On the ice surface, in the suture zone, there are many stripes of white and clear blue ice that are associated with echo-free zones (Fig. 6c). These can be traced back to the Reeves Glacier ice fall where crevasses fully fracture through the ice column, fill with sea water and refreeze (e.g., Khazendar et al., 2001, Tison et al, 2001). The filled rifts are then advected and stretched, producing visible stripes parallel to the ice flow direction along the suture zone. Khazendar et al. (2001) postulate that ice folding due to lateral compression may alter ice stratigraphy and dip angle.  These processes may contribute to the blue/white striping seen in the suture zone and also imply that whatever ice is visible at the surface is not necessarily the same as the ice beneath.*

EC6: Features in the transverse strain rates in the suture zone
* * *
The elongated red-blue stripes (transverse compression and extension) in the suture zone are a nice signal in the strain rate maps. I believe one explanation for this could be linked to secondary flow of ice into from areas of thick ice into areas of thin ice although, then we would need two zones for transverse extension which are not observed. Nevertheless, I suggest to put this observation into the context of the study from Wearing et al. (GRL, Secondary flow of ice shelf channels).

> *Thank you for this suggestion. We have added the following to the discussion lines 493-495:*
>
> *"Alternatively, this strain pattern may be indicative of secondary ice flow into the thinnest region of the suture zone, particularly if there is basal melt occurring in this region (Wearing et al, 2022)."*

Kind regards, Reinhard Drews

Minor Comments:
1. There are no results pertaining to the glider data in the abstract. This is another sign that the glider

data is not yet sufficiently connected to the analysis on the ice shelf

*We have now moved the glider data to an appendix - we think it's important to demonstrate ISW is present under this cold cavity ice shelf (not a given considering the cold local ocean conditions) so it's worth stating but we agree it is not a central part of the manuscript.*

l 58 What is "active basal melt"? Is there something like "passive basal melt"? Do you maybe mean "localized basal melt"?

*Yes thank you, this is what me meant.*

l 133 "an \rho_i is the f" ?

*Strange…corrected to 'the density of ice'*

l 140 ".,"

*Sorry – not sure what this refers to.*

l 149 "plotted in Euro? " ?

*Apparently word autocorrects (e) to € …fixed now.*
l. 152 another Euro

Fixed.

Figure 5: I don't get the caption (b) and (c). If both show longitudinal strain rate they should look the same. I guess (c) refers to the horizontal shear strain rates as suggested by the title of that subfigure?

*Yes, (c) should have referred to Strain. Corrected now.*

Equation 2: I understand what is done here, but the notation is sub-optimal. "i" is both an index and a distance. Maybe better use dl_i or something like this to indicate that you multiply with a length

*Changed to dl*

l. 160 "REMA hydrostatic product" better "we use the hydrostatic ice thickness derived in section 3.2" (otherwise it sounds as if the thickness product is delivered by the REMA team.)

*Changed as suggested*

l. 176 Instead of "high-resolution" better quantify what the resolution is in meters.

*Changed as suggested*

l. 180 Which are the transverse and which are the along-flow directions? I believe you define this all locally, meaning that transverse and along-flow are defined with the local velocity vector. A few words will make this clear.

*Yes, we have changed to clarify: "We calculated the principal strain in the longitudinal and transverse directions, having rotated to the direction of local flow"*

l 313 I find it uncommon to mention processing steps of the radar data in the figure caption.

*These have been removed*

In general: Always be specific which component of the strain rate tensor is shown,e. g., is the profile in 4d transverse or longitudinal? The y-label is unspecific.

*We have corrected this in Figure 4d and checked the manuscript for other occurrences of this.*

l 267 Much of this paragraph sounds like as if it belongs into the methods, but I don't insist on this.

*We have rearranged the manuscript to better focus on the data collected vs. data other people produced vs. calculations we applied to those data. As a result we have changed the 'methods' section to 'Datasets' and the 'results' section to 'results and analysis'. While this is a less conventional approach we believe it highlights our work and findings better than our previous configuration.*

l 273 v should be bold, also the divergence operator is missing, it is written as a gradient.

*Thanks for catching. Corrected.*

The equation near l. 273 is unnumbered. Also, in essence, this integral is an integral over space rather than time (because no datasets are available for t_1 and t_2). Wouldn't it be easier to mark this accordingly?

*We use the velocity products to estimate the times for flow between site 1 and site 3 to extract dt and therefore also t1 and t2. We believe it's standard to have the flux integrated between two times to represent the flux speed – in this case we're following Das et al (2020) who we cite in this section.*

l 284 Can you state what impact the smoothing has (i.e. compare the results for smoothed and unsmoothed).

*In a previous version of the manuscript, we used the unsmoothed transect data to perform this analysis.  Reviewers pointed out that this could lead to misleading results since, due to spreading, the apexes and troughs of the basal features might not align.  We agreed with this feedback and decided to, instead, focus on a much broader-scale changes within the suture zone.  This led us to remove the high frequency thickness changes along the transects to examine the suture's basal melt.  We are confident that this more general examination of the suture zone shape will avoid artifacts where features which don't necessarily align from one site to the other (different locations/sizes) are subtracted from each other.  To compare our current version to the unsmoothed version, please refer to Figure 5b in the second version of the manuscript.*

l 304 Can you differntiate more? There is no active (do you mean localized?) melt in the suture zone near the ice shelf front. How would that imprint on the distributiion of ISW?

*Following your suggestion we've decided to pull back on what we can say about the source of the ISW due to the presence of the sub meso-scale eddy (so we can't pull out non eddy-related flow directions from the data). We've added the following to this section:*

*"However, the presence of a sub-mesoscale eddy at the terminus of Nansen Ice Shelf (Friedrichs et al, 2022) means that we cannot use directional data from the glider to determine whether the suture zone is a greater source of ISW than other regions of the sub-shelf cavity. Instead, the data merely indicates that ice shelf melt is occurring at this cold cavity ice shelf. "*

l 311 "icec)"

*Corrected*

l 355 Not sure what the take away of section 5.2 is. This hydrostatic thickness is a damped version fo the real thickness, but this is known. It appears that features larger than one ice thickness are also damped, but why not. From what I understand this has never been fully formalized and will be a function of the specific strain regime. I suggest to significantly shorten this section.

*We have followed your suggestion above to shorten this section and focus on the features larger than the ice thickness which are damped as the primary finding.  This section has now been amalgamated with section 5.1.*

Which strain rates are shown in Fig 8? I guess it is transverse.

*Yes, we have now added this to the caption and figure.*

l 543 "marine Ice" -> "marine ice"

*Corrected*

**Reviewer 1**

In their TCD manuscript "The complex basal morphology and ice dynamics of Nansen Ice Shelf, East Antarctica" Dow et al. describe the morphology of the Nansen Ice Shelf by utilizing airborne and ground-based radar transects, remote sensing data on ice velocity and basal melt rates, and water temperature data from an ocean glider. Overall the authors present an interesting data set but I have the feeling a clear focus is missing throughout the manuscript. This review refers to the manuscript uploaded on 30 Apr 2023.

> *Thank you for your comments. We have now restructured the paper and added to the introduction to point more clearly towards our primary goals and findings, providing improved focus.*

General remarks:

The authors need to be more clear on the accuracy of the different data products throughout the paper. For example, as shown in Zeising et al. 2022 there are flaws in the Adusumilli data set which can be traced back to the surface velocity data set applied. I, therefore, suggest carefully checking the Adusumilli data set in the study area.

> *Unfortunately we don't have pRES instrumentation on Nansen Ice Shelf to do a similar comparison between in situ radar melt and the satellite derived melt and we are aware that there are errors associated with the satellite-derived data, which is what we point in in lines 579-592. We have attempted to be clear about the limitations of all of our datasets and our approach is that, despite the limitations of each of these approaches they all point towards the same information i.e. basal melt. We have included the Zeising reference and expanded on the potential errors by saying:*
>
> *"the satellite-derived basal melt rates signals (Adusumilli et al., 2020) are close to the noise floor (uncertainty) of the dataset for the NIS. Furthermore, the latter have been shown to differ from pRES radar measurements of basal melt at Filchner Ice Shelf due to less accurate velocity products for the satellite-based calculations (Zeising et al, 2022), and the same may be the case at NIS."*

2016 calving. The 2016 calving event is mentioned several times. As a reader I would be very interested where exactly the event occurred and what was the impact on the remaining ice shelf. Is there a detectable speedup in ice flow after the event? Are there any new fractures that can be related to the calving event?

*Yes we agree that the calving event is an interested aspect of the Nansen Ice Shelf evolution. There was a change in strain patterns (speed up) after the calving event and there was indeed a new fracture that appeared as a results, as stated in lines 51-56 and 498-506 of this manuscript. The reason we don't go into great detail about this is because it was extensively discussed as the subject of Dow et al (2018). We have referenced the latter paper multiple times through this manuscript.*

I found the discussion section hard to follow and suggest condensing it to the most important findings, which might change during the review process.

*Thank you for your suggestions. We have shortened the discussion in various sections now which will hopefully make it easier to follow.*

Specific comments:

L16: please state what parameters the ocean glider is measuring.

*We have moved the glider data to an appendix and so remove mention of it in the abstract.*

Figure 1: I suggest showing one pre- and one post-calving Landsat scene, potentially framing the calved iceberg in the pre-calving scene. Also the profiles in panel c) could be color-coded by distance from the grounding line.

*We don't focus on the calving event for this manuscript – for more details about that see Dow et al (2018). We tried color coding the profiles as suggested but the figure became too messy and we have retained our labelling method instead.*

L115: so basically the authors solely employ water temperature data? Is the temperature data resolved for the whole water column? Can you distinguish the temperature of different water depths? Please be more specific here.

*The ice shelf meltwater is fortunately the coldest water mass in the region (which is not true in West Antarctica), so yes we only employ water temperature and resolve those temperature data for the whole water column, and yes we can distinguish the temperature at different depths. We have moved the glider data to an Appendix, changed the glider data figure, and have expanded there on these points in order to clarify:*

*"This HSSW is one of the coldest water masses in the Southern Ocean (Grumbine, 1991), existing at temperatures close to the surface freezing point (~-1.9 °C, depending on the salinity; Rusciano et al., 2013; Yoon et al., 2020). Its interaction with glacial ice then produces even colder Ice Shelf Water (ISW), uniquely identifiable in the region by temperatures close to the subsurface freezing*

*point (<-1.9 °C, depending on pressure). We therefore take observations below the -1.94 °C isotherm (equivalent to the freezing point at a salinity of 34.7 and a depth of 50 m) as evidence of the presence ice shelf meltwater."*

L119-L125: if this is not a result of the ocean glider data I suggest moving this section to the site description. Are there any more references to back up this paragraph?

*Following comments from the editor we have we have moved the glider data and description of ocean conditions to an Appendix. We have added more references as suggested.*

L138: this is not really true. The REMA data are used for many other purposes, e.g. plotted against the ocean glider data in panel 2e used for ice flux calculations, etc.

*We have removed this sentence.*

L139-L140: ok, but why are the authors not including their REMA ice thickness product in this analysis? Please see also my comment on Figure 3.

*Following the suggestion of the editor we have shortened our hydrostatic analysis section and are therefore hesitant to add additional information from REMA here. We're investigating whether surface features can be used to accurately assess the basal topography, which will be more accurate with our in situ GNSS data. We tried adding in the REMA hydrostatic data here but the figures became very messy. Furthermore because the REMA data were not produced at the same time as our radar surveys there is an offset between our surface data and the REMA surface data. This can be seen in the figures below where we demonstrate the comparison and why it would be challenging to include in the manuscript.*

[Figure]

[Figure]

L156-L158: why not include the site 2 profile? This would increase the spatial resolution of the boxes.

> *We would have like to include Site 2 but we didn't get a wide enough transect at that site to either compare to Site 1 and 3 or to cover the entire width of the channel. We now note this in the text:*
>
> *"Site 2 is not included in this analysis because the radar cross-sections are not laterally extensive enough to cover the entire channel (see Fig. 2a)"*

L161: The ITS_LIVE dataset is not introduced yet. I suggest moving the flux gate section to the end of the methods chapter.

> *We have moved this section.*

L169-L170: ok, but this mass loss or gain can not be related to basal processes alone as the authors neglect smb and firn densification. Could be worth including model data here.

> *In these lines we don't state that the mass loss is from basal processes alone. We are just reporting change in the mass flux past the gates. In the results section we do an analysis of mass loss and take account of reported surface ablation rates. We don't include firn densification or snow accumulation because this is a blue ice zone as stated in lines 95, 144 and 340.*

L178: there are more velocity fields in Figure 5.

> *We have clarified that the primary analysis is for the 2016 strain rates and we now state the date range for the other strain calculations in the same paragraph.*

L178-180: yes, but this greatly minimizes geolocation and hence velocity errors in the dataset. For this reason, Alley et al. 2018 used a "stacked product including many overlapping pairs of Landsat scenes spanning many different time intervals, allowing for a significant reduction in both the geolocation and random velocity error." Interpreting strain rates from a noisy 32d image pair is rather difficult. I suggest showing pre- and post-calving strain rates from stacked velocity products.

> *We agree it is difficult and this is why we've included multiple years in Figure 5. If we attempt stacking similar to Alley et al 2018 it removes the features of interest which are on a small scale. Unfortunately without higher resolution velocity data this is the only approach for analysing small scale topographic strain features. To clarify we now include: "It is not possible to stack multiple velocity outputs in this region to reduce noise due to the combination of ice flow speed and the narrow nature of the ice shelf morphological features. As an alternative we demonstrate the consistency of transverse strain features at NIS across m*ultiple years." Lines 202-204.*

> *In terms of pre- and post calving, please see Dow et al, 2018 for strain calculations demonstrating the impact of the calving event.*

L181-183: maybe include Bindschadler et al. 1996 here?

> *We have added this reference.*

L189: I suggest an additional section about the reliability of the Adusumilli et al., 2020 dataset in the study region. Please see also Zeising et al., 2022 on this issue. Actually, I would go from the multi-mission Eulerian height-change rate time series as provided by Adusumilli et al., 2020 to your own basal melt products employing the ITS_LIVE or GOLIVE dataset, smb, and firn air content from modeling and ice thickness from REMA (validated at the survey sites).

> *We have included information about this as you suggest in the discussion":*

> *"the satellite-derived basal melt rates signals (Adusumilli et al., 2020) are close to the noise floor (uncertainty) of the dataset for the NIS. Furthermore, the latter have been shown to differ from pRES radar measurements of basal melt at Filchner Ice Shelf due to less accurate velocity products for the satellite-based calculations (Zeising et al, 2022), and the same may be the case at NIS." Lines 580-582.*

> *For the second part of your comment here we're not sure what you mean.*

Figure 3: please include also the ice thickness from REMA along the three gates.

> *See above response*

L192: based on REMA or IPR? How do both datasets agree?

*We now clarify by saying: 'Using REMA hydrostatic thickness for the wider region and our IPR data where available" In the first iteration of this manuscript we discussed how well the REMA and IPR datasets agreed but were asked to remove it.*

L268-286: this is rather methods than results. Furthermore, I doubt the reliability of this approach as already stated by the other reviewers and the editor.

*We have rearranged the manuscript to better focus on the data collected vs. data other people produced vs. calculations we applied to those data. As a result we have changed the 'methods' section to 'Datasets' and the 'results' section to 'results and analysis'. While this is a less conventional approach we believe it highlights our work and findings better than our previous configuration.*

Additional References:

Bindschadler R et al. (1996) Surface velocity and mass balance of Ice Streams D and E, West Antarctica. J. Glaciol., 42 (142), 461–475 (doi: 10.1017/s0022143000003452)

Zeising, O., Steinhage, D., Nicholls, K. W., Corr, H. F. J., Stewart, C. L., and Humbert, A.: Basal melt of the southern Filchner Ice Shelf, Antarctica, The Cryosphere, 16, 1469–1482, https://doi.org/10.5194/tc-16-1469-2022, 2022.

---

## Author Response (AR4)

Dear Authors,

thank you for another revision of this manuscript. Fortunately, we now have a positive review that suggests publication with minor revisions. I concur with this assessment and added some additional remarks which you should address prior to typesetting. This paper has had an usually long review history, but I believe in the end this has improved the paper to the degree required. I hope you feel the same way in spite of all the criticism that was raised.

We can now essentially move on with publishing this paper in The Cryosphere. Congratulations!
-- Reinhard Drews

*Many thanks for all of your work to help with this manuscript. We have addressed the*

*comments below as indicated by our responses in italics.*

l 18 missing comma "ice shelf draft, the cause and ..."

*Addressed*

l 24 "The combination of thinner ice ...." this is a long sentence which I found hard to digest. Split in two sentences? Also, can you find a better word than "complex" strain rates and "complex" morphology? I believe in terms of morphology you mean that it is "spatially variable on sub-kilometer scales". In terms of strain rates, do you refer to the "along-flow banding" seen in Fig 4b? I don't think I found the best words either, but "complex" by itself doesn't really contain useful information.

*We have changed the sentence to say:*

*"Enhanced melt rates near the ice shelf terminus and in steep regions of the channelized suture zone, along with relatively thin ice in the suture zone, appear to represent vulnerable areas in the NIS. This morphology, combined with ice dynamics, induce strain that has led to the formation of transverse fractures within the suture zone, resulting in large-scale calving events."*

*We have also removed/replaced the word 'complex' in all instances in the abstract.*

l 55 "thinning rates .. "

*Addressed*

l 62 to you mean "topography" or "thickness"? My guess is the latter.

*Following the comment from the reviewer we have changed to: "how does basal morphology relate to surface topography and influence the distribution of marine and meteoric ice?"*

l 99 "bed reflection" do you mean "reflection from the ice-ocean interface" or "ice base". Bed suggests sediments/bedrock to me.

*Good point – we have changed to 'ice base'*

l 144 "demonstrate I consistency " what does the "I" do in there ?

*Addressed*

l 201 I don't think "if" is the right wording but "how".

*Addressed*

l 205 "horizontal compression" --> "transverse compression" to be consistent with what is described in the legend of Fig. 4.

*Addressed*

eq (3) make sure that that the divergence is adequately done in typesetting. Currently it is not.

*We will make sure this is correct in the final version – it looks fine in our word version.*

l 263 "from horizontal" incomplete sentence

*Changed to 'horizontal flow'*

Please check the possibility of one column vs. two column Figures. Some Figs (e.g. Fig 3,7) are somehow in-between both options.

*All figures are now sized correctly for 1 vs. 2 columns (3,6 and 7 are now 1 column for example).*

**REVIEWER 1**

The authors describe and interpret new ice thickness transects from the Nansen Ice Shelf in combination with satellite datasets and oceanographic glider data to infer the quality of satellite-derived estimates of thickness. They estimate the distribution of basal melting and comment on the origin and evolution of basal fractures on the ice shelf.

The paper is clearly structured and interpretations are largely supported by the data. Parts of the abstract are difficult to follow, but in general I would recommend publication subject to some minor, largely typographical corrections, listed below:

> *Thank you for your review. We have addressed your comments as shown below.*

Line 15: "…ability of the ice shelf to buttress…"

> *Addressed*

Line 18: "…variations in ice shelf draft, the cause and effect…"

> *Addressed*

Line 19: The phrase "near the onset of floating ice convergence in the suture zone" is slightly confusing. Does this mean at the grounding line? I suggest a rephrase.

> *Addressed: "*We find that the Nansen Ice Shelf has a highly variable basal morphology driven primarily by the formation of basal fractures near the onset of the ice shelf suture zone".

Line 23: This sentence implies vertical variability in thickness, which doesn't make sense.

> *We have removed this reference to vertical variability.*

Line 33: I would use "under" instead of "undergoing"

> *Addressed*

Line 42: "…(e.g. the Global Land Ice…)"

> *Addressed*

Line 50: Consider splitting into two sentences.

> *We have now split the sentence.*

Line 61: The research questions listed here are quite vague. In particular 'What morphology does the ice shelf have' is somewhat unscientific. I would suggest rewording to something like 'how does basal morphology relate to surface topography and influence the distribution of marine and meteoric ice'

*We have changed as suggested*

Line 79: "…a supraglacial river…"

*Addressed*

Line 85: "…winds, which also strip the NIS…"

*Addressed*

Fig 1c: The fact that all sites are coloured the same colour slightly confusing, but I admit this may be a bit picky.

*We have kept as the same colour because of difficulty differentiating the colours on the satellite image if we changed them (plus problems fitting it all into the legend)*

Fig 2: I would avoid using black lines to denote both the transects and the fracture.

*We have changed the color of the fracture.*

Line 144: "…demonstrate consistency…"

*Addressed*

Fig 4: The blue-red colour scale here seems backwards to me (red normally implies positive values?), although this is clearly just a personal preference as there are other examples where red is negative.

*It's fairly standard to have red as compression even though we agree it's a bit counter intuitive, but we have retained as is to stay with convention.*

Line 259: Given that these cited surface ablation measurements are used extensively for calculation of basal melt, it would be valuable to discuss a bit about the potential uncertainty and what this would do to your overall conclusions.

*We have added in the following: "good correspondence between the estimated cumulative thinning rate and the measured thinning, particularly at the northern margin of the suture zone, suggesting this higher surface ablation rate is the more accurate estimate when averaged over time." Since we use two different estimates and plot the outputs in Figure 6 we think this demonstrates the uncertainty in surface ablation relative to the basal melt rates and strain thinning. Compared with our radar thickness change and other calculations, the higher ablation rate appears fairly accurate over time. Without past data on ablation rates its not possible to be more exact within the framework of this manuscript.*

Line 271 & 246: The time to travel between sites is inconsistent between these two lines.

*We have corrected the second time to the more accurate one of 123 years.*

Line 284: "Thickness change due to strain thinning reaches a maximum of 13 m between Site 1 and Site 3."

> *Addressed*

Line 302: I would split this sentence for readability.

> *Addressed*

Line 504: "Here, the limited width of the…"

> *Addressed*

Fig C1 caption: "(d) Temperature profiles from the…"

> *Addressed*